# Large Scale Transfer Learning for Tabular Data via Language Modeling

Josh Gardner[♮,*]      Juan C. Perdomo[#]      Ludwig Schmidt[♮,♭]

[♮]**University of Washington,** [#]**Harvard University,** [♭]**Stanford University**
[*]Corresponding author, `jpgard@cs.washington.edu`

## Abstract

Tabular data – structured, heterogeneous, spreadsheet-style data with rows and columns – is widely used in practice across many domains. However, while recent foundation models have reduced the need for developing task-specific datasets and predictors in domains such as language modeling and computer vision, this transfer learning paradigm has not had similar impact in the tabular domain. In this work, we seek to narrow this gap and present TABULA-8B, a language model for tabular prediction. We define a process for extracting a large, high-quality training dataset from the TabLib corpus, proposing methods for tabular data filtering and quality control. Using the resulting dataset, which comprises over 2.1B rows from 4.2M unique tables, we fine-tune a Llama 3-8B large language model (LLM) for tabular data prediction (classification and binned regression) using a novel packing and attention scheme for tabular prediction. Through evaluation across a test suite of 329 datasets, we find that TABULA-8B has zero-shot accuracy on unseen tables that is over 15 percentage points (pp) higher than random guessing, a feat that is not possible with existing state-of-the-art tabular prediction models (e.g. XGBoost, TabPFN). In the few-shot setting (1-32 shots), without any fine-tuning on the target datasets, TABULA-8B is 5-15 pp more accurate than XGBoost and TabPFN models that are explicitly trained on equal, or even up to $16\times$ more data. We release our model, code, and data along with the publication of this paper.[1]

## 1   Introduction

Transfer learning - the ability of a model to accurately solve prediction tasks on data it was not trained on - is one of the defining hallmarks of recent foundation models in domains such as vision [38] and language [6]. Among their many advantages, transferable models expand the scope of problems that can be tackled via machine learning by reducing the need for curated, task-specific models and datasets. Such models also can provide both absolute performance and sample-efficiency gains over task-specific models when applied to new tasks [38, 41, 53].

In this work, we introduce a new model and dataset for large-scale transfer learning on tabular data. Tabular, spreadsheet-style data underlies applications in healthcare, finance, government, and the natural sciences [4, 16, 50].

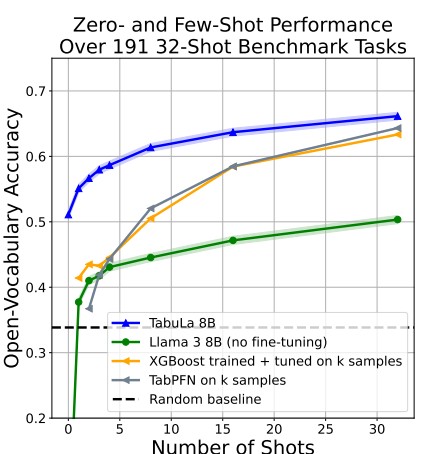

Figure 1: TABULA-8B outperforms SOTA tabular baselines across $0 - 32$-shot tasks from five tabular benchmarks.

---

[1]For links to all code, data, and model, see Section 7.

38th Conference on Neural Information Processing Systems (NeurIPS 2024).

Yet, despite the potential impacts of transferable foundation models for tabular data [58], the core practices of machine learning on tabular data have remained largely unchanged. The prevailing paradigm is still to train single-task models (e.g., XGboost [7]) using a fixed schema on data from the same distribution on which the model will be deployed.

Here, we aim to bridge this gap. We introduce TABULA-8B, a language model for tabular prediction which can flexibly solve classification tasks across unseen domains, including where data is *scarce*. Our methodology expands the scope what is possible in these settings, thereby democratizing access to prediction in low-resource contexts and providing state-of-the-art, training-free transfer learning on any tabular data. Since the model only requires a forward pass to perform inference on the target data, it also avoids the privacy or computational considerations that arise in other approaches that require fine-tuning on local, and potentially sensitive, datasets.

In particular, given a small number of examples (shots), *and without any fine-tuning on the task*, TABULA-8B outperforms state-of-the-art gradient-boosted decision trees and tabular deep learning methods *that are explicitly trained on the target data* (see Figure 1). Furthermore, TABULA-8B is capable of zero-shot prediction, a behavior which is not possible using these prior methods. To enable these results, we build a new dataset for tabular prediction, The Tremendous TabLib Trawl (T4), that allows us to scale up training by several orders of magnitude ($10,000\times$ more data) relative to previous work.

## 1.1 Our Contributions

This paper has the following three main contributions:

**TABULA-8B, a Tabular Prediction Model:** We build TABULA-8B (Tabular Llama 3 - 8B), a model for prediction on tabular data. On an evaluation suite consisting of 329 tables drawn from five tabular benchmarks, TABULA-8B has zero-shot accuracy 17 percentage points (pp) above random guessing. In the few-shot setting (1-32 examples), TABULA-8B is 5-15 pp more accurate than state-of-the-art methods (XGboost, TabPFN, CatBoost) that are trained on equal number of shots, and these methods require 2-8$\times$ more data to achieve the performance of our model. TABULA-8B outperforms a variety of strong tabular baselines and even commercial LLMs such as Claude Instant and Claude 3 Sonnet.

**T4 - A Large Scale, High Quality Training Dataset:** We build and release The Tremendous TabLib Trawl (T4), a filtered collection of 4.2M unique tables (consisting of over 2.1B rows, a total of 100B tokens) from TabLib [13]. We detail the recipe used to construct T4, including a suite of methods for filtering web-scale tabular data at several levels (table, row, column), removing unwanted information such as PII and code, and selecting unsupervised *prediction targets* from these tables.

**Open-Source Release:** As part of our publication, we release all relevant infrastructure (code, models, and data) with the hopes that the community will build on our work. We provide high-quality, efficient implementations of data pre-processing and model training pipelines, including our new row-causal tabular masking (RCTM) attention and packing scheme for training on tabular data. We also share the code used to filter T4 from TabLib, enabling future work that extends our dataset construction methodology.

## 1.2 Preliminaries & Project Scope

Our work is concerned with *prediction* models for *tabular data*. We define both below.

**Tabular Data:** For our purposes, tabular data has three main properties. ($i$) *Structured*: It consists of elements with a "key-value" structure, often represented as a table with keys (or "headers") representing column names, and rows that consist of values. ($ii$) Heterogeneous: The values are of mixed types, including numeric, boolean, categorical, ordinal, text, date/time, etc. Missing values may be present. ($iii$) Exchangeable: The ordering of rows and columns in the dataset is arbitrary. In particular, any permutation of the rows, or columns, still represents the same tabular dataset.

**Prediction Task Definition:** The main focus of this work is *prediction* on tabular data. In tabular prediction, the goal is to predict the value $y$ of a specific target column for a row in a dataset using the key-value pairs $x$ from all other columns. More specifically, we focus on classification tasks where values $y$ for the target column belong to a finite set $C$. Binned regression tasks, in which a real-valued $y$ is discretized into a finite set of numeric values (as in [59]) also fit this definition.

## 2 Related Work

Our work builds on a line of foundation modeling, tabular data prediction, and natural language processing research. Given space constraints, here we focus on the most closely related literature.

**Transfer Learning and Foundation Models:** The idea of building general purpose models via autoregressive next-token prediction on large scale datasets was pioneered in a series of papers in natural language processing [6, 11, 36, 40, 57]. These results have since led to the development of foundation models capable of solving diverse tasks in other modalities including vision [38, 63], audio [39, 66], code [22, 42], time series [10, 18, 21], and graphs [62], as well as multi-modal models [53, 55]. Our work also build upon on the demonstrated capacity of transformers to perform few-shot or in context-learning [6, 17], which entails making predictions on examples from a previously unseen dataset, given only a few labeled examples from that task.

**Large-Scale Dataset Curation:** The construction of large, high-quality datasets has emerged as one of the most critical, and challenging, issues in the development of transferable models. Several major milestones in this space [6, 40, 41, 53–56] stand out in their effort spent curating and cleaning web-scale datasets – often while using a model architecture and training recipe that only slightly differs from prior work. This has led to a number of modality-specific methods for large-scale dataset curation; for example, the use of heuristics [41] and model-based quality scoring to select high-quality text data [6, 9, 56]; methods for selecting aligned audio-transcript pairs for speech [39]; or the use of CLIP scores to filter for aligned image-text pairs [46]. However, to the best of our knowledge no prior work has developed analogous methods for *tabular* data. This lack of large-scale training data has been a critical bottleneck toward the development of tabular foundation models.

**Models for Tabular Prediction:** Despite the fact that deep learning methods are now the norm in domains such as computer vision or NLP, methods based on gradient-boosted decision trees (GBDTs) [5, 7, 37] continue to be at or near state-of-the-art in tabular prediction [20]. Drawing upon recent breakthroughs in other modalities, the field has now developed deep learning-inspired approaches [19, 52] that are competitive with tree-based models *in-distribution*, where models are trained and evaluated on the same data, but the benefits of such approaches relative to GBDTs appears to be limited in practice [20, 33]. In particular, [25] introduces TabPFN, a transformer model for tabular data that outperforms XGBoost in certain regimes [33] and is capable of making predictions on unseen datasets (with some constraints on dataset size and label space; see D.2). A related recent work, CARTE [30], explores the use of graph-based architectures for tabular transfer, based on key-value encodings and pretrained on a large knowledge graph.

Several recent works [12, 23, 59, 66] have explored fine-tuning LLMs on individual tables, or on small collections of tables ($< 200$). The main idea in this closely related line of work is to reduce classification to next-token prediction by first *serializing* a row as text (see Figure 2b for an illustration) and then training an LLM to predict the serialized labels. These studies demonstrate that this LLM approach is often competitive with trees or tabular deep learning methods in-distribution [12, 23, 59]. However, in cases where models were evaluated out-of-distribution, they were less accurate than SOTA methods trained on these held-out tables [65]. Our work builds on this promising line of work. Relative to these prior efforts, we specifically address the (1) lack of large-scale and training data; and (2) the inability of exiting methods to be competitive when evaluated out-of-distribution.

## 3 TABULA-8B - Model Design and Training

Our overall approach is to fine-tune the pretrained Llama 3-8B language model [54] on tabular prediction tasks using a new web-scale corpus, T4. We use Llama 3-8B as our starting point since it is a high-quality, open-source model trained on over 15T tokens that demonstrates strong performance on a diverse set of downstream tasks [54], particularly at its relatively modest size (which makes fine-tuning, inference, and deployment more accessible).

**Serialization and Tabular Language Models:** As discussed previously, our methodology extends ideas pioneered in previous work [12, 23, 59, 65] demonstrating how LLMs can be trained to perform tabular prediction tasks by serializing rows as text, converting the text to tokens, and then using the same loss function and optimization routines used in language modeling. *Serialization* refers to the procedure of converting a row of data into text, for instance by concatenating substrings of the form "the <key> is <value>". Prior works investigated the impact of different serialization formats

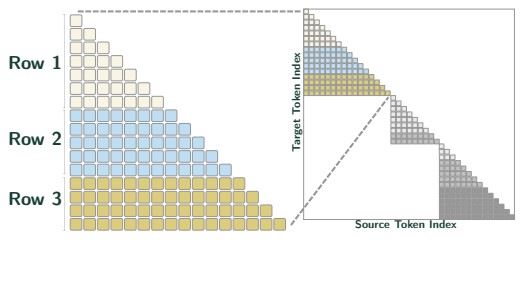

Figure 2: 2a: Illustration of the row-causal tabular mask (RCTM) representing a batch during training. Each triangular block represents potentially many rows from a *single* table (detail shown at left). Shaded groups within this block represent tokens from one row in the table. This structure implicitly trains the model for few-shot learning by permitting it to attend to previous rows from the table, but not to rows in other tables. 2b: Serialization of tabular data into text. The model is trained to produce the tokens following the <|endinput|> token.

[12, 23, 51], demonstrating that performance is largely insensitive to the exact mapping (e.g. using "{ <key>: <value> }") and other strategies do not improve upon this "the <key> is <value>" structure.

We adopt a similar serialization strategy, illustrated in Figure 2b. Given a row of data from a table, the corresponding serialization has three main parts: (*i*) a *prefix* containing a prompt (always "Predict the value of <target column name>") followed by a list of possible label values ("val1 || ... || valNumClasses ||), (*ii*) the *example* consisting of all key, value pairs for the columns used as features, and (*iii*) a *suffix* prompting the model with a question ("What is the value of <target column name>?") again followed by the possible labels. For multiple-shot samples, we concatenate their serializations. We introduce three special tokens into the Llama 3 vocabulary to ensure these sequences are properly tokenized: ||, to delimit answer choices; <|endinput|> to denote the end of an input sequence (the last token before the targets or model generation begin); and <|endcompletion|> to indicate the end of a completion.

**Training Procedure:** We train TABULA-8B using a standard language modeling setup where the model is trained to minimize the cross-entropy over the sequence of target tokens. We only compute loss over the subsequence of target tokens: the tokens starting after the <|endinput|> token, up to and including <|endcompletion|>. This objective focuses training on learning the desired target label, as in [12, 23, 59], rather than developing a broader generative model of tabular data as in [65].

Relative to prior studies on tabular prediction with LLMs, our work has one main methodological innovation. We introduce an efficient attention masking scheme, row-causal tabular masking (RCTM), tailored to few-shot tabular prediction whereby the model is allowed to attend to all previous samples from the same table in a batch, but *not* to samples from other tables (this is sometimes referred to as "cross-contamination" in the language modeling literature [31]). However, by appropriately masking out values, RCTM also enables packing examples into the same batch (as effectively zero padding is required during training despite the large variance in the size of each tokenized table or row), thereby increasing model throughput (see Figure 2a). Taken together, these have the effect of training the model to use multiple "shots" during training and mitigates the potential loss of few-shot learning capabilities that has been observed to occur during fine-tuning [29, 61, 64].

The RCTM masking structure is shown in Figure 2a. Lower-triangular blocks correspond to rows from the same table that are present in the batch. This is similar to the "in-context pretraining" method from [48], except that (*i*) our procedure encourages the model to aggregate information across multiple *rows* of a given *table*, rather than attending across documents, and (*ii*) our procedure only trains the model to predict the *target* tokens, not the input features. We show that RCTM has a drastic impact on few-shot performance through an ablation experiment (see Section F.1).

**Training Details:** The final model is trained for 40k steps with a global batch size of 24 (with sample packing, this is roughly equivalent to a global batch size of 600 rows of tabular data). The model

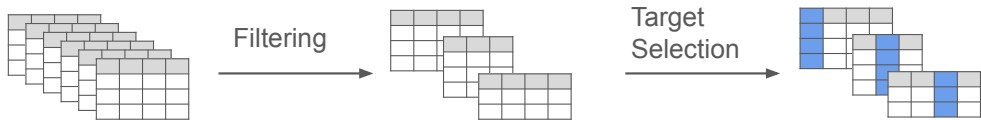

Figure 3: Sketch of dataset generation pipeline. 627M tables from TabLib [13] are filtered by applying rules at the table, row, and column level. Then, for each table, we identify valid and high-quality prediction targets in an unsupervised manner and use the results for training TABULA-8B.

sees roughly 8B tokens during training; we note that is less than 10% of the 100Btokens in T4, and less than one one thousandth of TabLib itself. We fully fine-tune all model parameters, as opposed to parameter-efficient fine-tuning, since full-fine tuning consistently benefits from scale [24, 64]. Reproducibility details are given in Appendix B.

## 4    Dataset Construction: Building The Tremendous TabLib Trawl (T4)

Beginning from a web-scale corpus of raw data (TabLib), we apply various filters to produce a high-quality subset, and transform the results into a set of prediction tasks for training. As the result of this procedure, we produce T4 (The Tremendous Tablib Trawl), which we release with this paper.

**Original Raw Data Source:** TabLib [13] is a publicly-available dataset consisting of 627M tables extracted from two main sources: Common Crawl and Github (see [13] for more details). Due to its scale and diversity, TabLib presents a unique opportunity for training foundation-scale models on the tabular data. However, like other web-scale datasets, the vast majority of its contents are low quality and not suitable for training. For instance, TabLib contains numerous system logs with inscrutable statistics, tables of software documentation, and call sheets with personally identifiable information (PII). To the best of our knowledge, no previous work has addressed the task of filtering TabLib into a usable training set, and no publicly-available models have been trained on this corpus.

**Filtering Strategies:** Filtering large collections of raw data to extract a higher-quality subset is an essential component in the development of foundation models [e.g. 41], yet to date, no previous work has addressed this core issue for tabular data. Filtering a web-scale dataset like TabLib into a usable subset of high-quality tables is critical to leverage its diversity and scale, but also raises unique challenges specific to tabular data, such as missing data, web "content" that is formatted as HTML tables that does not satisfy our definition of tabular data, and PII. To turn TabLib into a usable training set, we develop a set of filtering methods to identify high-quality tables for prediction. Conceptually, our filtering occurs at three levels, each applied sequentially: *tables* (entire tables are removed from the pool), *columns* (individual columns are removed from a table), and *rows* (rows are removed from a table).

Similar to previous approaches [38, 40, 41, 54, 56], we use a mix of heuristics and rule-based methods to remove low-quality sources from our pool. We present the full list of our filtering rules in Section A. At a high level, our emphasis across all filtering strategies is to: (1) remove non-tabular data (such as text or PDFs incorrectly identified as tabular data during TabLib's collection), (2) ensure the *safety* of chosen tables (e.g. by removing PII), and (3) find sources with *high semantic content* (e.g. by removing tables with too many missing values). As part of this filtering process we develop and apply simple methods for deduplication, English language filtering, filtering for missing data, PII removal, code removal, and more.

**Unsupervised Task Selection:** As described in Section 1.2, we focus on methods for tabular prediction: predicting the value of a target column given the values of all other columns for an instance. Therefore, as part our data pipeline we develop new methods for selecting, in an unsupervised fashion, which column is the *target column* for each table in the corpus. Selecting targets of prediction for tabular data at scale is an under-explored problem. Prior work in this space operated on at most a few hundred tables and used either a combination of expensive queries to commercial LLMs or manual curation to identify tabular prediction targets [59, 65]. However, when operating on hundreds of millions of distinct tables with potentially no associated metadata, these strategies are not feasible.

For each table, we select a prediction target programmatically by first identifying a subset of columns that are suitable for prediction according to various heuristics, and then choosing a specific column

at random from this set. The exact list of heuristics to arrive at this set is presented in Appendix A. Amongst others, these include excluding candidate columns if: the column name is numeric, it has only one unique value, or it has unique values for every row (excluding numeric columns).

**Final T4 Dataset Summary:** Running this entire filtering process (from raw data to serialized examples ready for training) on all 70TB of TabLib using our open-sourced implementation takes about 4 hours on a CPU cluster. It yields a total of 4.2M tables, which equates to a table filtering rate of approximately 97.91%. Additional descriptive statistics for the dataset are given in Appendix A.3. The resulting dataset contains over 2.1B rows (approximately 100B Llama 3 tokens) for training of the downstream model, and occupies roughly 2TB compressed on disk. We note that 100B tokens is larger than the total number of tokens TABULA-8B sees during training. Therefore, the model sees each distinct table at most once during training, and our pipeline could be scaled up to support larger models or longer training runs.

# 5 Experimental Results

## 5.1 Evaluation Methodology

We evaluate the transfer learning performance of TABULA-8B on a diverse set of established benchmarks previously considered in prior work (see Section 5.2 for a list). For each dataset, we use the predefined prediction target from the original benchmark. Due to computational constraints, we evaluate TABULA-8B on up to 128 test examples for each dataset and number of shots $k$.

The term "few-shot" is unfortunately overloaded. It is used both to refer to models that make predictions on instances never seen during training, and to models that directly *train* on these examples before predicting on unseen samples. We do not fine-tune our model on test examples, in contrast to [23, 59]. Our methodology only requires performing forward passes through the network to generate predictions and avoids the need for computationally-expensive gradient updates. In zero-shot evaluations, given a row of a dataset along with the corresponding set of columns and possible labels, we first serialize the row into the same format used during training, and feed it into the model to generate a prediction following the `<|endinput|>` token. For few-shot evaluations, we perform the same procedure, except that we preprend the serialized "shots" as in Figure 2b.

In contrast to methods like XGBoost that directly predict likelihoods of a set of labels, language models output likelihoods over a set of tokens (128k in the case of Llama 3). For each evaluation dataset, the values in the label set (e.g. "sun, rain, snow" in Figure 2b) can consist of a *sequence* of many individual tokens from this large vocabulary. Here, we use *open-vocabulary* (or "open-ended") accuracy [1, 8] as the main evaluation metric for our model. In this setup, once the model is prompted with a serialized example, it is allowed to generate an arbitrary sequence of tokens. Once it produces the `<|endofcompletion|>` token, the generated text is then directly compared to the correct completion. Only an exact match, including the terminating `<|endofcompletion|>` token, is counted as accurate. This is more challenging than *closed*-vocabulary evaluation, where the model is only rated on assigning the highest probability to the correct completion from a predetermined set.

## 5.2 Evaluation Datasets

We evaluate our model's predictive performance across a collection of 329 publicly-available tabular datasets drawn from five tabular benchmarks (see Appendix H for a full list). These include:

**UniPredict Benchmark (169 datasets) [59]:** We use the "supervised" subset of 169 datasets from the recently-introduced UniPredict benchmark. These are high-quality tabular datasets with generally informative column names and a mix of both categorical and continuous targets, drawn directly from Kaggle. While the model introduced in Wang et al. [59] was trained and tested on separate splits of these datasets, we only use them for testing. We make corrections to several datasets with targets erroneously treated as categorical in the original benchmark, described in Section 5.2.

**Grinsztajn Benchmark (45 datasets) [20]:** The Grinsztajn benchmark is a curated suite of datasets consisting of numeric and categorical features. This dataset is notable in that the original study by Grinsztajn et al. found that gradient boosted decision trees (GBDTs) consistently outperformed deep learning-based methods on these tasks.

**AutoML Multimodal Benchmark (AMLB) (8 datasets) [49] :** A suite of tables which include one or more free-text fields (such as an Airbnb description, or a product review). The benchmark is considered challenging for tree-based methods due to the non-standard text-based features. However, it also poses a challenge for LLMs since some columns can contain highly variable lengths of text.

**OpenML CC-18 Benchmark (72 datasets) [2]:** The OpenML Curated Classification Benchmark was created by applying filtering rules to extract a high-quality subset from the OpenML platform. The rules include: no artificial data sets, no subsets of larger data sets nor binarizations of other data sets, no data sets which are perfectly predictable by using a single feature or a simple decision tree.

**OpenML CTR-23 Benchmark (35 datasets) [14]:** The OpenML Curated Tabular Regression (CTR) Benchmark is a curated set of tables for regression drawn from OpenML. The curation process is similar to that of OpenML-CC18, for regression tasks. We note that, being primarily intended for the evaluation of AutoML methods, the OpenML benchmarks are notable for lacking informative column names (i.e. names such as "`Var1, Var2, ...`" are common in OpenML benchmark datasets).

We transform all regression tasks into a 4-class classification based on quartiles, as in [59] (see Appendix A.2 for details). Many datasets contain rows with missing data. We leave these as-is and do not remove any data. On a computational note, some datasets contain a large numbers of features and performing $k$-shot evaluations on these datasets at large $k$ can exceed the model's context window. Therefore, in few-shot evaluations, we always report results for the subset of datasets where $k$ shots fit into the model's context window, for the entire specified range of $k$. We provide more details on the evaluation datasets in Appendix D.1 and report per-dataset results in Appendix H.

## 5.3 Baselines

When inspecting TABULA-8B's performance, we compare against the following baselines:

**Llama 3-8B [54]:** This is the base model from which TABULA-8B is fine-tuned. Comparing to the base model isolates the effects of the fine-tuning process. It also controls for any contamination of evaluation datasets that may be contained in pretraining data for Llama 3 (the exact training data for Llama 3 are not currently disclosed). We return to this point in Section 5.6.

**XGBoost [7]:** XGBoost is a supervised learning gradient-boosted decision tree (GBDT) method. It is widely considered to be highly competitive in tabular prediction tasks [15, 20, 33].

**TabPFN [25]:** This a transformed-based hypernetwork pretrained to reflect a set of inductive biases germane to tabular data. TabPFN is thus considered especially effective for few-shot learning [25, 33].

Whenever possible, we perform hyperparameter tuning on XGBoost and TabPFN in order to maximize their performance. See Appendix D.2 for further details on baseline implementation and tuning. We also provide results comparing to additional supervised baseline models, and to commercial LLMs, in Section E.2.

## 5.4 Main Results: Assessing TABULA-8B's Transfer Learning

We present our main experiments evaluating the transfer learning ability of TABULA-8B in Figure 4. As a whole, TABULA-8B demonstrates strong transfer performance across the broad range of tasks.

In the zero-shot regime (seen in the left-most point for each plot in Figure 4) – where the model is presented with no further information about the target dataset except for the serialized key-value pairs and set of possible labels for a single row –TABULA-8B is between 5 to 25 pp more accurate than a random baseline and 50 pp more accurate than the base Llama 3 model. This illustrates one of the key benefits of using language models for tabular prediction: after fine-tuning, TABULA-8B can leverage semantic information contained in the serialized data to make high-quality predictions.

While XGboost and TabPFN are not capable of zero-shot prediction, this behavior has been observed in the original Llama 3 model [54, 57]. However, in our evaluations, Llama 3 performs below random guessing in the zero-shot setting. We hypothesize that the base Llama 3 model requires a small number of samples to understand the input-output format and task (as indicated by the large leap in Llama 3 performance from $0 \rightarrow 1$ shot).

In the few-shot setting, where each method additionally sees a small number of labeled examples, TABULA-8B's performance steadily improves with the number of shots. In the regime of 1 to 32

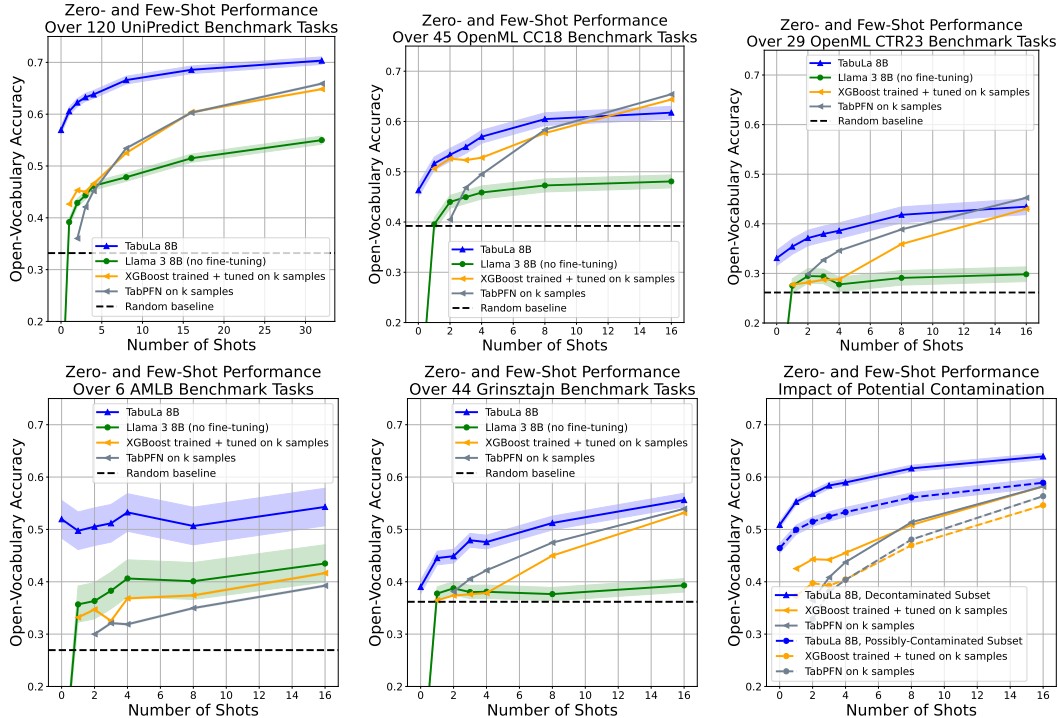

Figure 4: Zero- and few-shot accuracy across five tabular benchmarks. For each benchmark, we evaluate on all tasks, but in the figures above we only display the subset of tasks where $k$ shots fit into the 8192-token context window of TABULA-8B. Complete results are in Supplementary Section H. The final plot (lower right) shows curves separately over decontaminated vs. potentially-contaminated evaluation tasks (see Section 5.6); we find no impact on our overall findings due to contamination (and performance on tasks which may be in our training set is *lower* on average, across all models).

shots, it outperforms state-of-the-art models (XGBoost and TabPFN) that are directly trained (and hyperparameter tuned) on each specific dataset by 5-20pp. Once we evaluate performance on 32, 64, or 128 shots (see Figure 7b), this gap begins to diminish, but the number of datasets that can fit > 32 shots into the 8192-token context window is both small and a relatively biased sample (due to their small number of features). TABULA-8B is consistently 10 to 20pp above the Llama 3-8B base model for the full range of shots, highlighting the benefits of our training procedure on T4.

**Improvements in Sample Efficiency:** As discussed previously, the main benefit of transferable models is that they reduce the amount of data necessary to achieve good performance on new tasks. For instance, as seen in Figure 4, TABULA-8B only needs one shot to achieve 60% average accuracy on UniPredict tasks. However, both TabPFN and XGBoost only reaches 60% accuracy after 16 shots. Therefore, TABULA-8B reduces the amount of data necessary to achieve 60% accuracy by 16 fold relative to XGBoost annd TabPFN. We refer to this statistic as the relative sample efficiency (see D.3). TABULA-8B in general achieves higher accuracy than the benchmarks using less data. Hence, the relative sample efficiency is always > 1 (the exact ratio varies across benchmarks).

**Impact of Informative Column Headers:** As shown in Figure 4, while TABULA-8B generally has higher accuracy than the baselines, this accuracy gap varies across benchmarks and the number of shots. For instance, for the UniPredict benchmark – which was specifically constructed to include datasets with semantically-meaningful column headers [59] – the gap to supervised baselines is much larger than in the OpenML benchmarks, which tend to have less semantically-meaningful column names. If meaningful column headers are absent, the model still performs well (matching or outperforming XGBoost at shots $k \le 8$), but its advantage over these strong baselines is lessened. We investigate this effect in further detail with a controlled experiment in Section F.2.

## 5.5 Further Robustness Evaluations and Ablation Experiments

**Robustness to Column Ordering.** Apart from evaluating TABULA-8B's transfer learning ability, we also investigate its *robustness* and the degree to which performance is affected by the order in which columns are presented (serialized); this order invariance is cited as a necessary attribute of tabular foundation models in [58]. We present these experiments in Appendix F.4. Our results demonstrate that changing column order does not alter performance in a statistically significant way, but that there may be a small ($\sim$ 1pp) drop which we hypothesize is due to manual ordering of tabular columns in certain benchmark datasets which sometimes reflects a more "natural" ordering.

**Robustness to Feature Dropout.** Language models may be uniquely susceptible to small changes in the downstream data; for example, the removal of specific features may affect language models' prediction performance more than traditional supervised methods. We conduct an ablation study to assess the behavior of TABULA-8B as columns are removed from the test data. We assess removal both in order of descending and ascending importance. The results of these experiments, in Appendix F.3, demonstrate that TABULA-8B's performance declines at a similar rate to an XGBoost model trained directly on the subset of features.

**Robustness to Column Header Removal.** Another potential risk of tabular language models is that, while these models are able to utilize the semantic information in column names, the model may also be overly reliant on the presence of informative column names. In Appendix F.2 we conduct an ablation study to assess this. The results in Appendix F.2 demonstrate that there is a small decline in performance when column headers are removed (replaced with uninformative headers), but that TABULA-8B still outperforms baselines across all numbers of shots. We believe that this drop in performance is commensurate to the loss in information when column headers are eliminated.

**Importance of Row-Causal Tabular Mask.** We evaluate the impact of the attention masking scheme introduced and described in Section 3. We conduct an ablation study, replacing this component of the model with a sample-wise causal attention (the same form of attention used during standard language model training, where attention across documents is prevented). Our results, detailed in Appendix F.1 and Figure 11, illustrate that this modification is central to the few-shot learning capabilities of TABULA-8B: when our mechanism is replaced with sample-wise attention the resulting model does not demonstrate few-shot learning capacity, and its performance degrades for $k \geq 16$ (see Figure 11).

**Influence of the Base LLM.** We conduct an ablation study of the base LLM to evaluate how TABULA-8B improves as the base LLM improves. In particular, we rerun our main training pipeline described in Section 3, but using LLama 1 and 2 as the initial language model instead of LLama 3. These results, provided in Section F.6, demonstrate that TABULA-8B improves along with the performance of the underlying base model. Taken together, these results highlight how the primary contribution of the paper is not the specific model we produce, as much as it is the methodology we present for generating tabular predictors from base language models. As LLMs continue to improve, so will the tabular models that are produced by applying our training methodology to new LLMs.

## 5.6 Assessing the Potential Impact of Data Contamination

Given that T4 consists of 4.2Mtables sourced from public data sources (Common Crawl, Github) and that our evaluations are also comprised of public benchmarks, we investigate the extent and possible impact of data contamination – that is, training datasets that are part of the evaluation suite. In Section G, we explain our methodology to test for the potential presence of benchmark datasets in T4. Using a conservative identification strategy based on column matching (likely to include false positives). We find at most one-third of benchmark tables may occur at least once in the training set. When training large-scale models for transfer learning, it is not always clear *a priori* what the eventual application domains will be. Therefore, we believe that it is an important research question to understand the extent to which contamination may affect performance, as contamination may be difficult to prevent in some cases. Initial foundation modeling efforts in non-tabular domains adopted a similar approach, and found mixed or no impact from overlap [38, 40].

We evaluate the impact of contamination in our experimental setup by evaluating TABULA-8B separately on "potentially contaminated" vs. uncontaminated tables. Our results are shown in the bottom right plot of Figure 4, as well as in Figures 17 and 18. Summarizing, we find no clear evidence that contamination affects model performance on the test suite, or that transfer ability is affected by contamination. In fact, as seen in Figure 4, the gap between TABULA-8B and XGBoost is in fact

*larger* if we restrict evaluation to the benchmark tables which we verify are not in T4. In addition to verifying that our results continue to hold over a diverse set of tables which we know the model did not see in training, it also shows that having some amount of potential contamination did not upwardly bias our estimate of TABULA-8B's transfer learning ability. We hypothesize that the observed gap in Figure 4 is due to our conservative duplication procedure being more likely to flag datasets with generic or common column names, which also leads to *worse* baseline performance on these tasks. We present more comprehensive investigation on the effects of contamination in Appendix G.

## 6 Discussion

**Limitations:** TABULA-8B has several limitations. First, TABULA-8B has a limited context window of 8192 tokens. This restricts the number of examples that can be utilized for few-shot learning, as well as the additional information (such as text context or extended feature descriptions) that are available to the model. We expect that this limitation will be eased as the availability of longer-context models grows [e.g. 53]. Second, TABULA-8B has 8B parameters, which makes serving and inference expensive and limits the environments it may be deployed in. Lastly, given that it uses a pretrained LLM as a base-model and fine tunes on a web-scale corpora of historical datasets that likely contain various social biases, TABULA-8B introduces new potential fairness considerations that are not present when using preexisting supervised methods such as XGBoost. We hope that by open sourcing the model, data, and code, we might enable future research addressing these important directions.

**Future Work:** Our work on transfer learning for tabular data is the first its kind at this scale, and there are several avenues for future research. These can be coarsely categorized as either improvements (of the existing dataset and model) or extensions (deeper investigations into the model and data itself).

On the *improvements* side, we see several promising directions. These include improvements in tabular data filtering (this has been the main axis of improvement in recent generations of language models); scaling the model + data + compute; exploring the use of inference-time strategies to improve prediction (such as self-consistency [60], prompt ensembling [1, 38], or in-context example selection [43]); and introducing extra information during both training and inference, such as contextual information or samples from different, but related, tables.

On the *extensions* side, we hope that our work opens avenues toward deeper understanding of tabular foundation models including: understanding potential biases or unwanted behavior with respect to sensitive features (features such as race, age, and gender are common in tabular datasets); using tabular foundation models to address small-sample problems which might be aided by a high-quality pretrained model (such as in the Fragile Families Challenge [44]); and extending this approach to new tasks beyond prediction, such as data generation, explanation, data wrangling, and more.

## 7 Accessing Open-Source Code, Data, and Model Weights

**TabLib Preprocessing Code:** A Python module for filtering TabLib, `tabliblib`, along with scripts and configurations used to perform the filtering, are available at `https://github.com/mlfoundations/tabliblib`.

**Model Training and Inference Code:** We provide `rtfm`, a Python module used to train TABULA-8B, perform inference and evaluation, and process data, at `https://github.com/mlfoundations/rtfm`.

**T4 Dataset:** The T4 dataset is available via public credentialized access on Hugging Face datasets at `https://huggingface.co/datasets/mlfoundations/t4-full`. Because the dataset is derived from TabLib, users must first obtain permission to access TabLib at `https://huggingface.co/datasets/approximatelabs/tablib-v1-full`.

**Evaluation Datasets:** The full evaluation suite used to evaluate TABULA-8B is available via Hugging Face Datasets at `https://huggingface.co/datasets/mlfoundations/tabula-8b-eval-suite`. Each dataset includes: a CSV file containing the raw data; a TableShift [16] FeatureList JSON object; and a YAML file with associated metadata.

**Model Weights:** TABULA-8B weights are available via Hugging Face at `https://huggingface.co/mlfoundations/tabula-8b`.

## Acknowledgments and Disclosure of Funding

JG was supported by a Microsoft Grant for Customer Experience Innovation. This work was also in part supported by the NSF AI Institute for Foundations of Machine Learning (IFML, CCF-2019844), Google, Open Philanthropy, and the Allen Institute for AI. JCP was supported by the Harvard Center for Research on Computation and Society.

We are grateful to Foundry[2] for providing the compute infrastructure used to train TABULA-8B. Our research also utilized computational resources and services provided by the Hyak computing cluster at the University of Washington, and from Stability AI. The authors also gratefully acknowledge the Gauss Centre for Supercomputing[3] for funding this project by providing computing time on the GCS Supercomputer JUWELS[28] at Julich Supercomputing Centre (JSC). We are particularly appreciative of support from Jenia Jitsev at JSC.

We also acknowledge Approximate Labs[4] and express our appreciation for their development and release of TabLib, along with their communication and support as we utilized the dataset.

We are grateful to Jeffrey Li, Mike Merrill, Jonathan Hayase, and Nilesh Tripuraneni for feedback on a early versions of this paper. We also benefited greatly from advice from Matt Jordan, Alex Fang, and Jeffrey Li on large-scale data preprocessing.

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

# A T4 Data Filtering Details

As discussed previously, in order to generate T4, we filter TabLib [13] at the table, column and row level. Having filtered down the tables, we then programatically select a target column for prediction according to another set of heuristics. The remaining columns are used as features, but not as prediction targets. We select only a single target column for each table.

## A.1 Table, Column, and Row Filtering Rules

The following tables describe the entire set of heuristics we use as filters. A precise implementation may be found as a part of our software release associated with this paper. Our pipeline involves a language identification step; for language detection, we make use of the `fasttext` library [27].[5]

### List of Table Filtering Rules

| Level | Name | Description | Motivation / Hypothesis |
|-------|------|-------------|-------------------------|
| Table | English Filtering | Drop where a language ID model score is below a fixed threshold | All downstream benchmark datasets are in English |
| Table | Schema Heterogeneity | Drop tables where every cell is of the same type | Encourages understanding of mixed data types |
| Table | Row Count | Drop table with fewer than $k$ rows | Anecdotally, many "very small" tables in TabLib are general web-text tables not useful/suitable for ML. |
| Table | Column Count | Drop tables with fewer than $k$ columns *after column filters are applied* | Exclude tables that lack a reasonable amount of features |
| Table | Parse Error | Drop tables where the headers suggest there was a parsing error. | These tables are likely the result of bad table detection, and they almost definitely contain low-quality headers. |
| Table | Drop PII | Drop table where $> x\%$ of the cells match a regex for phone number or email | Don't want to train on PII for privacy reasons. Also not likely to be present in downstream tasks. |
| Table | Drop Code | Drop table with any cell that has probability $> p$ of containing code. | Lots of the data in TabLib is from GitHub and other technical documentation. Code is common. Code also confuses the model a lot, apparently due to special characters and whitespace. This code can be unevenly broken/spread across cells due to the tablib parser. |
| Table | Too many unnamed columns | Drop table if the fraction of "Unnamed: " columns is greater than a threshold. | Discard low-quality data; unnamed columns tend to be of significantly lower quality based on manual data inspection. |

---

[5]https://pypi.org/project/fasttext-langdetect/

## List of Row and Column Filtering Rules

| Level | Name | Description | Motivation / Hypothesis |
|---|---|---|---|
| Column | Drop Free-Text | Drop columns with long headers ($>$ 256 characters) | The TabLib process used to scrape the tables can result in tables with "headers" that are actually just rows of data. One indicator of this is very long headers (e.g. a text column that ends up as a header). |
| Column | Drop Numeric | Drop columns with names that are numeric. | TabLib's parsing removed tables with all-numeric headers "for most file formats". This means that $(a)$ some formats were missed, and $(b)$ tables with many numeric headers and even one non-numeric header were still included. |
| Column | Drop Missing | Drop any column with $>$ $x\%$ values that are None, NaN, whitespace or empty string values | Columns that are mostly missing will waste compute processing headers; empty cells usually won't be informative (although sometimes a header alone can be useful). |
| Column | English Filtering | Drop any text columns where average probability of English over rows is less than $p$ | Some tables contain English headers and non-English data. All of our downstream data is English. It's hard to assess quality of non-English data. |
| Column | Drop Constant | Drop columns where all values are the same. | Constant features are not useful for prediction |
| Row | Drop Missing | Drop any row with too many values that are None, NaN, whitespace or empty string values | Rows with mostly missing data are likely to be uninformative. |
| Row | Drop Duplicates | Drop duplicate rows | This is non-standard in downstream tasks |
| Row | Drop PII (regex-based) | Drop any row where PII is detected (phone number, email) | Tables with small numbers of rows containing PII can still pass through the table-level PII filtering. |
| Row | Drop Code (regex-based) | Drop any row where code is detected. | Tables with small numbers of rows containing code can still pass through the table-level code filtering |
| Row | Drop $\lfloor$ | Drop any row where any of the values contain $\lfloor$ symbol | This is exclusively used as an indicator of hierarchy (again, common in technical documentation such as that found on Github). This is a sign that the row of the table isn't self-contained and therefore probably not a candidate for a meaningful prediction task. |

## List of Target Column Selection Rules

| Name | Description | Motivation / Hypothesis |
|------|-------------|-------------------------|
| Drop All Unique | Drop non-numeric columns if all values all distinct | This is probably not a classification target (most likely a unique identifier, a date, a number, etc.) |
| Drop "Unnamed:" | Drop any column whose name starts with "Unnamed:" | "Unnamed:" is a special prefix used for unnamed columns in an Arrow table; we avoid making predictions on columns where there is no clear semantic information about the prediction target. |
| Drop dates | Drop any column with any date or time data type | Not useful as classification targets in most cases |
| Drop Too Long | Drop any column which, when serialized, is greater than 256 characters | This helps avoid choosing free-text columns as targets. |
| Drop dates | Drop any column with any date or time data type | Not useful as classification targets in most cases |

### A.2 Target Column Selection

Our procedure for target column selection is based on the rules described in the above tables. Given a table, we consider all columns to be potentially valid targets unless they do not satisfy one of the listed criteria.

Once target candidates are identified for a table, we choose a single candidate at random and use this as the prediction target. When there are both continuous and categorical candidates present, we choose a categorical candidate with probability $p = 0.9$ and a continuous candidate with probability $1 - p$. This decision reflects our qualitative observation that our selection method tends to produce higher-quality categorical columns than numeric columns, and has the effect of showing the model classification tasks more often than (binned) regression tasks during training.

In the case where we select a continuous target, we discretize it into a discrete set by selecting a number of quantiles uniformly at random over the interval $[3, 8]$, and then discretizing the target value into these columns. We serialize the resulting quantiles as "less than 1," "between 1 and 2.5", "greater than 2.5" etc.

Even after filtering, the tables in our pool contain columns which may not be meaningful prediction *targets* (for example, timestamps or UIDs). For a given table, we propose and apply a series of heuristics for identifying suitable candidates for target columns. These heuristics include excluding columns if: the name is numeric; only one unique value; has unique values for every row (excluding numeric columns); any row has a value longer than 256 characters; the column is of a date or timestamp type. We provide a detailed list of the exact target selection rules in Section A.2, and an implementation in our code. We do *not* drop such columns from the table; columns not meeting these criteria are simply kept as predictors but will never be used as prediction targets.

Once target candidates are identified for a table, we choose a single candidate at random and use this as the prediction target. When there are both continuous and categorical candidates present, we choose a categorical candidate with probability $p = 0.9$ and a continuous candidate with probability $1 - p$. This decision reflects our qualitative observation that our selection method tends to produce higher-quality categorical columns than numeric columns, and has the effect of showing the model classification tasks more often than (binned) regression tasks during training.

### A.3 Descriptive Statistics

This section provides some basic descriptive statistics of the final dataset, shown in Figures 5 and 6. Figure 5a shows that T4 represents a wide variety of data types across its tables, but that data are primarily represented as float, int, and object (string/categorical) data types. Figure 5b shows

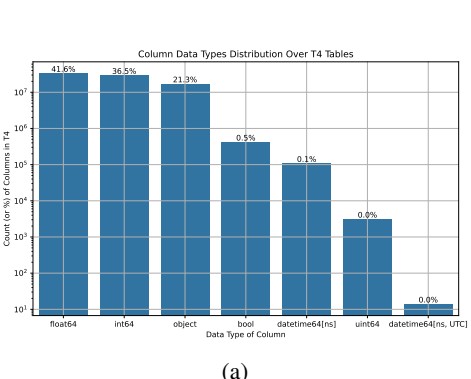

(a)

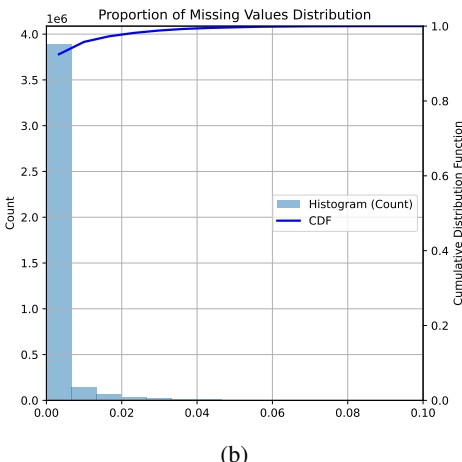

(b)

Figure 5: Summary metrics for the T4 dataset.

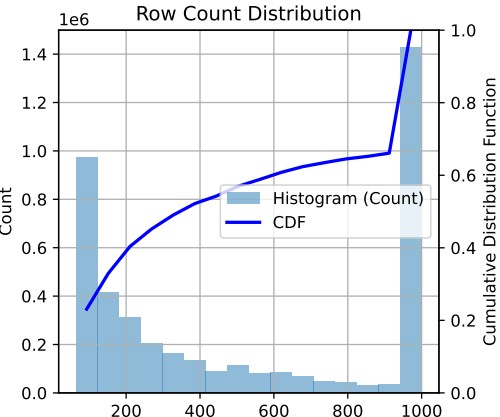

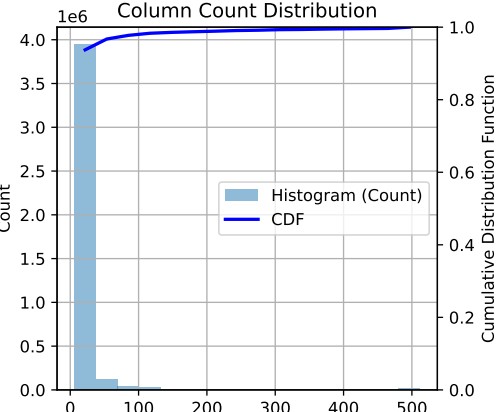

Figure 6: Distribution of counts of rows per table and columns per table in T4.

that, while most tables in T4 have little to no missing data, tables can have as much as 10% of data missing (a maximum threshold enforced by heuristics described above).

Figure 6 provides a sense of the "shape" of tables in T4, showing that roughly 30% of tables have 64 columns, and roughly 30% have $1,000$ columns, the minimum and maximum number of rows set by our heuristic filters. In contrast, nearly all tables have fewer than 50 columns, with only a very small fraction ($< 0.01\%$) having 500 or more columns (in practice, rows from tables with 500 columns would almost never fit inside the context window of T4 after serialization and tokenization).

## A.4 Implementation Details

Our data processing implementation uses Ray Datasets[6] to process tables in parallel. Our pipeline utilizes TabLib's existing `content_hash` feature to read only a set of unique tables; this avoids performing an expensive deduplication step during online processing. The shards of TabLib are processed in chunks to avoid extra overhead due to very large collections in Ray. Our pipeline includes certain additional optimizations: for example, in TabLib, data is stored as serialized Arrow bytes (and the deserialized bytes cannot easily be passed between stages of the Ray pipeline); as a result, our pipeline avoids repeatedly deserializing these bytes and only does so at a single filtering step, and at the write step.

---

[6] https://docs.ray.io/en/latest/data/overview.html

Our preprocessing implementation is provided as a separate library, `tabliblib`, along with the release of this paper and in the supplementary material.

## B  Training Details

TABULA-8B is trained for $40$ thousand steps with a global batch size of 24. We use a peak learning rate of $1e-5$ warmed up over 10% of the total training to the peak learning rate, and then use a cosine decay schedule to zero. We do not use weight decay. We found that our model was quite sensitive to the selection of the learning rate, and that evaluation loss during training correlated strongly with performance on downstream tasks.

## C  Description of Compute Resources Used

Our final training run for TABULA-8B took approximately 6 days on a single node of 8 NVIDIA 80GB A100 GPUs on a commercial cloud provider. For our TabLib filtering and XGboost experiments, we used an academic CPU cluster. Our evaluations were distributed across two academic GPU clusters consisting of NVIDIA 40GB A40 GPUs and NVIDIA A100 GPUs. As a rule of thumb, evaluating the model on a single dataset over a grid of 8 values for $k$ (number of shots) consumes around 4 GPU-hours. We estimate that the total number of GPU hours used across training, evaluation, and development is approximately $5,000 - 10,000$.

## D  Evaluation Details

In this section we provide more detail on the benchmarks datasets in our evaluation suite. We do not preprocess the datasets (no normalization, one-hot encoding, etc), as many datasets have been filtered for specific properties, and some of the downstream methods (e.g. TabPFN) perform best when preprocessing is not applied [25].

### D.1  Evaluation Datasets

#### D.1.1  UniPredict Benchmark

We use the "supervised" subset of 169 datasets from the recently-introduced UniPredict [59] benchmark, obtained through correspondence with the authors. These are high-quality tabular datasets with generally informative column names and a mix of both categorical and continuous targets, drawn directly from Kaggle. The datasets are prostprocessed in [59] using a commercial LLM; however, we do not have access to the results of this postprocessing and instead use the datasets exactly as they are obtained from the Kaggle API. Note that while the model introduced in [59] was trained and tested on separate splits of these datasets, we only use them for testing.

We obtain the complete set of tasks, and the corresponding target columns for each task, for UniPredict, via correspondence with the authors of UniPredict. However, we found that several tasks in the benchmark were incorrectly labeled as categorical when, in fact, these tasks were continuous (this is likely due to the use of an LLM to determine the target columns and their attributes in [59]). We manually verify the correct target type (continuous vs. categorical) for each of the 169 datasets, and apply corrections to TODO datasets in the benchmark. The exact set of benchmarks, target columns, and target type (continuous vs. non-continuous) used in our paper are provided in the supplementary material.

We note that our results are *not* directly comparable to the original results in UniPredict. This is for at least two reasons: (1) due to the modifications described above, where there are errors present in the original categorization of the targets (continuous vs. non-continuous) and (2) because some of the Kaggle datasets in UniPredict are continuously updated (e.g. stock datasets) and the data or access dates are not reported in [59]. We do provide our full evaluation suite, including all UniPredict datasets, in the supplementary material in order for future comparisons to our work.

### D.1.2 Grinsztajn Benchmark

The Grinsztajn benchmark [20] is a curated suite of 45 datasets consisting of numeric and categorical features. This dataset is notable in that the original study by Grinsztajn et al. found that gradient boosted decision trees (GBDTs) consistently outperformed deep learning-based methods on these tasks. The benchmark is comprised of a mix of classification and regression tasks; for all regression tasks, we apply the discretization method used in UniPredict [59] and discretize the targets into quartiles.

### D.1.3 AutoML Multimodal Benchmark

The AutoML Multimodal Benchmark [49][7] is a suite of tables which include one or more free-text fields (such as an Airbnb description, or a product review). The benchmark is considered challenging for tree-based methods due to the non-standard text-based features. However, it also poses a challenge for LLMs since some columns can contain highly variable lengths of text.

### D.1.4 OpenML CC-18 Benchmark

The OpenML Curated Classification Benchmark [2] was created by applying filtering rules to extract a high-quality subset from the OpenML platform. The rules include: no artificial data sets, no subsets of larger data sets nor binarizations of other data sets, no data sets which are perfectly predictable by using a single feature or a simple decision tree.

### D.1.5 OpenML CTR-23 Benchmark

The OpenML Curated Tabular Regression (CTR) Benchmark [14] is a curated set of tables for regression drawn from OpenML. The curation process is similar to that of OpenML-CC18, for regression tasks. As in [59], we convert the continuous regression targets into a finite set of discrete labels.

We remove the `solar_flare` task from the benchmark, as 82% of the observations have the same regression target value (0) and thus we cannot apply our quartile-transformation method.

## D.2 Baselines

For the supervised learning baselines (XGBoost, TabPFN, CatBoost, Logistic Regression), we conduct 10 independent trials, drawing separate training sets of size $k$ shots, for each value of $k$. We use the full remaining dataset as the test set.

**Hyperparameter tuning:** For each of the 10 independent trials, we tune the hyperparameters of the model. For XGBoost, we conduct 10 iterations of hyperparameter tuning using the HyperOpt hyperparameter optimization library and the hyperparameter grid defined in [16]. For TabPFN and L2-regularized Logistic Regression, we conduct a full grid search (since there is only a single hyperparameter).

### D.2.1 Llama 3 Base Model

We use the pretrained Llama 3 model available on Hugging Face. For this model, we do not modify the tokenizer (i.e. by adding special tokens that are used by TABULA-8B), but the serialized data format is identical to the format seen by our model during training (Figure 2b).

### D.2.2 XGBoost

For every dataset and number of shots, we tune the hyperparameters according to the grid from [16]. We use 10 iterations of the adaptive hyperparameter tuning method HyperOpt on this grid with 3-fold cross-validation, whenever the number of samples is greater than or equal to 3; when the number of samples is less than 3, we use the default settings.

---

[7] https://github.com/sxjscience/automl_multimodal_benchmark/tree/main

### D.2.3 TabPFN

TabPFN [25] is a hypernetwork that predicts the parameters of a neural network that can be used to classify the data. TabPFN is a pretrained Transformer model that takes a training dataset as input, and produces the parameters of a network as output; that network is then used to classify the test data. TabPFN is widely considered to be among the state-of-the-art methods for prediction on tabular data [25, 33], and has been shown to be especially effective for few-shot learning.

We use the official TabPFN implementation[8]. TabPFN has one tunable hyperparameter, the number of model predictions that are ensembled with feature and class rotations (`N_ensemble_configurations` in the TabPFN codebase). We sweep over all values in the range $[3, 2 \cdot d]$ where $d$ is the number of features. As noted in the package documentation, when `N_ensemble_configurations` $> 2 \cdot d$ for a binary classification task, no further averaging is applied.

The official implementation of TabPFN has three limitations relevant to our experiments. First, TabPFN is limited to 100 features.[9] As a result, when the number of features is greater than 100, we use TabPFN's feature subsampling, which randomly selects 100 features. Second, TabPFN cannot be trained on datasets that have more than 10 input classes. We do not report the results of experiments where TabPFN cannot be trained on at least one of the 10 random iterates of each value of $k$. Third, TabPFN cannot be trained on fewer than $|C|$ examples, where $C$ is the set of potential classes.

### D.3 Relative Sample Efficiency

For two classifiers $f, g$, let $N_D(f, \alpha)$ and $N_D(g, \alpha)$ denote the number of samples required for the classifiers to reach a performance level $\alpha$ on data $D$. The *relative efficiency* of $f$ relative to $g$ on dataset $D$ at level $\alpha$ is equal to

$$N_D(f, \alpha)/N_D(g, \alpha). \tag{1}$$

### D.4 Generation Procedure

We use the default generation settings of the Llama 3 Hugging Face model.[10] This includes: temperature of 0.6, top-$p$ 0.9. We do not tune these generation settings.

## E  Detailed Results

### E.1 Results Beyond 32 Shots

In this section, we provide additional results for larger numbers of shots not provided in the main text. Figure 7 provides extended results for both the baseline models, and for TABULA-8B.

As shown in Figure 7, all models tend to improve with more shots. However, on the subset of datasets that can be used for 64- and 128-shot learning, we observe a narrower gap between TABULA-8B and baselines. We hypothesize that this is due to the fact that there is a selection bias: only datasets with small numbers of features and short column headers are candidates for 128-shot learning (due to the limited context window size of TABULA-8B). As a result, this biases those evaluations away from semantically-rich datasets where we expect TABULA-8B to excel.

### E.2 Additional Baseline Comparisons

In this section, we provide comparisons to additional baselines not included in the main text. These include supervised baselines (Logistic Regression, CatBoost), and commercial LLMs (variants of Claude[11]).

---

[8]https://github.com/automl/TabPFN
[9]https://github.com/automl/TabPFN/blob/main/tabpfn/scripts/transformer_prediction_interface.py#L105
[10]https://huggingface.co/meta-llama/Meta-Llama-3-8B
[11]https://www.anthropic.com/claude

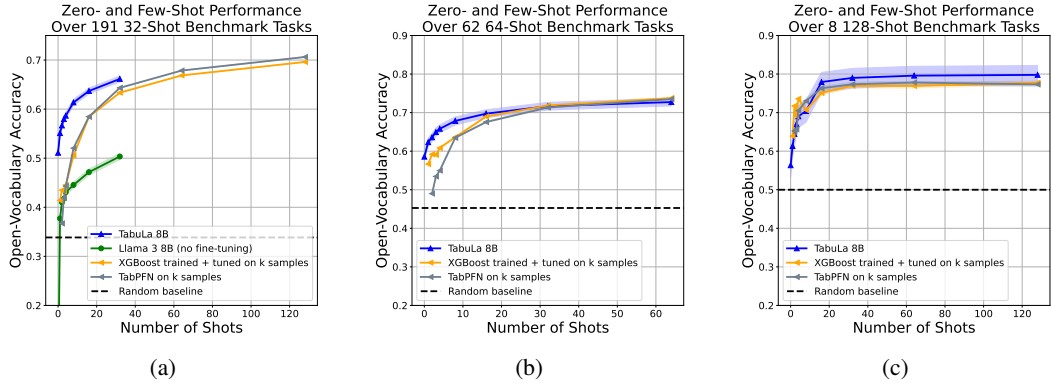

(a)           (b)           (c)

Figure 7: 7a: Results for 32-shot tasks, with extended curves for baselines. 7b, 7c: Results for 64- and 128-shot tasks. All models continue to improve as $k$ increases, but the gap between methods may narrow. However, the *nature* of the datasets that can be used for 64-shot learning with TABULA-8B are considerably different – fewer features, with shorter, less semantically-meaningful column names – which may downwardly bias the observed performance of TABULA-8B with large $k$.

For the Logistic Regression and Catboost baselines, we follow the same hyperparameter tuning and evaluation procedure described in the main text. These results, along with our main results, are presented in Figure 8.

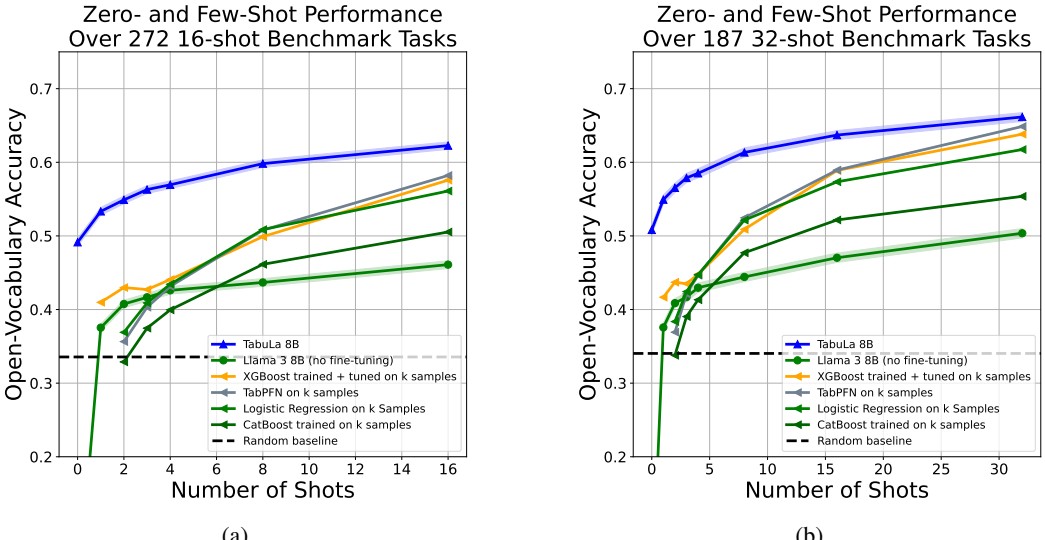

(a)           (b)

Figure 8: Few-shot curves for TABULA-8B along with additional supervised baselines (Logistic Regression, CatBoost).

We conduct additional comparisons to strong commercial LLMs. In particular, we compare TABULA-8B to two variances of Anthropic's Claude models: (1) Claude Instant, a fast, relatively small model likely to be on the same order of magnitude of compute as TABULA-8B (the exact parameter counts of all Claude models are not publicly disclosed); and (2) Claude 3 Sonnet, a highly performant instruction-tuned LLM likely to be larger than TABULA-8B both in terms of parameter count and total training compute. Due to the cost of accessing these commercial models, we conduct this evaluation on a subset of our evaluation suite.

Figure 9 shows that TABULA-8B significantly outperforms both models. Figure 9 shows two particularly interesting results: first, both Claude models show little to no improvement with increasing number of shots. This is also consistent with the behavior of the base Llama 3 model. We hypothesize that this behavior demonstrates the value of explicit training for few-shot learning, and likely

highlights the gap betweenn more generalized, task-agnostic training of these models relative to the task-specific training of TABULA-8B. Second, the commercial models perform *worse* than any other baseline, on average, beyond 3 shots (below which most baselines perform similarly). We hypothesize that this is due, at least in part, to the instruction-following training of the Claude models; no other set of models in our comparison undergo this second post-training phase likely to be utilized in the Claude training pipeline. This instruction-following may improve models alignment with user intentions but could decrease their ability to explicitly learn from data.

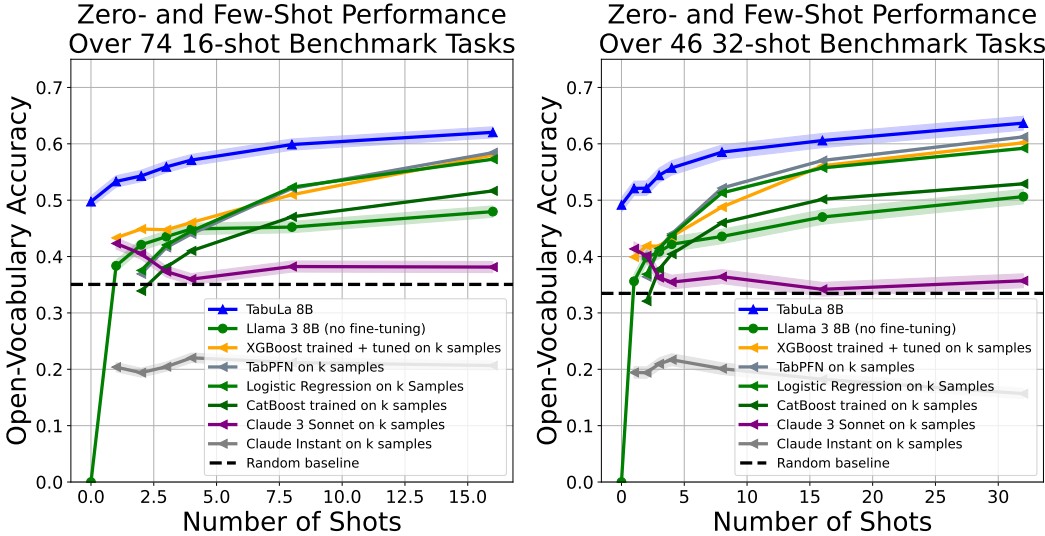

Figure 9: Comparison of few-shot performance with commercial models, Claude Instant and Claude 3 Sonnet, along with other baselines.

### E.3 Performance on Numeric Tasks

One concern with tabular LLMs is their perceived inability to represent numeric data: since language models represent all data as tokens in a learned embedding space, it may be more challenging for language models to learn the complex relationships between numeric features which are required for many classification tasks.

In order to provide evidence of TABULA-8B's performance on numeric data, in this section we report two different slices of our main results. First, we present results on the subset of our evaluation tasks that contain *at least one* numeric column (int or float data type). This excludes tables that are strictly non-numeric. Second, we present results on *only* the subset of our evaluation tasks that contain entirely numeric data. We note that both of these subsets *exclude* tables with purely textual data – precisely the datasets where we might expect language models to perform strongly.

The results on these two subsets of our evaluation suite are shown in Figure 10.

Figure 10a shows that, on the evaluation subset where all tables contain at least one numeric column, TABULA-8B still outperforms baselines across all numbers of shots. Figure 10a demonstrates that the mere presence of numeric features does not erode the performance of TABULA-8B relative to existing SOTA baseline methods.

Figure 10b shows that, on the evaluation subset where all tables contain *only* numeric columns, TABULA-8B performs on par with existing SOTA baselines but generally only matches their performance. This result is perhaps unsurprising, as numeric-only data is the most advantageous setting for GBDT models and TabPFN (as GBDTs can directly learn splits over numeric values, and TabPFN is exclusively trained on numeric features). However, the ability of TABULA-8B to match these strong baselines, while exceeding them on non-numeric data and possessing capabilities no other baseline possesses (such as zero-shot prediction and transfer), is an indication of its strength and potential utility as a general tabular classifier.

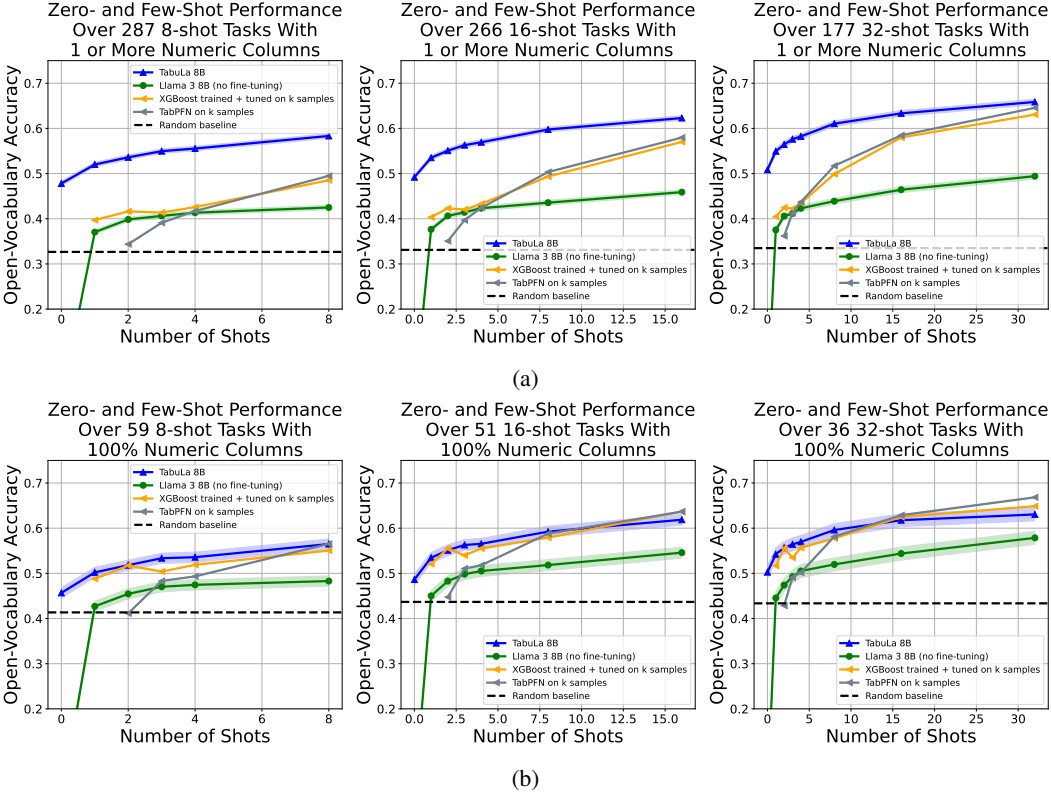

(a)

(b)

Figure 10: Few-shot curves for TABULA-8B on the subset of evaluation tasks that contain at least one numeric column (10a) and on the subset of evaluation tasks that contain *entirely* numeric columns (10b).

# F  Robustness and Ablation Studies

We conduct a series of additional ablation studies to investigate robustness to column ordering, robustness to feature dropout, robustness to header removal, the impact of our causal masking procedure, and the impact of our data filtering procedure. We note that other aspects of our pipeline, such as the target selection procedure and the individual parameters of several of our TabLib processing heuristics, also affect the quality of our resulting model, but a comprehensive evaluation of these individual decisions is left to future work. We describe each ablation study below.

### F.1  Ablation Study: Row-Causal Table Masking (RCTM)

In this section we conduct an ablation study of the row-causal table masking (RCTM) procedure used to train our model. To summarize the masking procedure (also described in Section 3 and Figure 2a): we explicitly allow the model to attend across samples within the same table in a batch. We hypothesize that this structure will encourage few-shot learning and will mitigate catastrophic forgetting which could cause the base model's few-shot capabilities to deteriorate during fine-tuning.

To do this, we design a controlled experiment. Both arms of the experiment use 10% of the compute of TABULA-8B (trained for $4k$ steps) but are otherwise identical. In one run, we remove the RCTM strategy described in Figure 2a, replacing it with a per-sample causal attention mask (the model is not allowed to attend to any samples besides the target sample, regardless of which table they are derived from). We evaluate both models over the full test suite (all benchmark tasks).

The results of this study are shown in Figure 11. Our proposed masking scheme improves the models' ability to attend across samples, while removing this masking causes the model not to learn from additional shots (for $k \le 8$) and to deteriorate as the number of shots grows. This figure

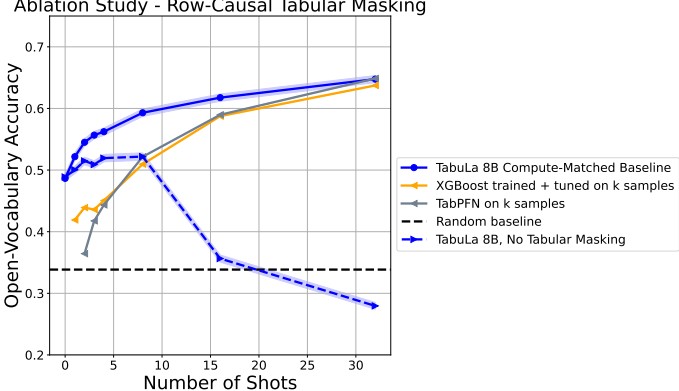

Figure 11: Results of an ablation study comparing a model trained without our novel row-causal tabular masking (RCTM) scheme (described in Section 3 and illustrated in Figure 2a) vs. a baseline compute-matched version of TABULA-8B. RCTM improves the models' ability to attend across samples, while removing RCTM causes the model not to learn from additional shots (for $k \leq 8$) and to *deteriorate* as the number of shots grows. This result demonstrates the potential loss of few-shot capabilities during fine-tuning if it is not explicitly encouraged by the fine-tuning task.

also demonstrates the potential loss of few-shot capabilities during fine-tuning if it is not explicitly encouraged by the fine-tuning task – but also that these capabilities can be maintained or improved over the base model if they are a part of the fine-tuning task.

## F.2 Robustness Evaluation: Informative Header Removal

A potential disadvantage of a language modeling approach to tabular data prediction is that language models may be particularly reliant on semantically-informative column "headers" (column names, the keys in the tabular key-value structure), in contrast to traditional supervised learning methods which do not utilize the headers at all. This risk has been noted in previous work; for example, UniPredict [59] suggests an approach that uses a commercial LLM to "rewrite" the headers of a table to make them more informative. In this work, however, we seek to avoid expensive preprocessing and use only the provided headers from our training data.

To understand this sensitivity to the semantic quality of the headers, we conduct a controlled experiment to test the effect of removing informative column headers from a dataset. Specifically, we do the following: starting from a benchmark with high-quality headers (UniPredict), we *replace* the original headers with "X1", "X2", ... and the target with "Y". Then, we evaluate TABULA-8B on the data. We do not alter the data itself; only the feature names are replaced.

The results of this study are shown in Figure 12. We highlight a few key findings from these results. First, for small number of shots, the semantically-meaningful headers provide a performance benefit: for instance, in the 16-shot subset of Figure 12, semantically-meaningful headers provide a consistent accuracy gain of 3-5pp. Second, TABULA-8B can still outperform supervised baselines even without these headers: for example, Figure 12 shows that TABULA-8B still outperforms all baselines on the benchmark even without semantically-meaningful headers. This finding is further supported by our results on the OpenML benchmarks (CC18, CTR23) in Figure 4; these datasets also tend to have uninformative headers. Third, we observe that the utility of semantically-meaningful headers decreases as the number of shots increase. For example, at 32 shots, the performance with and without the original headers is effectively identical. We hypothesize that, as the number of shots grows, the model is increasingly utilizing the *values* provided in the shots (and their distribution) and is less reliant on the *keys* for providing information about the task.

Collectively, the results of this ablation study suggest that TABULA-8B is robust to the semantic content of the headers, and that TABULA-8B is capable of providing effective tabular data predictions even in the absence of rich column headers.

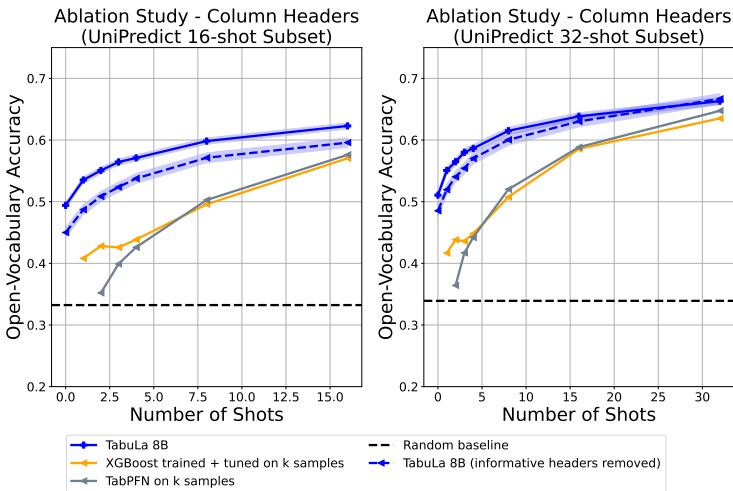

Figure 12: Results of column header ablation study described in Section F.2. For low numbers of shots, there tends to be a positive effect from informative headers. However, as the number of shots increases, the utility of the headers decreases.

## F.3 Robustness Evaluation: Feature Dropout

A potential risk of using language models for tabular data prediction may be their brittleness: language models, particularly in the few-shot setting, are known to be sensitive to details which should be irrelevant to the task difficulty, including order of examples [3, 32], whitespace [47], and prompt formatting [26, 45, 47]. Here, we are particularly interested in probing robustness to the *removal* of features. Some supervised models, including XGBoost, can be trained in a way that allows them to handle missing features at inference time, at a cost of slightly decreased predictive accuracy due to the loss of information. However, whether language models possess similar characteristics is unknown.

We design an experiment to test how TABULA-8B's zero- and few-shot performance degrades as features are removed from the evaluation datasets. First, on the training split of each dataset, we train a single XGBoost model using the same hyperparameter tuning procedure used for our baselines, and we extract the feature importance for all features in the dataset according to this model. Next, we evaluate TABULA-8B on each dataset, removing the top $k$ features for $k \in [1, 5]$. These features are removed in either descending or ascending order of importance ("important first" and "important last", respectively). For comparison, we also train and evaluate XGBoost models. For these XGBoost models, we set $1/k$ fraction of the data to missing, uniformly at random. Then we train a hyperparameter-tuned model on this data and evaluate it on clean test data where the top-$k$ features are set to missing (no other data is set to missing in the test data); this allows us to naturally leverage XGBoost's robustness to missing data.

This experiment is conducted on the same random sample of 32 datasets described in Section F.4. The results of this study are shown in Figure 13. Across 0-,4-,16-,and 32-shot evaluations, TABULA-8B shows a similar or favorable rate of decline in performance, relative to XGBoost, as the number of removed features increases (as indicated by the similar slope of the lines). We note that the XGBoost models have better absolute performance because these are *full-shot* models trained on the full training split; we use these models to compare the rate of decrease in accuracy as dropout increases, not to compare the accuracy itself. The results in Figure 13 also provide further evidence of when TABULA-8B may be favorable to standard supervised learning methods: namely, when the amount of missing data at test time is large. Finally, Figure 13 suggests that TABULA-8B is *not* brittle with respect to the features present in our evaluation datasets; removing these features causes only the expected drop in performance (and removing *unimportant* features is even associated with an increase in performance relative to the full feature set when the number of shots is larger than 8).

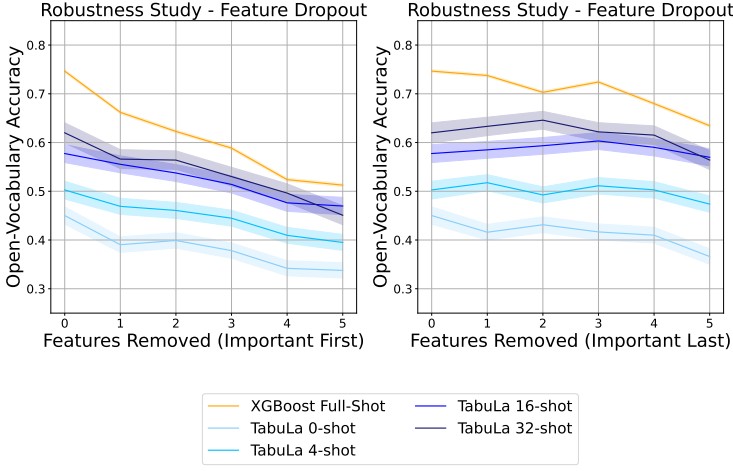

Figure 13: Feature dropout ablation study results. We compare the few-shot performance of TABULA-8B to the performance of an XGBoost model trained on the full training split of each dataset, progressively removing features at evaluation time. Features are removed in order of decreasing ("important first") or increasing ("important last") variable importance (see Section F.3 for details). TABULA-8B's performance decreases at a rate consistent with XGBoost.

## F.4 Robustness Evaluation: Column Order Invariance

Recent work has suggested that invariance to column ordering is an important property of tabular foundation models [58]. In this subsection, we conduct a controlled experiment to assess the sensitivity of our model to the *ordering* of the columns in the dataset.

To assess this, we conduct the following experiment. We first choose a random subset of 32 tasks from our evaluation suite (we exclude the tasks from AMLB due to their relatively irregular structure as discussed above since our goal is to compare performance on relatively standard tabular data). For each task, we evaluate the model on a *permuted* version of the original data: that is, we randomly permute the columns, but otherwise leave the data unchanged. Due to computational constraints, we conduct this evaluation only at a coarse grid of $(0, 8, 16, 32)$ shots. We compare the accuracy on these tasks to the accuracy on the original data. We use the same TABULA-8B model for both the permuted and non-permuted evaluations.

The results of this study are shown in Figure 14. We observe a small drop (with Clopper-Pearson intervals overlapping at all points) after permuting the columns, but the general shape and rate of increase of the model under both cases is quite similar. We hypothesize that this drop is due to the fact that many tabular datasets, including those from our evaluation benchmark (which are drawn from manually-curated sources such as Kaggle, OpenML, and UCI Machine Learning Repository), have manually-selected column *orderings* that slightly improve prediction performance. This might include, for example, a "date" preceding the rest of the columns, or a "high" and "low" column located near each other in a stock dataset. These feature relationships, we hypothesize, can make it easier for models to pool information between related features and may contribute to the small drop observed on permuting the columns. We note that this sensitivity to order has been observed for language models in other contexts [32, 34].

Our results broadly show that TABULA-8B maintains consistent performance above baselines even under feature permutation. However, they also suggest that the sensitivity to prompting and formatting that affects LLMs in other contexts [32, 34] could possibly affect tabular LLMs. We believe further research into this issue is necessary, and may indeed point toward future, more effective methods for leveraging feature ordering to improve model performance.

## F.5 Ablation Study: Data Filtering

We conduct a controlled experiment to assess the impact of our filtering strategies. In particular, we compare a dataset filtered according to the strategies described in Section 4 to a "minimally-filtered"

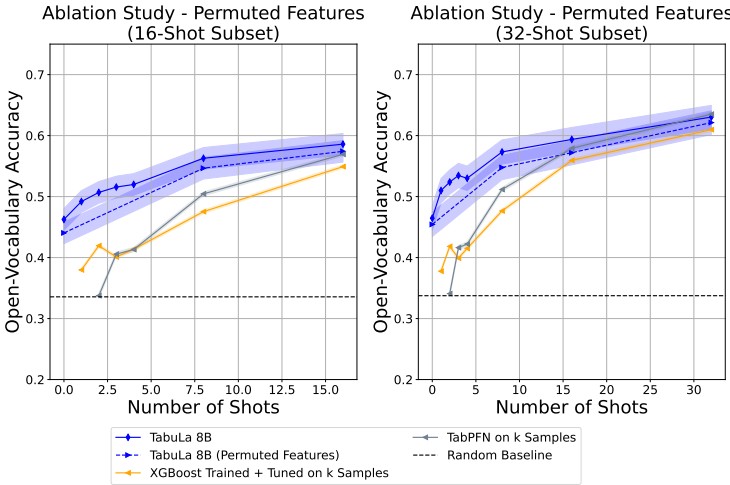

Figure 14: Results of column permutation study described in F.4. We randomly permute the columns for a randomly-selected subset of 32 tasks, and evaluate TABULA-8B on the permuted data. We hypothesize that the slight drop in model performance is due to manually-crafted and semantically meaningful feature orderings in the data. However, our results broadly show that TABULA-8B maintains consistent performance above baselines even under feature permutation.

TabLib dataset. We use a "minimally-filtered" dataset rather than an *unfiltered* dataset because applying no filtering could potentially result in a small number of very large tables dominating training. For the minimally-filtered baseline, we only apply the max-row count filter, max column filter, and max header length filter; the latter two help ensure that the resulting serializations are not too long so that the model is still able to perform few-shot training.

The results of this study are shown in Figure 15. While our filtering strategy has a positive impact at larger numbers of shots ($k \geq 16$), there is no impact evident at $k < 16$, with the minimally-filtered baseline performing similarly. We hypothesize that this is a reflection of the relatively limited additional filtering performed by the rest of our pipeline relative to the "minimal" baseline (which consists mainly of language filtering, PII and code filtering, and heuristics to remove excessive amounts of missing data). Additionally, given the clear impact of improvements in data quality for language model pretraining (e.g. [54]), we hypothesize that further refinements of our filtering pipeline would be likely to achieve further gains over minimal filtering. Finally, we emphasize that some of our filtering strategies (in particular, the PII detection) were designed to improve the *safety* of the model, not the quality, and we believe that some form of safety filtering should still be used regardless of its downstream effects.

## F.6   Ablation Study: Base Language Model

In this section, we evaluate the improvements of TABULA-8B due to improvements in the underlying base language model. In order to do so, we train variants of TABULA-8B which are identical to the final version, except we use Llama 1 and Llama 2 as the base language models. This allows us to investigate the improvements in TABULA-8B as the underlying language model improves. We compare the Llama 1 and Llama 2 variants to a compute-matched Llama 3 variant (we use only 10% of the compute relative to our final model, as in our other ablation studies, but also compare to the 100%-compute TABULA-8B for reference). We note that, due to the smaller context sizes of the Llama 1 and Llama 2 models, we ensure the comparisons are *example-matched*; that is, we train Llama 2 (context size 4096) for 2x the update steps relative to Llama 3 (context size 8192), and train Llama 1 (context size 2048) for 4x the update steps. This ensures that all models see roughly the same number of tokens or examples during fine-tuning.

The results of this study are shown in Figure 16. We only report the results up to 8 shots to allow for fair comparisons across all models, as Llama 1's context size of 2048 is 4x smaller than TABULA-8B and can only fit up to 8 shots for many tasks. Figure 16 shows the clear improvement from better base models: as the underlying base models improve (Llama 1 $\rightarrow$ Llama 2 $\rightarrow$ Llama 3), the fine-tuned

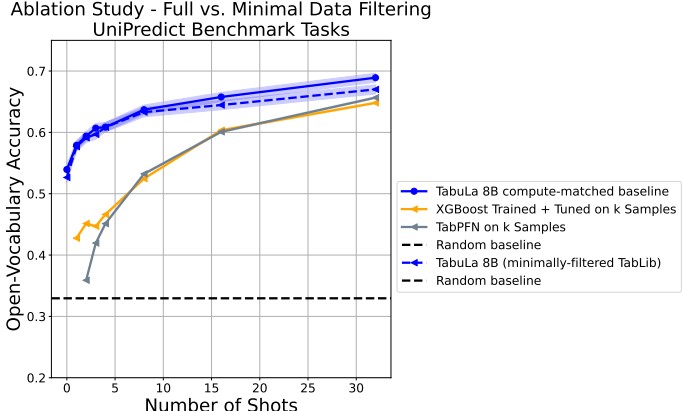

Figure 15: Compute-matched comparison of TABULA-8B with our full filtering vs. an identical model trained on minimally-filtered TabLib (both models are trained for 10% of the number of steps as the final model, with identical hyperparameters).

model also improves. These results provide hope that further improvements in language modeling could also lead to gains in tabular models.

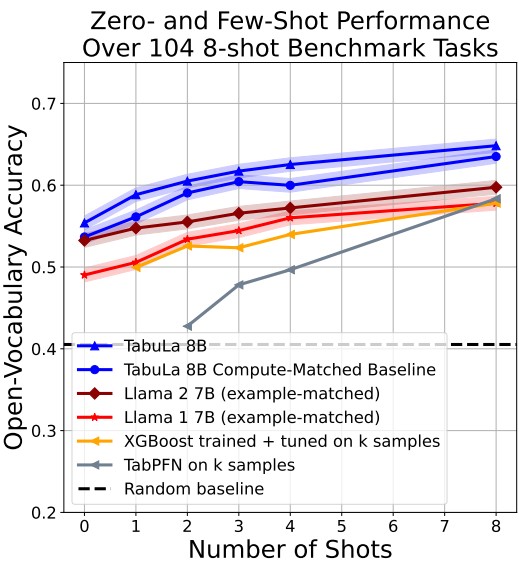

Figure 16: Results of base model ablation study.

## G   Benchmark Contamination Study

### G.1   Identifying Potential Contamination

Identifying contamination in tabular data is complex. As mentioned above, tabular data is invariant to permutations of rows and columns; as a result, the "same" tabular dataset could appear in a number of different permutations of its respective rows or columns. As a result, so-called "exact" deduplication methods are likely to be imperfect for tabular data (we also perform exact deduplication in our filtering step, so at most each individual dataset should only appear once in T4).

When assessing the impact of duplication on our downstream evaluations, we are particularly concerned about the possibility of a model overfitting to the *overall features and distribution* of a dataset, not individual points in that distirbution – a model could learn unwanted information about a

| Benchmark | Potentially Leaked Tables Count (% of Benchmark Tables) | |
| --- | --- | --- |
| | Fuzzy Deduplication | Strict Deduplication |
| OpenML-CC18 | 23 (31.9%) | 13 (18.1%) |
| OpenML-CTR23 | 12 (34.2%) | 6 (17.1%) |
| AMLB | 1 (12.5%) | 0 |
| UniPredict | 50 (29.5%) | 39 (23%) |
| Grinsztajn | 16 (35.5%) | 4 (8.9%) |

Table 1: Counts of potentially contaminated tables according to "fuzzy" and "strict" procedures described. Note that "fuzzy" decontamination is stricter and is more likely to generate both true and false positives when checking for contamination.

test set, for example, by observing points strictly from the training set of an i.i.d. split. As a result, we focus on eliminating datasets which have the same *schema* as a proxy for a dataset; we do not search for individual *data points* within that schema.

We propose two levels of searching for contamination; we refer to these methods as "fuzzy" and "strict". In *fuzzy* deduplication, we search for whether *every column name* in an evaluation dataset is present in the training dataset. In *strict* deduplication, we further add the condition that the number of columns must be identical. Note that we do not compute a matching directly between the columns of the two datasets, as checking for membership is much more efficient than checking for a compatible mapping between columns and our datasets can be up to 512 columns. Note that *fuzzy deduplication will potentially exclude more datasets* at the risk of potentially more false positives, since strict deduplication only adds conditions to the fuzzy deduplication. Fuzzy is therefore a *more conservative* deduplication mechanism than strict.

We search over all 1.55M tables in T4 and apply both "fuzzy" and "strict" checks. Some descriptive metrics for this search are shown in Table 1.

It is perhaps expected that, in most cases, benchmark tables pass our T4 filtering procedure and make it into T4 (one notable exception is AMLB, where most tables likely fail to pass our rules which filter for cells with large numbers of characters). Indeed, these are public benchmarks designed for learning on tabular data, and TabLib includes a significant component of datasets sourced from GitHub; many users likely work with these datasets and have uploaded them to GitHub.

## G.2 Impact of Contamination

In this section we investigate the impact of potential contamination in our training data. In particular, we investigate whether the number of repetitions of a dataset in the training data is correlated with the downstream performance on that task, and we assess performance on "potentially contaminated" vs. "decontaminated" tasks.

Figure 18 shows the relationship between the number of potential contamination instances for each dataset, and the TABULA-8B accuracy on that dataset. We find no clear relationship between this potential contamination and downstream performance, and believe that this reflects, at least in part, the conservative nature of our contamination test, which is likely to have may false positives for datasets with generic column names that may occur frequently in TabLib (for example, columns "Date", "Open", "Close" are common among stock datasets; columns "v1" "v2" are common generic variable names).

Additionally, we believe that other considerations likely explain the lack of correlation between (potential) contamination in the training data and downstream benchmark berformance. For example:

- TABULA-8B does not train on the full corpus; datasets that appear a small number of times may in fact not be seen during training.
- Prior work has suggested mixed impact of contamination [e.g. 6, 40]; contamination does not always improve performance and can sometimes reduce performance.
- We are only fine-tuning the model which can be thought of as an alignment step. It is possible that contamination in *pretraining* affects this more and that fine-tuning is less susceptible to memorization.

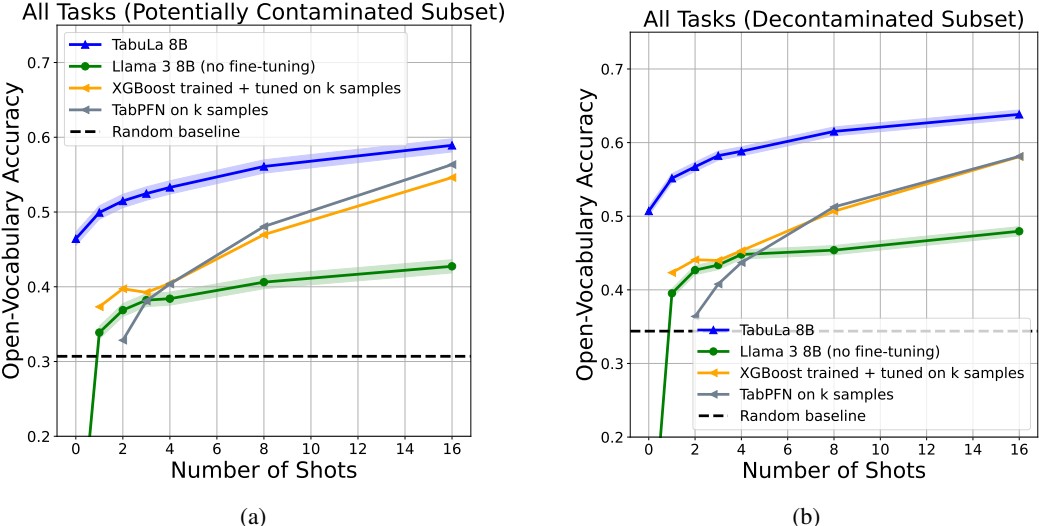

(a)                                              (b)

Figure 17: 17a: Results curves on the subset of tasks identified as potentially contaminated according to our fuzzy decontamination procedure. 17b: Results curves on the subset of tasks not identified as potentially contaminated according to our fuzzy decontamination procedure.

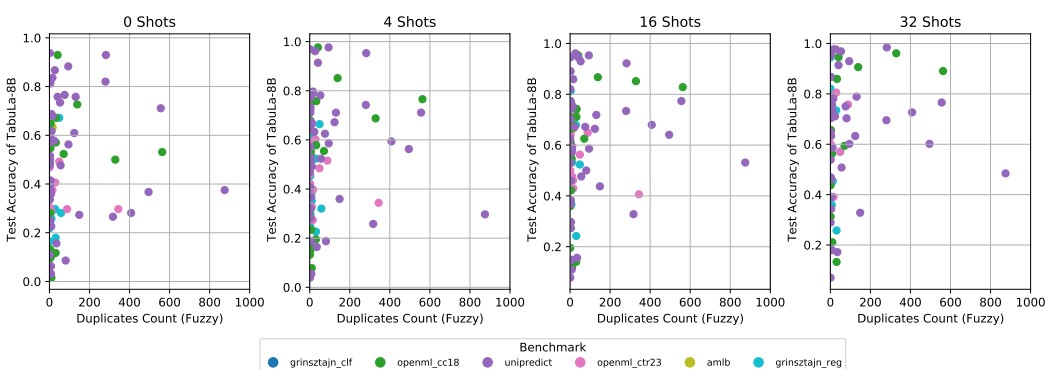

Figure 18: Contamination rates vs. accuracy across varying numbers of shots. We find no clear relationship between contamination in the training set and downstream task performance.

- Due to our use of deduplication, if a dataset does recur, it is not identical – so the model never really sees an identical table more than once unless we make multiple passes over the training set or sample with replacement.

- Due to our use of row-level deduplication and random shuffling, the exact evaluation datasets in the same order are unlikely to ever be seen by the model even if provided during training; this may be a guard against memorization.

We also separately report our results on the "possibly contaminated" vs. "decontaminated" subset of our evaluation suite, in Figure 17. Figure 17 shows that our model (and all baseline models) tend to perform *better* on the decontaminated subset. We hypothesize that this reflects a few factors. In particular, datasets in the "decontaminated" subset are likely to have unique column names. This "uniqueness" likely correlates positively with semantic quality; our model will tend to perform better on such datasets. Furthermore, we hypothesize that this reflects the strictness of our fuzzy decontamination check: it is likely that many of the tables flagged as "potentially contaminated" are false positives (our manual inspection confirmed this in many cases, although it is difficult to verify that two shuffled tabular datasets are identical; we leave such an analysis to future work).

# H List of Evaluation Datasets and Per-Task Results

We provide results for TABULA-8B on each individual task, along with the accuracy of a random-class predictor (which is equivalent to 1 / number of classes) in this section. For the complete results for all values of shots and all baselines, see the supplementary material. Note that 'NA' results for TABULA-8B indicate that the serialized data with a given value of shots $k$ exceeds the model's context window size.

| Task | TabuLa-8B | | | XGBoost | | Random |
| | 0 | 4 | 32 | 4 | 32 | |
|---|---|---|---|---|---|---|
| amlb/data_scientist_salary | 0.328 | 0.398 | 0.391 | 0.202 | 0.283 | 0.167 |
| amlb/imdb_genre_prediction | 0.844 | 0.742 | 0.828 | 0.528 | 0.554 | 0.500 |
| amlb/jigsaw_unintended_bias100K | 0.938 | 0.922 | NA | 0.942 | NA | 0.500 |
| amlb/kick_starter_funding | 0.594 | 0.680 | 0.609 | 0.644 | 0.649 | 0.500 |
| amlb/melbourne_airbnb | 0.727 | NA | NA | NA | NA | 0.100 |
| amlb/news_channel | 0.297 | 0.227 | NA | 0.213 | NA | 0.167 |
| amlb/product_sentiment_machine_hack | 0.406 | 0.438 | 0.531 | 0.542 | 0.762 | 0.250 |
| amlb/wine_reviews | 0.648 | 0.711 | NA | 0.080 | NA | 0.033 |
| grinsztajn/cat_clf/albert | 0.445 | 0.500 | NA | 0.499 | NA | 0.500 |
| grinsztajn/cat_clf/compas-two-years | 0.547 | 0.539 | 0.602 | 0.499 | 0.582 | 0.500 |
| grinsztajn/cat_clf/covertype | 0.422 | 0.570 | NA | 0.500 | NA | 0.500 |
| grinsztajn/cat_clf/default-of-credit-card-clients | 0.508 | 0.531 | 0.539 | 0.500 | 0.618 | 0.500 |
| grinsztajn/cat_clf/electricity | 0.461 | 0.641 | 0.664 | 0.503 | 0.644 | 0.500 |
| grinsztajn/cat_clf/eye_movements | 0.508 | 0.430 | NA | 0.504 | NA | 0.500 |
| grinsztajn/cat_clf/road-safety | 0.508 | 0.477 | NA | 0.502 | NA | 0.500 |
| grinsztajn/cat_reg/Bike_Sharing_Demand | 0.352 | 0.344 | 0.508 | 0.250 | 0.424 | 0.250 |
| grinsztajn/cat_reg/Brazilian_houses | 0.344 | 0.547 | 0.797 | 0.256 | 0.769 | 0.250 |
| grinsztajn/cat_reg/Mercedes_Benz_Greener_Manufa... | 0.266 | NA | NA | NA | NA | 0.250 |
| grinsztajn/cat_reg/OnlineNewsPopularity | 0.289 | 0.273 | NA | 0.253 | NA | 0.250 |
| grinsztajn/cat_reg/SGEMM_GPU_kernel_performance | 0.094 | 0.602 | 0.898 | 0.249 | 0.891 | 0.250 |
| grinsztajn/cat_reg/analcatdata_supreme | 0.461 | 0.453 | 0.914 | 0.624 | 0.980 | 0.333 |
| grinsztajn/cat_reg/black_friday | 0.203 | 0.234 | 0.297 | 0.250 | 0.331 | 0.250 |
| grinsztajn/cat_reg/diamonds | 0.344 | 0.547 | 0.688 | 0.248 | 0.736 | 0.250 |
| grinsztajn/cat_reg/house_sales | 0.258 | 0.453 | NA | 0.249 | NA | 0.250 |
| grinsztajn/cat_reg/nyc-taxi-green-dec-2016 | 0.242 | 0.445 | NA | 0.249 | NA | 0.250 |
| grinsztajn/cat_reg/particulate-matter-ukair-2017 | 0.305 | 0.375 | NA | 0.248 | NA | 0.250 |
| grinsztajn/cat_reg/visualizing_soil | 0.281 | 0.383 | 0.688 | 0.248 | 0.789 | 0.250 |
| grinsztajn/cat_reg/yprop_4_1 | 0.203 | 0.219 | NA | 0.251 | NA | 0.250 |
| grinsztajn/num_clf/Diabetes130US | 0.531 | 0.508 | 0.484 | 0.500 | 0.523 | 0.500 |
| grinsztajn/num_clf/MagicTelescope | 0.555 | 0.703 | 0.695 | 0.494 | 0.670 | 0.500 |
| grinsztajn/num_clf/MiniBooNE | 0.461 | 0.539 | NA | 0.505 | NA | 0.500 |
| grinsztajn/num_clf/bank-marketing | 0.500 | 0.578 | 0.656 | 0.502 | 0.695 | 0.500 |
| grinsztajn/num_clf/california | 0.453 | 0.578 | 0.742 | 0.500 | 0.720 | 0.500 |
| grinsztajn/num_clf/covertype | 0.500 | 0.484 | 0.477 | 0.500 | 0.553 | 0.500 |
| grinsztajn/num_clf/credit | 0.562 | 0.578 | 0.656 | 0.505 | 0.659 | 0.500 |
| grinsztajn/num_clf/default-of-credit-card-clients | 0.516 | 0.477 | 0.523 | 0.500 | 0.601 | 0.500 |
| grinsztajn/num_clf/electricity | 0.547 | 0.609 | 0.602 | 0.503 | 0.655 | 0.500 |
| grinsztajn/num_clf/eye_movements | 0.469 | 0.547 | NA | 0.504 | NA | 0.500 |
| grinsztajn/num_clf/heloc | 0.500 | 0.508 | NA | 0.494 | NA | 0.500 |
| grinsztajn/num_clf/house_16H | 0.477 | 0.609 | NA | 0.503 | NA | 0.500 |
| grinsztajn/num_clf/jannis | 0.500 | 0.594 | NA | 0.500 | NA | 0.500 |
| grinsztajn/num_clf/pol | 0.523 | 0.484 | 0.773 | 0.512 | 0.810 | 0.500 |
| grinsztajn/num_reg/Ailerons | 0.266 | 0.289 | NA | 0.278 | NA | 0.250 |
| grinsztajn/num_reg/Bike_Sharing_Demand | 0.234 | 0.414 | 0.398 | 0.250 | 0.424 | 0.250 |
| grinsztajn/num_reg/Brazilian_houses | 0.250 | 0.570 | 0.812 | 0.256 | 0.769 | 0.250 |
| grinsztajn/num_reg/MiamiHousing2016 | 0.250 | 0.352 | NA | 0.250 | NA | 0.250 |
| grinsztajn/num_reg/california | 0.289 | 0.406 | NA | 0.247 | NA | 0.250 |
| grinsztajn/num_reg/cpu_act | 0.359 | 0.320 | NA | 0.272 | NA | 0.250 |

| | | | | | | |
|---|---|---|---|---|---|---|
| grinsztajn/num_reg/diamonds | 0.391 | 0.602 | 0.688 | 0.248 | 0.742 | 0.250 |
| grinsztajn/num_reg/elevators | 0.188 | 0.266 | NA | 0.266 | NA | 0.250 |
| grinsztajn/num_reg/fifa | 0.250 | 0.406 | 0.469 | 0.248 | 0.434 | 0.250 |
| grinsztajn/num_reg/house_16H | 0.336 | 0.320 | NA | 0.249 | NA | 0.250 |
| grinsztajn/num_reg/house_sales | 0.328 | 0.492 | NA | 0.249 | NA | 0.250 |
| grinsztajn/num_reg/houses | 0.258 | 0.312 | 0.445 | 0.247 | 0.418 | 0.250 |
| grinsztajn/num_reg/isolet | 0.262 | NA | NA | NA | NA | 0.250 |
| grinsztajn/num_reg/medical_charges | 0.328 | 0.555 | 0.727 | 0.249 | 0.801 | 0.250 |
| grinsztajn/num_reg/nyc-taxi-green-dec-2016 | 0.273 | 0.406 | NA | 0.249 | NA | 0.250 |
| grinsztajn/num_reg/pol | 0.602 | 0.617 | 0.867 | 0.683 | 0.815 | 0.500 |
| grinsztajn/num_reg/sulfur | 0.258 | 0.227 | 0.383 | 0.253 | 0.441 | 0.250 |
| grinsztajn/num_reg/superconduct | 0.242 | 0.438 | NA | 0.247 | NA | 0.250 |
| grinsztajn/num_reg/wine_quality | 0.578 | 0.508 | 0.516 | 0.538 | 0.622 | 0.333 |
| grinsztajn/num_reg/year | 0.227 | 0.289 | NA | 0.262 | NA | 0.250 |
| openml_cc18/Fashion-MNIST | 0.070 | NA | NA | NA | NA | 0.100 |
| openml_cc18/GesturePhaseSegmentationProcessed | 0.266 | 0.312 | NA | 0.270 | NA | 0.200 |
| openml_cc18/MiceProtein | 0.125 | 0.211 | NA | 0.118 | NA | 0.125 |
| openml_cc18/PhishingWebsites | 0.516 | 0.656 | NA | 0.500 | NA | 0.500 |
| openml_cc18/adult | 0.727 | 0.812 | 0.828 | 0.763 | 0.764 | 0.500 |
| openml_cc18/analcatdata_authorship | 0.234 | 0.438 | NA | 0.365 | NA | 0.250 |
| openml_cc18/analcatdata_dmft | 0.125 | 0.156 | 0.125 | 0.182 | 0.188 | 0.167 |
| openml_cc18/bank-marketing | 0.508 | 0.758 | 0.844 | 0.889 | 0.880 | 0.500 |
| openml_cc18/banknote-authentication | 0.477 | 0.680 | 0.922 | 0.509 | 0.862 | 0.500 |
| openml_cc18/blood-transfusion-service-center | 0.516 | 0.664 | 0.586 | 0.787 | 0.749 | 0.500 |
| openml_cc18/breast-w | 0.938 | 0.961 | 0.984 | 0.569 | 0.900 | 0.500 |
| openml_cc18/car | 0.789 | 0.898 | 0.898 | 0.763 | 0.755 | 0.250 |
| openml_cc18/churn | 0.703 | 0.758 | 0.914 | 0.870 | 0.863 | 0.500 |
| openml_cc18/cmc | 0.320 | 0.383 | 0.352 | 0.330 | 0.450 | 0.333 |
| openml_cc18/cnae-9 | 0.172 | NA | NA | NA | NA | 0.111 |
| openml_cc18/connect-4 | 0.297 | 0.414 | NA | 0.522 | NA | 0.333 |
| openml_cc18/credit-approval | 0.484 | 0.555 | 0.742 | 0.428 | 0.764 | 0.500 |
| openml_cc18/credit-g | 0.508 | 0.523 | 0.703 | 0.500 | 0.725 | 0.500 |
| openml_cc18/diabetes | 0.617 | 0.656 | 0.680 | 0.661 | 0.722 | 0.500 |
| openml_cc18/dna | 0.312 | 0.430 | NA | 0.467 | NA | 0.333 |
| openml_cc18/electricity | 0.438 | 0.633 | 0.711 | 0.544 | 0.695 | 0.500 |
| openml_cc18/eucalyptus | 0.273 | 0.352 | 0.359 | 0.211 | 0.407 | 0.200 |
| openml_cc18/first-order-theorem-proving | 0.211 | 0.195 | NA | 0.284 | NA | 0.167 |
| openml_cc18/har | 0.125 | NA | NA | NA | NA | 0.167 |
| openml_cc18/isolet | 0.022 | NA | NA | NA | NA | 0.038 |
| openml_cc18/jm1 | 0.812 | 0.750 | 0.805 | 0.746 | 0.784 | 0.500 |
| openml_cc18/jungle_chess_2pcs_raw_endgame_complete | 0.477 | 0.555 | 0.516 | 0.409 | 0.589 | 0.333 |
| openml_cc18/kc1 | 0.852 | 0.836 | 0.820 | 0.775 | 0.843 | 0.500 |
| openml_cc18/kr-vs-kp | 0.469 | 0.508 | NA | 0.490 | NA | 0.500 |
| openml_cc18/letter | 0.039 | 0.070 | 0.266 | 0.038 | 0.200 | 0.038 |
| openml_cc18/madelon | 0.531 | NA | NA | NA | NA | 0.500 |
| openml_cc18/mfeat-factors | 0.078 | 0.172 | NA | 0.092 | NA | 0.100 |
| openml_cc18/mfeat-fourier | 0.086 | 0.172 | NA | 0.092 | NA | 0.100 |
| openml_cc18/mfeat-karhunen | 0.078 | 0.141 | NA | 0.092 | NA | 0.100 |
| openml_cc18/mfeat-morphological | 0.141 | 0.188 | 0.555 | 0.092 | 0.493 | 0.100 |
| openml_cc18/mfeat-pixel | 0.109 | NA | NA | NA | NA | 0.100 |
| openml_cc18/mfeat-zernike | 0.141 | 0.141 | NA | 0.092 | NA | 0.100 |
| openml_cc18/mnist_784 | 0.148 | NA | NA | NA | NA | 0.100 |
| openml_cc18/nomao | 0.398 | 0.742 | NA | 0.542 | NA | 0.500 |
| openml_cc18/numerai28.6 | 0.539 | 0.469 | NA | 0.495 | NA | 0.500 |
| openml_cc18/optdigits | 0.070 | 0.094 | NA | 0.092 | NA | 0.100 |
| openml_cc18/ozone-level-8hr | 0.445 | 0.852 | NA | 0.969 | NA | 0.500 |
| openml_cc18/pc1 | 0.766 | 0.945 | 0.898 | 0.964 | 0.964 | 0.500 |
| openml_cc18/pc3 | 0.820 | 0.875 | NA | 0.834 | NA | 0.500 |
| openml_cc18/pc4 | 0.609 | 0.914 | NA | 0.890 | NA | 0.500 |

| | | | | | | |
|---|---|---|---|---|---|---|
| openml_cc18/pendigits | 0.039 | 0.242 | 0.523 | 0.099 | 0.534 | 0.100 |
| openml_cc18/phoneme | 0.453 | 0.641 | 0.734 | 0.624 | 0.723 | 0.500 |
| openml_cc18/qsar-biodeg | 0.508 | 0.523 | NA | 0.579 | NA | 0.500 |
| openml_cc18/satimage | 0.180 | 0.320 | NA | 0.186 | NA | 0.167 |
| openml_cc18/segment | 0.195 | 0.438 | NA | 0.144 | NA | 0.143 |
| openml_cc18/semeion | 0.148 | NA | NA | NA | NA | 0.100 |
| openml_cc18/sick | 0.938 | 0.914 | NA | 0.950 | NA | 0.500 |
| openml_cc18/spambase | 0.555 | 0.680 | NA | 0.560 | NA | 0.500 |
| openml_cc18/splice | 0.336 | 0.328 | NA | 0.378 | NA | 0.333 |
| openml_cc18/steel-plates-fault | 0.141 | 0.273 | NA | 0.201 | NA | 0.143 |
| openml_cc18/texture | 0.070 | 0.180 | NA | 0.090 | NA | 0.091 |
| openml_cc18/tic-tac-toe | 0.383 | 0.609 | 0.523 | 0.454 | 0.702 | 0.500 |
| openml_cc18/vehicle | 0.219 | 0.258 | 0.484 | 0.239 | 0.495 | 0.250 |
| openml_cc18/vowel | 0.055 | 0.125 | 0.172 | 0.106 | 0.358 | 0.091 |
| openml_cc18/wall-robot-navigation | 0.250 | 0.305 | NA | 0.398 | NA | 0.250 |
| openml_cc18/wilt | 0.594 | 0.812 | 0.969 | 0.959 | 0.957 | 0.500 |
| openml_ctr23/Moneyball | 0.477 | 0.375 | 0.609 | 0.243 | 0.595 | 0.250 |
| openml_ctr23/QSAR_fish_toxicity | 0.266 | 0.305 | 0.484 | 0.245 | 0.462 | 0.250 |
| openml_ctr23/abalone | 0.273 | 0.430 | 0.406 | 0.296 | 0.483 | 0.250 |
| openml_ctr23/airfoil_self_noise | 0.258 | 0.305 | 0.383 | 0.242 | 0.379 | 0.250 |
| openml_ctr23/auction_verification | 0.266 | 0.344 | 0.398 | 0.252 | 0.652 | 0.250 |
| openml_ctr23/brazilian_houses | 0.438 | 0.578 | 0.633 | 0.256 | 0.551 | 0.250 |
| openml_ctr23/california_housing | 0.320 | 0.375 | 0.461 | 0.247 | 0.419 | 0.250 |
| openml_ctr23/cars | 0.359 | 0.336 | 0.656 | 0.228 | 0.637 | 0.250 |
| openml_ctr23/concrete_compressive_strength | 0.328 | 0.484 | 0.562 | 0.246 | 0.446 | 0.250 |
| openml_ctr23/cps88wages | 0.391 | 0.367 | 0.422 | 0.255 | 0.332 | 0.250 |
| openml_ctr23/cpu_activity | 0.250 | 0.375 | NA | 0.272 | NA | 0.250 |
| openml_ctr23/diamonds | 0.438 | 0.680 | 0.742 | 0.247 | 0.751 | 0.250 |
| openml_ctr23/energy_efficiency | 0.391 | 0.461 | 0.719 | 0.238 | 0.749 | 0.250 |
| openml_ctr23/fifa | 0.352 | 0.359 | NA | 0.243 | NA | 0.250 |
| openml_ctr23/fps_benchmark | 0.242 | 0.367 | NA | 0.250 | NA | 0.250 |
| openml_ctr23/geographical_origin_of_music | 0.281 | NA | NA | NA | NA | 0.250 |
| openml_ctr23/grid_stability | 0.242 | 0.297 | NA | 0.239 | NA | 0.250 |
| openml_ctr23/health_insurance | 0.406 | 0.500 | 0.633 | 0.482 | 0.591 | 0.333 |
| openml_ctr23/kin8nm | 0.312 | 0.242 | 0.297 | 0.251 | 0.342 | 0.250 |
| openml_ctr23/kings_county | 0.305 | 0.422 | NA | 0.249 | NA | 0.250 |
| openml_ctr23/miami_housing | 0.273 | 0.391 | NA | 0.250 | NA | 0.250 |
| openml_ctr23/naval_propulsion_plant | 0.211 | 0.227 | NA | 0.249 | NA | 0.250 |
| openml_ctr23/physiochemical_protein | 0.242 | 0.289 | 0.281 | 0.250 | 0.334 | 0.250 |
| openml_ctr23/pumadyn32nh | 0.219 | 0.273 | NA | 0.251 | NA | 0.250 |
| openml_ctr23/red_wine | 0.523 | 0.508 | 0.656 | 0.476 | 0.619 | 0.333 |
| openml_ctr23/sarcos | 0.227 | 0.203 | NA | 0.254 | NA | 0.250 |
| openml_ctr23/socmob | 0.391 | 0.523 | 0.695 | 0.248 | 0.598 | 0.250 |
| openml_ctr23/space_ga | 0.203 | 0.219 | 0.305 | 0.261 | 0.412 | 0.250 |
| openml_ctr23/student_performance_por | 0.375 | 0.258 | NA | 0.274 | NA | 0.250 |
| openml_ctr23/superconductivity | 0.250 | 0.352 | NA | 0.247 | NA | 0.250 |
| openml_ctr23/video_transcoding | 0.250 | 0.336 | NA | 0.249 | NA | 0.250 |
| openml_ctr23/wave_energy | 0.234 | 0.273 | NA | 0.256 | NA | 0.250 |
| openml_ctr23/white_wine | 0.484 | 0.594 | 0.594 | 0.651 | 0.660 | 0.333 |
| unipredict/aakashjoshi123/exercise-and-fitness-... | 0.383 | 0.641 | 0.766 | 0.253 | 0.761 | 0.250 |
| unipredict/aakashjoshi123/spotify-top-hits-data | 0.555 | 0.797 | 0.781 | 0.750 | 0.694 | 0.077 |
| unipredict/abcsds/pokemon | 0.977 | 0.984 | 0.992 | 0.060 | 0.128 | 0.059 |
| unipredict/adityakadiwal/water-potability | 0.562 | 0.469 | 0.516 | 0.524 | 0.591 | 0.500 |
| unipredict/agirlcoding/all-space-missions-from-... | 0.891 | 0.875 | 0.867 | 0.910 | 0.874 | 0.333 |
| unipredict/ahsan81/food-ordering-and-delivery-a... | 0.273 | 0.266 | 0.242 | 0.296 | 0.323 | 0.250 |
| unipredict/ahsan81/superstore-marketing-campaig... | 0.516 | 0.727 | 0.766 | 0.848 | 0.836 | 0.500 |
| unipredict/akshaydattatraykhare/diabetes-dataset | 0.586 | 0.742 | 0.672 | 0.661 | 0.704 | 0.500 |
| unipredict/alexisbcook/pakistan-intellectual-ca... | 0.609 | 0.656 | NA | 0.592 | NA | 0.083 |
| unipredict/alirezachahardoli/bank-personal-loan-1 | 0.758 | 0.773 | 0.914 | 0.920 | 0.913 | 0.500 |

| | | | | | | |
|---|---|---|---|---|---|---|
| unipredict/altruistdelhite04/gold-price-data | 0.562 | 0.680 | 0.867 | 0.241 | 0.603 | 0.250 |
| unipredict/amirhosseinmirzaie/countries-life-ex... | 0.766 | 0.719 | NA | 0.244 | NA | 0.250 |
| unipredict/amirhosseinmirzaie/pistachio-types-d... | 0.547 | 0.570 | NA | 0.523 | NA | 0.500 |
| unipredict/ananthr1/weather-prediction | 0.703 | 0.828 | 0.812 | 0.369 | 0.787 | 0.200 |
| unipredict/andrewmvd/fetal-health-classification | 0.211 | 0.734 | NA | 0.736 | NA | 0.333 |
| unipredict/andrewmvd/udemy-courses | 1.000 | 1.000 | 0.992 | 0.311 | 0.433 | 0.250 |
| unipredict/arashnic/time-series-forecasting-wit... | 0.992 | 1.000 | 0.984 | 0.251 | 0.903 | 0.250 |
| unipredict/arnabchaki/data-science-salaries-2023 | 0.898 | 0.961 | 0.969 | 0.256 | 0.898 | 0.250 |
| unipredict/arnabchaki/indian-restaurants-2023 | 0.281 | 0.289 | 0.242 | 0.257 | 0.308 | 0.250 |
| unipredict/arnavsmayan/netflix-userbase-dataset | 0.344 | 0.305 | 0.352 | 0.362 | 0.438 | 0.333 |
| unipredict/arnavsmayan/vehicle-manufacturing-da... | 0.055 | 0.047 | 0.062 | 0.077 | 0.094 | 0.059 |
| unipredict/arslanr369/bitcoin-price-2014-2023 | 1.000 | 0.992 | 0.984 | 0.247 | 0.899 | 0.250 |
| unipredict/ashishkumarjayswal/diabetes-dataset | 0.516 | 0.680 | 0.742 | 0.661 | 0.712 | 0.500 |
| unipredict/ashishkumarjayswal/movies-updated-data | 0.656 | 0.555 | 0.641 | 0.158 | 0.276 | 0.091 |
| unipredict/atharvaingle/crop-recommendation-dat... | 0.055 | 0.320 | 0.695 | 0.049 | 0.438 | 0.045 |
| unipredict/awaiskaggler/insurance-csv | 0.398 | 0.547 | 0.695 | 0.245 | 0.519 | 0.250 |
| unipredict/azminetoushikwasi/-lionel-messi-all-... | 0.219 | 0.414 | 0.469 | 0.634 | 0.594 | 0.111 |
| unipredict/barun2104/telecom-churn | 0.844 | 0.836 | 0.836 | 0.717 | 0.857 | 0.500 |
| unipredict/bhanupratapbiswas/bollywood-actress-... | 0.258 | 0.430 | 0.656 | 0.567 | 0.440 | 0.125 |
| unipredict/bhanupratapbiswas/fashion-products | 0.359 | 0.367 | 0.406 | 0.328 | 0.349 | 0.333 |
| unipredict/bhanupratapbiswas/uber-data-analysis | 0.617 | 0.820 | 0.922 | 0.931 | 0.928 | 0.500 |
| unipredict/bhanupratapbiswas/world-top-billiona... | 0.383 | 0.531 | NA | 0.254 | NA | 0.250 |
| unipredict/bharath011/heart-disease-classificat... | 0.531 | 0.641 | 0.844 | 0.564 | 0.906 | 0.500 |
| unipredict/bhavkaur/hotel-guests-dataset | 0.430 | 0.594 | 0.867 | 0.865 | 0.836 | 0.333 |
| unipredict/bhavkaur/simplified-titanic-dataset | 0.555 | 0.523 | 0.734 | 0.647 | 0.735 | 0.500 |
| unipredict/blastchar/telco-customer-churn | 0.711 | 0.719 | 0.711 | 0.696 | 0.748 | 0.500 |
| unipredict/bretmathyer/telemedicine-used | 0.500 | 0.555 | NA | 0.494 | NA | 0.500 |
| unipredict/buntyshah/auto-insurance-claims-data | 0.602 | 0.594 | NA | 0.700 | NA | 0.500 |
| unipredict/carolzhangdc/imdb-5000-movie-dataset | 0.484 | 0.516 | NA | 0.247 | NA | 0.250 |
| unipredict/chirin/africa-economic-banking-and-s... | 0.953 | 0.953 | 0.977 | 0.877 | 0.969 | 0.500 |
| unipredict/christinestevens/cstevens-peloton-data | 0.984 | 0.992 | 1.000 | 0.151 | 0.164 | 0.143 |
| unipredict/cpluzshrijayan/milkquality | 0.406 | 0.406 | 0.711 | 0.361 | 0.836 | 0.333 |
| unipredict/crxxom/manhwa-dataset | 0.695 | 0.859 | NA | 0.589 | NA | 0.250 |
| unipredict/dansbecker/aer-credit-card-data | 0.609 | 0.742 | 0.930 | 0.545 | 0.967 | 0.500 |
| unipredict/deependraverma13/diabetes-healthcare... | 0.602 | 0.680 | 0.734 | 0.661 | 0.706 | 0.500 |
| unipredict/desalegngeb/german-fintech-companies | 0.758 | 0.789 | NA | 0.280 | NA | 0.250 |
| unipredict/dileep070/heart-disease-prediction-u... | 0.789 | 0.750 | 0.805 | 0.768 | 0.827 | 0.500 |
| unipredict/dsfelix/us-stores-sales | 0.547 | 0.648 | NA | 0.258 | NA | 0.250 |
| unipredict/elakiricoder/gender-classification-d... | 0.594 | 0.719 | 0.883 | 0.502 | 0.935 | 0.500 |
| unipredict/fedesoriano/stroke-prediction-dataset | 0.945 | 0.977 | 0.945 | 0.951 | 0.950 | 0.500 |
| unipredict/gabrielsantello/cars-purchase-decisi... | 0.383 | 0.609 | 0.820 | 0.580 | 0.843 | 0.500 |
| unipredict/gauravduttakiit/resume-dataset | 0.906 | 0.969 | NA | 0.043 | NA | 0.040 |
| unipredict/geomack/spotifyclassification | 0.422 | 0.641 | 0.984 | 0.536 | 0.981 | 0.500 |
| unipredict/gyanprakashkushwaha/laptop-price-pre... | 0.328 | 0.562 | 0.570 | 0.251 | 0.550 | 0.250 |
| unipredict/hansrobertson/american-companies-pro... | 0.250 | 0.297 | 0.242 | 0.250 | 0.314 | 0.250 |
| unipredict/harishkumardatalab/medical-insurance... | 0.336 | 0.586 | 0.750 | 0.235 | 0.681 | 0.250 |
| unipredict/harshitshankhdhar/imdb-dataset-of-to... | 0.430 | 0.445 | NA | 0.310 | NA | 0.250 |
| unipredict/hashemi221022/bank-loans | 0.773 | 0.844 | 0.867 | 0.920 | 0.887 | 0.500 |
| unipredict/hashemi221022/diabetes | 0.508 | 0.750 | 0.656 | 0.661 | 0.721 | 0.500 |
| unipredict/hawkingcr/airbnb-for-boston-with-fra... | 0.789 | 0.719 | 0.797 | 0.780 | 0.796 | 0.500 |
| unipredict/hemanthhari/psycological-effects-of-... | 0.414 | 0.391 | 0.570 | 0.245 | 0.615 | 0.143 |
| unipredict/hesh97/titanicdataset-traincsv | 0.797 | 0.734 | 0.703 | 0.587 | 0.706 | 0.500 |
| unipredict/iamsumat/spotify-top-2000s-mega-dataset | 0.305 | 0.406 | 0.516 | 0.258 | 0.276 | 0.250 |
| unipredict/iqmansingh/company-employee-dataset | 0.641 | 0.539 | 0.586 | 0.073 | 0.199 | 0.050 |
| unipredict/ishadss/productivity-prediction-of-g... | 0.391 | 0.430 | NA | 0.233 | NA | 0.250 |
| unipredict/jainilcoder/netflix-stock-price-pred... | 1.000 | 0.992 | 0.992 | 0.261 | 0.884 | 0.250 |
| unipredict/jillanisofttech/brain-stroke-dataset | 0.969 | 0.961 | 0.953 | 0.948 | 0.948 | 0.500 |
| unipredict/kabure/german-credit-data-with-risk | 0.594 | 0.586 | 0.695 | 0.500 | 0.720 | 0.500 |
| unipredict/kandij/diabetes-dataset | 0.562 | 0.758 | 0.711 | 0.661 | 0.717 | 0.500 |

| | | | | | |
|---|---|---|---|---|---|
| unipredict/kanths028/usa-housing | 0.195 | 0.281 | NA | 0.242 | NA | 0.250 |
| unipredict/kingabzpro/cosmetics-datasets | 0.859 | 0.836 | NA | 0.164 | NA | 0.167 |
| unipredict/kreeshrajani/human-stress-prediction | 0.547 | 0.633 | 0.664 | 0.500 | 0.496 | 0.500 |
| unipredict/kumargh/pimaindiansdiabetescsv | 0.117 | 0.172 | 0.141 | 0.156 | 0.171 | 0.077 |
| unipredict/larsen0966/student-performance-data-set | 0.297 | 0.266 | NA | 0.088 | NA | 0.077 |
| unipredict/lightonkalumba/us-womens-labor-force... | 0.891 | 0.992 | 1.000 | 0.508 | 0.986 | 0.500 |
| unipredict/mahnazarjmand/bank-personal-loan | 0.758 | 0.828 | 0.922 | 0.920 | 0.918 | 0.500 |
| unipredict/maryalebron/life-expectancy-data | 0.023 | 0.023 | NA | 0.030 | NA | 0.028 |
| unipredict/maryammanoochehry/bank-personal-loan | 0.844 | 0.891 | 0.883 | 0.920 | 0.914 | 0.500 |
| unipredict/mathchi/diabetes-data-set | 0.500 | 0.695 | 0.680 | 0.661 | 0.704 | 0.500 |
| unipredict/mayankpatel14/second-hand-used-cars-... | 0.234 | 0.203 | 0.297 | 0.226 | 0.659 | 0.250 |
| unipredict/mayurdalvi/simple-linear-regression-... | 0.453 | 0.516 | 0.594 | 0.548 | 0.543 | 0.500 |
| unipredict/mayuriawati/bangalore-chain-restaura... | 0.891 | 0.875 | NA | 0.086 | NA | 0.045 |
| unipredict/mazlumi/ielts-writing-scored-essays-... | 0.109 | 0.227 | NA | 0.122 | NA | 0.083 |
| unipredict/mfaisalqureshi/spam-email | 0.703 | 0.891 | 0.984 | 0.846 | 0.846 | 0.500 |
| unipredict/mirichoi0218/insurance | 0.250 | 0.633 | 0.719 | 0.258 | 0.708 | 0.250 |
| unipredict/nancyalaswad90/review | 0.617 | 0.641 | 0.680 | 0.661 | 0.703 | 0.500 |
| unipredict/naveenkumar20bps1137/predict-student... | 0.039 | 0.086 | NA | 0.061 | NA | 0.059 |
| unipredict/nikhil1e9/netflix-stock-price | 0.984 | 0.977 | 0.984 | 0.246 | 0.898 | 0.250 |
| unipredict/noordeen/insurance-premium-prediction | 0.461 | 0.531 | 0.734 | 0.258 | 0.677 | 0.250 |
| unipredict/oles04/bundesliga-seasons | 1.000 | 1.000 | NA | 0.529 | NA | 0.500 |
| unipredict/oles04/top-leagues-player | 0.484 | 0.641 | 0.609 | 0.264 | 0.271 | 0.250 |
| unipredict/patelprashant/employee-attrition | 0.805 | 0.820 | NA | 0.837 | NA | 0.500 |
| unipredict/pavansubhasht/ibm-hr-analytics-attri... | 0.820 | 0.883 | NA | 0.837 | NA | 0.500 |
| unipredict/phangud/spamcsv | 0.664 | 0.883 | 0.969 | 0.846 | 0.846 | 0.500 |
| unipredict/prevek18/ames-housing-dataset | 0.445 | 0.625 | NA | 0.252 | NA | 0.250 |
| unipredict/primaryobjects/voicegender | 0.445 | 0.516 | NA | 0.500 | NA | 0.500 |
| unipredict/prkhrawsthi/bitcoin-usd-daily-price-... | 0.961 | 0.969 | 0.984 | 0.248 | 0.899 | 0.250 |
| unipredict/rajyellow46/wine-quality | 0.352 | 0.453 | 0.438 | 0.372 | 0.426 | 0.143 |
| unipredict/ravibarnawal/mutual-funds-india-deta... | 0.203 | 0.227 | 0.219 | 0.228 | 0.310 | 0.167 |
| unipredict/receplyasolu/6k-weather-labeled-spot... | 0.141 | 0.172 | 0.273 | 0.124 | 0.211 | 0.125 |
| unipredict/redwankarimsony/heart-disease-data | 0.273 | 0.484 | 0.555 | 0.430 | 0.528 | 0.200 |
| unipredict/reihanenamdari/breast-cancer | 0.344 | 0.289 | 0.297 | 0.250 | 0.295 | 0.250 |
| unipredict/rishikeshkonapure/hr-analytics-predi... | 0.820 | 0.836 | NA | 0.837 | NA | 0.500 |
| unipredict/rkiattisak/student-performance-in-ma... | 0.453 | 0.477 | 0.547 | 0.250 | 0.494 | 0.250 |
| unipredict/rounakbanik/pokemon | 0.977 | 0.969 | NA | 0.963 | NA | 0.500 |
| unipredict/rpaguirre/tesla-stock-price | 0.977 | 0.992 | 0.984 | 0.247 | 0.882 | 0.250 |
| unipredict/rtatman/chocolate-bar-ratings | 0.109 | 0.141 | 0.227 | 0.138 | 0.149 | 0.100 |
| unipredict/ruchi798/student-feedback-survey-res... | 0.117 | 0.062 | 0.070 | 0.086 | 0.099 | 0.100 |
| unipredict/ruchi798/tv-shows-on-netflix-prime-v... | 0.250 | 0.312 | 0.477 | 0.283 | 0.457 | 0.167 |
| unipredict/sabasaeed1953/stock-prices-of-2023 | 0.984 | 0.969 | 0.977 | 0.274 | 0.854 | 0.250 |
| unipredict/saloni1712/chatgpt-app-reviews | 0.625 | 0.602 | 0.633 | 0.365 | 0.519 | 0.200 |
| unipredict/sanjanchaudhari/bankloan | 0.656 | 0.648 | 0.594 | 0.628 | 0.669 | 0.500 |
| unipredict/sanjanchaudhari/netflix-dataset | 0.602 | 0.500 | 0.500 | 0.155 | 0.293 | 0.100 |
| unipredict/sanjanchaudhari/user-behavior-on-ins... | 0.469 | 0.547 | 0.766 | 0.500 | 0.796 | 0.500 |
| unipredict/saunakghosh/nba-players-dataset | 0.828 | 0.719 | 0.836 | 0.472 | 0.763 | 0.125 |
| unipredict/saurabh00007/diabetescsv | 0.578 | 0.695 | 0.703 | 0.661 | 0.703 | 0.500 |
| unipredict/sbhatti/financial-sentiment-analysis | 0.594 | 0.602 | 0.680 | 0.471 | 0.527 | 0.333 |
| unipredict/shashankshukla123123/marketing-campaign | 0.266 | 0.812 | NA | 0.888 | NA | 0.500 |
| unipredict/shivamb/disney-movies-and-tv-shows | 0.883 | 0.969 | 0.984 | 0.752 | 0.899 | 0.500 |
| unipredict/shivamb/hm-stores-dataset | 0.211 | 0.547 | NA | 0.458 | NA | 0.250 |
| unipredict/shreyanshverma27/imdb-horror-chillin... | 0.398 | 0.484 | 0.500 | 0.257 | 0.301 | 0.250 |
| unipredict/shreyapurohit/anime-data | 0.266 | 0.789 | 0.906 | 0.246 | 0.889 | 0.250 |
| unipredict/shroukgomaa/babies-food-ingredients | 0.289 | 0.320 | NA | 0.274 | NA | 0.250 |
| unipredict/shubhamgupta012/titanic-dataset | 0.742 | 0.703 | 0.734 | 0.697 | 0.682 | 0.500 |
| unipredict/siddharthss/crop-recommendation-dataset | 0.102 | 0.227 | 0.625 | 0.049 | 0.438 | 0.045 |
| unipredict/sidhus/crab-age-prediction | 0.047 | 0.148 | 0.195 | 0.088 | 0.192 | 0.053 |
| unipredict/suraj520/dairy-goods-sales-dataset | 0.617 | 0.508 | NA | 0.268 | NA | 0.250 |
| unipredict/surajjha101/stores-area-and-sales-data | 0.250 | 0.234 | 0.250 | 0.253 | 0.244 | 0.250 |

| | | | | | | |
|---|---|---|---|---|---|---|
| unipredict/surajjha101/top-youtube-channels-data | 0.508 | 0.508 | 0.469 | 0.142 | 0.183 | 0.077 |
| unipredict/tahzeer/indian-startups-by-state | 0.062 | 0.141 | NA | 0.227 | NA | 0.012 |
| unipredict/tarkkaanko/amazon | 0.469 | 0.766 | NA | 0.750 | NA | 0.200 |
| unipredict/team-ai/spam-text-message-classifica... | 0.750 | 0.867 | 0.961 | 0.846 | 0.846 | 0.500 |
| unipredict/teertha/ushealthinsurancedataset | 0.312 | 0.617 | 0.711 | 0.258 | 0.708 | 0.250 |
| unipredict/tejashvi14/employee-future-prediction | 0.516 | 0.531 | 0.477 | 0.608 | 0.659 | 0.500 |
| unipredict/tejashvi14/engineering-placements-pr... | 0.617 | 0.594 | 0.773 | 0.487 | 0.724 | 0.500 |
| unipredict/thedevastator/cancer-patients-and-ai... | 0.039 | 0.109 | NA | 0.010 | NA | 0.029 |
| unipredict/thedevastator/employee-attrition-and... | 0.805 | 0.812 | NA | 0.837 | NA | 0.500 |
| unipredict/thedevastator/higher-education-predi... | 0.672 | 0.867 | NA | 0.654 | NA | 0.500 |
| unipredict/therealsampat/predict-movie-success-... | 0.617 | 0.883 | NA | 0.762 | NA | 0.500 |
| unipredict/timoboz/tesla-stock-data-from-2010-t... | 0.992 | 0.984 | 1.000 | 0.244 | 0.907 | 0.250 |
| unipredict/uciml/mushroom-classification | 0.555 | 0.734 | 0.961 | 0.502 | 0.952 | 0.500 |
| unipredict/uciml/pima-indians-diabetes-database | 0.648 | 0.664 | 0.703 | 0.661 | 0.705 | 0.500 |
| unipredict/uciml/red-wine-quality-cortez-et-al-... | 0.344 | 0.344 | 0.461 | 0.385 | 0.479 | 0.200 |
| unipredict/varpit94/tesla-stock-data-updated-ti... | 0.984 | 1.000 | 1.000 | 0.255 | 0.924 | 0.250 |
| unipredict/vedavyasv/usa-housing | 0.242 | 0.305 | NA | 0.242 | NA | 0.250 |
| unipredict/vijayvvenkitesh/microsoft-stock-time... | 0.984 | 0.977 | 0.953 | 0.260 | 0.859 | 0.250 |
| unipredict/vikramamin/customer-churn-decision-t... | 0.711 | 0.711 | 0.664 | 0.696 | 0.746 | 0.500 |
| unipredict/vikramamin/time-series-forecasting-u... | 0.211 | 0.359 | 0.578 | 0.256 | 0.247 | 0.250 |
| unipredict/vstacknocopyright/blood-transfusion-... | 0.484 | 0.570 | 0.672 | 0.787 | 0.749 | 0.500 |
| unipredict/warcoder/earthquake-dataset | 0.820 | 0.820 | NA | 0.259 | NA | 0.250 |
| unipredict/whenamancodes/predict-diabities | 0.672 | 0.695 | 0.781 | 0.661 | 0.718 | 0.500 |
| unipredict/whenamancodes/students-performance-i... | 0.500 | 0.453 | 0.500 | 0.252 | 0.481 | 0.250 |
| unipredict/yasserh/titanic-dataset | 0.711 | 0.727 | 0.805 | 0.587 | 0.721 | 0.500 |
| unipredict/yasserh/wine-quality-dataset | 0.391 | 0.359 | 0.500 | 0.346 | 0.511 | 0.200 |

# I  Model Card

We provide a Model Card for TABULA-8B, as outlined in [35].

## I.1  Model Details

**Person or organization developing model:** This model was developed by the authors of this paper. Organizations providing computational support are listed in the Acknowledgements, but this model is not officially developed as part of any organization. The author affiliations are listed on the first page of this paper.

**Model date:** This paper describes the May 2024 version of TABULA-8B.

**Model version:** This paper describes version 1.0 of TABULA-8B.

**Model type:** TABULA-8B is an autoregressive language model, identical in architecture to Llama 3 [54].

**Information about training algorithms, parameters, fairness constraints or other applied approaches, and features:** Our training procedure is described in Section 3. Our procedure for dataset construction, which includes methods for removing sensitive PII, is described in Sections 4 and A.

**Paper or other resource for more information:** This paper is the primary resource for information about TABULA-8B. Implementation details can also be found at the open-source code release associated with the project.

**Citation details:** See the first page of this paper.

**License:** The model uses the Meta Llama 3 license (see https://llama.meta.com/llama3/license/).

**Where to send questions or comments about the model:** Send questions or comments directly to the corresponding authors, or file issues on the project git repo.

## I.2 Intended Use

**Primary intended uses:** This is a research-only release. The primary intended use of this model is for research on tabular data modeling, or for research applications on tabular data.

**Primary intended users:** The primary intended users are scientific researchers interested in understanding, training, and applying tabular foundation models.

**Out-of-scope use cases:** Commercial use, use of the model to attempt to identify, harm, or violate the privacy of individuals represented in the training data, and any other behavior that violates the Meta Llama 3 license is out of scope.

## I.3 Factors

**Relevant factors:** The original Model Cards paper [35] identifies factors as "groups, instrumentation, and environments" relevant to summaries of model performance. One group relevant to our models' performance is the task type (classification vs. binned regression). We report performance on these tasks separately; our results are discussed in Section 5. Broadly, we find that TABULA-8B's overall performance profile relative to baselines is similar for both classification and binned regression tasks. Similarly, the different benchmarks may be viewed as different *environments*, each testing a different type of dataset. For example, UniPredict tests performance on datasets with informative headers; OpenML-CC18 tests performance on datasets without such headers and where traditional supervised learning methods can be tuned to good performance; Grinsztajn tests performance on datasets where GBDTs tend to perform best; and AMLB tests performance on tasks including free-form text. Our main results show that TABULA-8B's overall performance relative to baselines is similar across these tasks; we analyze the differences in detail in the paper.

**Evaluation factors:** Evaluating language models (LMs) is different from evaluating standard supervised learning methods: while the latter directly output a score or probability over the set of target labels, LMs only output next-token probabilities over their vocabularies; as a result, predicted probabilities are not directly available (although these can be obtained through the use of various heuristics). In order to avoid introducing additional degrees of freedom into the evaluation process, we do not use score-based evaluation methods that rely on evaluating predicted probabilities; we only evaluate based on exact matching (as in several works both in the tabular literature [12, 23] and in the broader language modeling literature [1, 8]). As a consequence, our evaluation does not use metrics which are sometimes used to evaluate tabular classification models, such as Area Under the Receiver Operating Characteristic Curve (AUC).

## I.4 Metrics

**Model performance measures:** Our primary evaluation measures are based on accuracy. We use exact-match accuracy for language model generations, and top-1 accuracy for supervised learning model predictions.

**Decision thresholds:** We use top-1 accuracy for supervised learning model predictions, but do not apply a specific threshold.

**Variation approaches:** N/A

## I.5 Evaluation Data

**Datasets:** We use a suite of five previously-proposed tabular benchmarks, comprising a total of 329 tables. Our evaluation datasets are described in detail in Sections 5.2 and D.1.

**Motivation:** Using preexisting benchmark datasets allows us to compare the performance of our models to prior work in the tabular prediction literature. Additionally, using high-quality, curated benchmarks ensures that we are able to make reliable conclusion about overall model quality and performance relative to baselines.

**Preprocessing:** Our preprocessing is described in Sections 5.2 and D.1. We perform minimal preprocessing on the datasets (no one-hot encoding, standardization, etc.) except for the logistic regression baseline, which requires this for best performance.

### I.6 Training Data

Our training data is described in 4, with further details in the supplementary.

### I.7 Quantitative Analyses

**Unitary results:** Our unitary results are summarized in Section 5. We provide detailed analysis in the supplementary section and give per-dataset results in Section H.

**Intersectional results:** We do not explicitly investigate tasks which include sensitive attributes, and so do not consider intersectional analysis in this work. We call for future work understanding the fairness properties of tabular foundation models in the future work (and our first-of-its-kind model will enable such research).

### I.8 Ethical Considerations

There are important ethical considerations of both the data and model presented in this work. We discuss these in our Impact Statement.

### I.9 Caveats and Recommendations

This model is for research use only. We recommend that more thorough research on both the impact of tabular training datasets, and the downstream performance of fine-tuned language models, be conducted before the deployment of tabular foundation models for real-world decisionmaking deployments.

## J Datasheet

### J.1 Motivation

**For what purpose was the dataset created?**

The dataset was created for training tabular data foundation models, serving a purpose similar to C4 [41] or other large-scale corpuses in the natural language processing community.

**Who created the dataset (e.g., which team, research group) and on behalf of which entity (e.g., company, institution, organization)?**

The dataset was created by the authors of this paper in their roles at the institutions listed in the affiliations section of this paper.

**Who funded the creation of the dataset?**

No funding was provided with the explicit purpose of creating this dataset. However, JG was supported by a Microsoft Grant for Customer Success. JCP was supported by the Harvard Center for Research on Computation and Society.

### J.2 Composition

**What do the instances that comprise the dataset represent (e.g., documents, photos, people, countries)?**

The instances that comprise the dataset represent tables extracted from the web (or individual rows of tables, depending on the downstream use of the data). All tables are publicly available and extracted from the Internet; in particular, all tables are available in the TabLib dataset from which T4 is filtered.

**How many instances are there in total (of each type, if appropriate)?**

The dataset consists of 4.2M tables where each table has many rows. The total number of rows across tables is 2.1B.

**Does the dataset contain all possible instances or is it a sample (not necessarily random) of instances from a larger set?**

The dataset consists of a deterministically filtered subset of the TabLib dataset.

**What data does each instance consist of?**

The data consists of tables drawn from Github and CommonCrawl, as initially captured by the TabLib authors.

**Is there a label or target associated with each instance?**

Not by default. As part of our work, we select a target at random from filtered subset of the columns for each dataset. See A.2.

**Is any information missing from individual instances?**

Yes, many tables contain missing values for certain rows and columns.

**Are relationships between individual instances made explicit (e.g., users' movie ratings, social network links)?**

In some cases, the tables have informative column headers describing the relationships between features for individual rows.

**Are there recommended data splits (e.g., training, develop- ment/validation, testing)?**

Not by default. We implement these as part of our training.

**Are there any errors, sources of noise, or redundancies in the dataset?**

We deduplicate the T4 dataset so that each table appears at most once. However, as is the case with most internet scale datasets, many of the tables contain noisy values whose correctness we do not manually inspect.

**Is the dataset self-contained, or does it link to or otherwise rely on external resources (e.g., websites, tweets, other datasets)?**

It is self-contained.

**Does the dataset contain data that might be considered confidential (e.g., data that is pro- tected by legal privilege or by doctor–patient confidentiality, data that includes the content of individuals' non- public communications)?**

We do our best to remove any kind of data that might be considered confidential or personally identifying. If someone finds tables with confidential information that still remain, we would appreciate if they contact us so we might remove them.

**Does the dataset contain data that, if viewed directly, might be offen- sive, insulting, threatening, or might otherwise cause anxiety?**

It is possible that there are tables with information that might be anxiety inducing. We do not explicitly filter for this type of information, but believe it is not common in our dataset.

**Does the dataset identify any subpopulations (e.g., by age, gender)?**

The instances (table rows) in the data represent a variety of entities, and the majority of these do not represent persons. However, for the subset of tables where each row does represent an individual person, it is possible that the dataset does identify subpopulations.

**Is it possible to identify individuals (i.e., one or more natural persons), either directly or indirectly (i.e., in combination with other data) from the dataset?**

While we aim to reduce obviously personally identifying data, we do not use techniques like differential privacy to formally defend against reidentification attacks.

**Does the dataset contain data that might be considered sensitive in any way (e.g., data that reveals race or ethnic origins, sexual orienta- tions, religious beliefs, political opinions or union memberships, or locations; financial or health data; biometric or genetic data; forms of government identification, such as social security numbers; criminal history)?**

We aim to remove this kind of information.

**Any other comments?**

### J.3 Collection Process

**How was the data associated with each instance acquired?**

The data was filtered from the original TabLib dataset [13].

**What mechanisms or procedures were used to collect the data (e.g., hardware apparatuses or sensors, manual human curation, software programs, software APIs)?**

The data was filtered programatically according to hand picked heuristics as described in 4.

**If the dataset is a sample from a larger set, what was the sam- pling strategy (e.g., deterministic, probabilistic with specific sam- pling probabilities)?**

It was deterministically chosen according to hand coded filtering rules.

**Who was involved in the data collection process (e.g., students, crowdworkers, contractors) and how were they compensated (e.g., how much were crowdworkers paid)?**

The authors performed the data collection process. No external crowdsourcing or contractors were employed.

**Over what timeframe was the data collected?**

The original TabLib dataset was collected in 2023 by the original authors and contains tables published from a wide range of years. Our filtering was conducted during the spring of 2024.

**Were any ethical review processes conducted (e.g., by an institu- tional review board)?**

No.

**Did you collect the data from the individuals in question directly, or obtain it via third parties or other sources (e.g., websites)?**

Data was filtered from TabLib and hence not collected directly.

**Were the individuals in question notified about the data collection?**

We notified the TabLib authors of our effort, but not the owners of the publicly available tables in the original corpus.

**Did the individuals in question consent to the collection and use of their data?**

To the best of our knowledge, the data scraped in TabLib did not have a consent procedure.

**If consent was obtained, were the consenting individuals provided with a mechanism to revoke their consent in the future or for certain uses?**

NA

**Has an analysis of the potential impact of the dataset and its use on data subjects (e.g., a data protection impact analysis) been conducted?**

No.

### J.4 Preprocessing/cleaning/labeling

**Was any preprocessing/cleaning/labeling of the data done (e.g., discretization or bucketing, tokenization, part-of-speech tagging, SIFT feature extraction, removal of instances, processing of missing values)?**

Yes.

**Was the "raw" data saved in addition to the preprocessed/cleaned/labeled data (e.g., to support unanticipated future uses)?**

The raw data is available as part of the TabLib release [13].

**Is the software that was used to preprocess/clean/label the data available?**

Yes, all the code used to filter the original corpus is available as part of our open source release.

### J.5 Uses

**Has the dataset been used for any tasks already?**

We are not aware of any other uses of T4 apart from the training of TABULA-8B.

**Is there a repository that links to any or all papers or systems that use the dataset?**

Models trained on the dataset can be found using the Hugging Face dataset page.

**What (other) tasks could the dataset be used for?**

The dataset could be used to train generative models, LLM data science assistants, amongst others.

**Is there anything about the composition of the dataset or the way it was collected and preprocessed/cleaned/labeled that might impact future uses?**

Our filtering choices were optimized to train a tabular prediction (classification) model and may lead to suboptimal behavior for other tasks.

**Are there tasks for which the dataset should not be used?**

The dataset should not be used to try and identify private individuals.

### J.6 Distribution

**Will the dataset be distributed to third parties outside of the entity (e.g., company, institution, organization) on behalf of which the dataset was created?**

The dataset will be made publicly available on HuggingFace.

**How will the dataset will be distributed (e.g., tarball on website, API, GitHub)?**

It will be published at HuggingFace.

**When will the dataset be distributed?**

June 2024

**Will the dataset be distributed under a copyright or other intellectual property (IP) license, and/or under applicable terms of use (ToU)?**

The dataset is subject to the same usage and copyright restrictions as the original TabLib release.

**Have any third parties imposed IP-based or other restrictions on the data associated with the instances?**

Yes, the original TabLib authors place usage restrictions. See [] for more details.

**Do any export controls or other regulatory restrictions apply to the dataset or to individual instances?**

The dataset is subject to the same usage and copyright restrictions as the original TabLib release.

### J.7 Maintenance

**Who will be supporting/hosting/maintaining the dataset?**

The authors of the paper.

**How can the owner/curator/manager of the dataset be contacted (e.g., email address)?**

Please contact jpgard@cs.washington.edu.

**Is there an erratum?**

The current manuscript, as published on the arxiv server, will serve as the main source of documenting errors.

**Will the dataset be updated (e.g., to correct labeling errors, add new instances, delete instances)?**

We do not foresee any updates.

**If the dataset relates to people, are there applicable limits on the re- tention of the data associated with the instances (e.g., were the individuals in question told that their data would be retained for a fixed period of time and then deleted)?**

NA

**Will older versions of the dataset continue to be supported/hosted/maintained?**

NA

**If others want to extend/augment/build on/contribute to the dataset, is there a mechanism for them to do so?**

Others are free to build on the dataset as long as they adhere to the original terms of use put forth by [13].

## NeurIPS Paper Checklist

The checklist is designed to encourage best practices for responsible machine learning research, addressing issues of reproducibility, transparency, research ethics, and societal impact. Do not remove the checklist: **The papers not including the checklist will be desk rejected.** The checklist should follow the references and follow the (optional) supplemental material. The checklist does NOT count towards the page limit.

Please read the checklist guidelines carefully for information on how to answer these questions. For each question in the checklist:

- You should answer [Yes] , [No] , or [NA] .
- [NA]  means either that the question is Not Applicable for that particular paper or the relevant information is Not Available.
- Please provide a short (1–2 sentence) justification right after your answer (even for NA).

**The checklist answers are an integral part of your paper submission.** They are visible to the reviewers, area chairs, senior area chairs, and ethics reviewers. You will be asked to also include it (after eventual revisions) with the final version of your paper, and its final version will be published with the paper.

The reviewers of your paper will be asked to use the checklist as one of the factors in their evaluation. While "[Yes] " is generally preferable to "[No] ", it is perfectly acceptable to answer "[No] " provided a proper justification is given (e.g., "error bars are not reported because it would be too computationally expensive" or "we were unable to find the license for the dataset we used"). In general, answering "[No] " or "[NA] " is not grounds for rejection. While the questions are phrased in a binary way, we acknowledge that the true answer is often more nuanced, so please just use your best judgment and write a justification to elaborate. All supporting evidence can appear either in the main paper or the supplemental material, provided in appendix. If you answer [Yes]  to a question, in the justification please point to the section(s) where related material for the question can be found.

IMPORTANT, please:

- **Delete this instruction block, but keep the section heading "NeurIPS paper checklist",**
- **Keep the checklist subsection headings, questions/answers and guidelines below.**
- **Do not modify the questions and only use the provided macros for your answers**.

