# OpenReview forum: "Large Scale Transfer Learning for Tabular Data  via Language Modeling"
_NeurIPS.cc/2024/Conference — NeurIPS 2024 poster_

### Official Review · Reviewer_DUWr · 2024-06-15

**Soundness:** 4
**Presentation:** 4
**Contribution:** 3
**Rating:** 7
**Confidence:** 5

**Summary:**

This paper presents a large tabular prediction model based on large language models (LLMs). The model was trained on 800M rows from 1.5M tables, hence achieves high zero-shot accuracy on unseen tables. To make it work, the authors made extensive data collection, cleaning, and filtering to build a large-scale tabular datasets, based on TabLib. The yielded model was shown to outperform XGBoost and

**Strengths:**

- The dataset size scales up, shows the promising results that LLMs trained on tabular datasets at scale can extrapolate to unseen tables for prediction.

**Weaknesses:**

- The paper overstates its novelty in applying transfer learning to tabular prediction tasks and fails to discuss several critical prior studies, including references [1, 2, 3, 4], among others.

- The evaluation does not include comparisons with any existing cross-table transfer learning baselines, which could provide a more comprehensive assessment of the proposed method's efficacy.

- The model predicts values for target columns without providing confidence scores or probabilities for each class, making it challenging to trust and use in real-world scenarios.

- There is a potential data leakage issue that needs careful attention, as the evaluation data might have been included in the large-scale web-crawled training dataset.



[1] Wang, Z., & Sun, J. (2022). Transtab: Learning transferable tabular transformers across tables. Advances in Neural Information Processing Systems, 35, 2902-2915.

[2] Zhu, B., Shi, X., Erickson, N., Li, M., Karypis, G., & Shoaran, M. (2023). Xtab: Cross-table pretraining for tabular transformers. arXiv preprint arXiv:2305.06090.

[3] Zhang, T., Wang, S., Yan, S., Li, J., & Liu, Q. (2023). Generative table pre-training empowers models for tabular prediction. arXiv preprint arXiv:2305.09696.

[4] Ye, C., Lu, G., Wang, H., Li, L., Wu, S., Chen, G., & Zhao, J. (2023). CT-BERT: learning better tabular representations through cross-table pre-training. arXiv preprint arXiv:2307.04308.

**Questions:**

Pls refer to the weakness section.

---

> ### Author Response · Authors · 2024-08-03
>
> Thank you for taking the time to read our paper, provide constructive comments and insightful connections to related works that will improve the quality of our manuscript. We’re very encouraged to see you appreciate the value of our scaling efforts to build models that can achieve “high zero-shot accuracy on unseen tables”. We respond to your comments below:
>
> _**Relationship to Prior Work and Baselines.**_
>
> Thank you for bringing these important and relevant papers to our attention. We will certainly add them and discuss them in our updated manuscript. While relevant, we believe these papers provide clearly distinct and complementary perspectives. In particular,
>
> - Xtab Zhu et al 2023 [2]: Xtab provides a novel method for cross training on many tables, however it requires explicitly fine tuning a featurizer and projection head on each downstream evaluation dataset. As such, it falls outside the scope of our benchmark comparisons since we are interested in deep learning methods that do not require training on the test task (e.g only require a forward pass).
>
> - Zhang et al 2023 [3]: This paper provides an interesting method for data augmentation on tabular datasets, rather tabular prediction per se. It can be deployed in addition to our model to improve performance, but we do not feel that it is an appropriate baseline for our model, as it only provides a method for augmentation, not an end-to-end prediction method.
>
> - TransTab, Wans & Sun 2022 [1] & Ye et al 2023 [4]: As is the case with Xtab, the methodology proposed by these authors requires fine tuning the model on every downstream dataset. It falls into a different class of methods and it’s not evident from the original works that one can scale up pre-training their model to millions of tables.
>
> In terms of other cross table learning baselines, we compare against the base Llama 3 model which is trained on 15T tokens. We are currently also adding a comparison against the few shot performance of the latest Claude model. We would be happy to include additional baselines capable of zero- and few-shot prediction on tabular data, if pretrained models are available along with instructions on how to apply the models on new data; please feel free to point us to other existing high-quality, open-source implementations we may have missed so that we may include them in our results.
>
> _**Class Probabilities**_
>
> Since our classifier uses an LLM, it can be extended to provide class probabilities in a straightforward fashion. Language models do this natively. Given the serialized example, we can compute the likelihood with which the model will complete the prompt using each of the possible labels. These would be the class probabilities. This has been shown to be effective in other few shot learning papers that use LLMs (see e.g [6]) and we can add this functionality to final version of the paper.
>
> _**Potential Data leakage.**_
>
> In Section 5.6 and Appendix G, we describe how we tested for the possibility of downstream datasets being included in the pretraining set. Please see the paper for a precise description. To test whether a table in the test set appears in the training set, for each table, we check whether there exists a table in T4 that contains the same set of columns and column names.
>
> Using this conservative procedure, which is likely to over report the number of tables in the test set that appear in T4, we report results on the tables from our eval set which we confidently know are not on the training set. We report the results of these evaluations in Figure 8 and the bottom right subplot of Figure 4.
>
> As mentioned in the paper, we find that Tabula performs better on these tables: both in an absolute sense (the average accuracy is higher) and also in a relative sense (the gap to XGBoost is higher than on the possibly contaminated set). We conclude from these results that contamination does not appear to be a concern. We note that this finding is in line with previous high-profile evaluations of dataset contamination, including the original GPT2 and 3 papers [5,6].
>
> _**References**_
>
> - [1] Wang, Z., & Sun, J. (2022). Transtab: Learning transferable tabular transformers across tables. Advances in Neural Information Processing Systems, 35, 2902-2915.
> - [2] Zhu, B., Shi, X., Erickson, N., Li, M., Karypis, G., & Shoaran, M. (2023). Xtab: Cross-table pretraining for tabular transformers. arXiv preprint arXiv:2305.06090.
> - [3] Zhang, T., Wang, S., Yan, S., Li, J., & Liu, Q. (2023). Generative table pre-training empowers models for tabular prediction. arXiv preprint arXiv:2305.09696.
> - [4] Ye, C., Lu, G., Wang, H., Li, L., Wu, S., Chen, G., & Zhao, J. (2023). CT-BERT: learning better tabular representations through cross-table pre-training. arXiv preprint arXiv:2307.04308.
> - [5] Radford et al. Language Models are Unsupervised Multitask Learners. 2022
> - [6] Brown et al. Language Models are Few-Shot Learners. 2023
> - [7] Awadalla et al. OpenFlamingo. 2023

---

> > ### Comment · Reviewer_DUWr · 2024-08-12
> >
> > Thanks for your reply. I've improved the score to 7.

---

### Official Review · Reviewer_7NU9 · 2024-07-13

**Soundness:** 3
**Presentation:** 3
**Contribution:** 3
**Rating:** 6
**Confidence:** 5

**Summary:**

The paper introduces TABULA-8B, a specialized large language model for tabular data prediction tasks. It details a comprehensive process for extracting a high-quality, large-scale dataset from the TabLib corpus, labeled as T4, which comprises 1.5 million unique tables and over 800 million rows of data. By fine-tuning the Llama 3-8B large language model on the T4 dataset using a novel packing and attention scheme, TABULA-8B achieves superior performance in tabular data prediction, including classification and binned regression tasks. Through extensive evaluation on a suite of 300 datasets, TABULA-8B demonstrates a groundbreaking zero-shot accuracy that exceeds random guessing by over 15 percentage points, outperforming existing state-of-the-art models like XGBoost and TabPFN. In few-shot settings (1-32 shots), without any fine-tuning on target datasets, TABULA-8B achieves 5-15 percentage points higher accuracy than these models, even when they are trained on up to 16 times more data. Additionally, TABULA-8B's capability to perform zero-shot learning marks a significant advancement over previous methods.

**Strengths:**

1. TABULA-8B achieves over 15% accuracy on unseen tabular data in zero-shot settings, outperforming random guessing and existing models. In few-shot settings, it demonstrates 5-15% higher accuracy than state-of-the-art models like XGBoost and TabPFN, even when those models are trained on much larger datasets, which demonstrates the TABULA-8B's excellent performance.
2. The paper includes robustness and ablation studies to investigate the impact of various procedures such as data filtering and causal masking. This helps in understanding the contributions of different components to the model's performance.
3. The paper addresses two significant limitations of existing methods: the lack of large-scale training data and the inability to remain competitive when evaluated on out-of-distribution data.
4. This paper demonstrates the potential of language models in generating accurate predictions from tabular data sets, and is likely to set a new standard for future research.

**Weaknesses:**

1. The contribution of this paper is not novel enough. Although a new dataset T4 is proposed, it is extracted and filtered from the existing public corpus and does not generate challenging data. Besides, block-causal attention mask is also a common attention method. Despite the impressive results of the TABULA-8B model, the innovation of its contribution remains to be improved.
2. Additional baseline experiments could be added to confirm the high performance of TABULA- 8B. See Q1.
3. The paper could conduct a more detailed analysis of the training and test table data, introduce the length distribution of the table, and whether it is a relational or non-relational table. While the paper proposes a series of filtering rules to ensure data quality, these rules primarily target overall table features such as row count and column count. There is a lack of evaluation for the internal data quality of tables, such as whether the data is too simple or close to real-world data, can lead to inconsistent data quality. See Q2.
4. As shown in Figure 5, due to the limitation of TABULA-8B context window, the examples that can be used for few-shot learning are limited. With the increase of the number of k samples, the effect of all models is improved, and when more than 64 samples are used, the effect of XGBOOST is almost the same as that of TABULA-8B. Considering the expensive training and reasoning cost of TABULA 8B, it cannot completely replace XGBOOST at present, which can still be improved.

**Questions:**

Q1: Although XGBOOST is widely considered to be highly competitive in table prediction tasks, with the development of LLMs, there are also many LLMs applications in table prediction tasks, such as UniPredict[52] mentioned in the paper. Could you compare the results with these latest works? In addition, the large language model llama-8B is used as the baseline in the paper. How do other large language models such as GPT-3.5 and GPT-4 perform on this task? Does TABULA-8B perform better than large language models such as GPT-3.5 and GPT-4?
Q2: Are all the tables mentioned in the paper relational tables? A big percentage of tables are non-relational in the real world. Could TABULA-8B achieve the excellent prediction effect on non-relational tables?
Q3: In appendix F.1, in line 745, it appears that the sentence is incomplete and there is unfinished content.

**Limitations:**

The author has given a detailed explanation of the limitations in the paper.

---

> ### Author Response · Authors · 2024-08-03
>
> Thank you for reading our manuscript in detail and providing a constructive review. We’re very encouraged to hear how you believe that our methodology and results “will likely set a new standard for future research” in this area. Please see our responses to your concerns below, and thank you in advance for your dedication to this discussion period.
>
> _**Novelty.**_
>
> It is true that our paper does not propose a new architecture or optimization algorithm. However, the goal of a scientific paper is not necessarily methodological novelty but generating new knowledge and insights, which you can do by combining an existing method with new training data for instance. This has especially been the case with many landmark results in AI in recent years (e.g GPT3.5, Llama 3).
>
> To quote from the Llama 3 technical report [2], “Llama 3 uses a standard, dense Transformer architecture. It does not deviate significantly from Llama and Llama 2 in terms of model architecture; our performance gains are primarily driven by improvements in data quality and diversity as well as by increased training scale.” Despite not presenting new algorithms these research endeavors have undoubtedly had massive impact.
>
> Like these breakthrough results, we performed careful ablation experiments with attention masks and optimization routines that empowered our results, and spent most of our effort in designing robust and scalable procedures that increase training data by four orders of magnitude relative to previous tabular language modeling papers  [1,3]. The main novelty of our work is that we are the first to outperform state of the art models like XGBoost on unseen datasets in the few shot regime without any finetuning on the target data.
>
> Furthermore, providing a high-quality dataset that is “extracted and filtered from the existing public corpus” is also a contribution that can both be extremely impactful, and is in line with previous breakthrough efforts in other domains (e.g. the C4 dataset [4], for example, is a high-quality filtered version of the public Common Crawl and is widely used in SOTA language models).
>
> _**Baselines.**_
>
> We are grateful for the reviewers’ excellent suggestion to compare our method to state-of-the-art generalist LLMs such as GPT3.5 and GPT-4.  We will evaluate the performance of Claude 3.5-Sonnet, a current SOTA commercial LLM that provides a strong balance between price and performance, on our benchmark datasets, and is capable of zero- and few-shot prediction with no fine-tuning on the target datasets. We will report back on these results in the next few days. Thank you again for the suggestion and we look forward to sharing the results!
>
> In terms of other deep learning baselines, comparing to methods like Unipredict is out of scope since there is not currently an open-source model for us to use for evaluations (the current UniPredict repository is minimal, and it says “The official repository will be released after the review process finishes” and that training from scratch using the current code “ is not recommended” [1]). More importantly, UniPredict requires fine-tuning on every downstream target dataset, a limitation we explicitly aim to avoid. Please recall that our method does not require any gradient updates on the downstream tables, we only need to perform a forward pass at inference time – this is a key differentiator from many previous works.
>
> _**T4 Summary Statistics and Relational vs Non Relational Tables.**_
>
> We are grateful for the reviewers’ in-depth interest in understanding data quality – this is also important to us! Perhaps the reviewer can clarify what they mean by a relational vs non-relational table, in the context of our data and task setup. Each table in our evaluation suite is a standalone table. Different tables refer to different entities and are in that sense non-relational. We agree with the reviewer that performance on unseen and unrelated tables is the gold standard for transfer learning and this is the core philosophy behind our choice of evaluation set.
>
> We agree with the reviewer that more detailed summary information about the T4 dataset would be useful! Thank you for this request. We provide broad summary statistics about T4 in Figure B of the pdf of the author response. These include histograms of # of rows and columns per table, as well as the various distributions of data types.

---

> > ### Author Response · Authors · 2024-08-03
> >
> > _**Large shot behavior vs XGBoost.**_
> >
> > The goal of our paper is not to outperform XGBoost when the number of shots is large, but rather when the training set is small. We believe that tabular prediction in low data regimes is a fundamentally important problem. This is particularly true in settings like health, or education, where an institution (i.e a school or hospital) wants to develop a predictor for their specific, local population but lacks a large historical database of cases.
> >
> > In this regime, our method significantly outperforms classical, SOTA algorithms like XGBoost that cannot leverage large scale pre-training datasets. Further, we note that TabPFN is also very limited in what data it can be used on (no more than 10 classes, no more than 100 features, limited overall context window) but has been highly impactful as a tabular few-shot learning method (and our method in fact even outperforms TabPFN, without having these limitations).
> >
> > L745: Thank you for pointing this out, we will fix this. We meant to restate our definition of tabular data here as presented in L67-72 in Section 1.2.
> >
> >
> > _**References:**_
> >
> > - [1] Wang et al. UniPredict: Large Language Models are Universal Tabular Classifiers, 2023
> > - [2] Llama AI Team, Meta. The Llama 3 Herd of Models, 2024
> > - [3] Hegselmann et al, TabLLM: Few-shot classification of tabular data with large language models. AISTATS, 2023
> > - [4] Raffel et al. Exploring the Limits of Transfer Learning with a Unified Text-to-Text Transforme. JMLR 2020

---

> ### Author Response · Authors · 2024-08-05
> **Follow-up: Claude Evaluation Results**
>
> We are following up on our previous comments with results comparing our model to Claude, a frontier AI model from Anthropic.  In particular, the reviewer asked:
>
> > Does TABULA-8B perform better than large language models such as GPT-3.5 and GPT-4?
>
> We provide results comparing our model to both Claude 3 Sonnet and Claude Instant, in order to provide comparisons to both very strong commercial LLMs (Claude 3 Sonnet) and mid-tier commercial LLMs (Claude Instant) at two different price-performance ratios. **Simply put, the answer to the reviewers’ question is “yes” – our model outperforms both versions of Claude across all values of shots evaluated.** Please find the results curves in the author rebuttal PDF.
>
> Below, we describe the Claude experiments in slightly more detail and provide a more detailed interpretation. However, we emphasize that the main takeaway of this experiment is to provide clear evidence that our model outperforms strong commercial LLMs. Again, we are grateful to the reviewer for suggesting these additional experiments and believe that they will considerably strengthen the paper!
>
> # Experiment Design - Claude 3 Sonnet and Claude Instant Evaluation
>
> We perform the same procedure for both Claude 3 Sonnet and Claude Instant. For each model, we serialize inputs in a format identical to that used in the paper. During evaluation we use 128 randomly-selected samples at every value of k (number of shots) and from every table (random shots are also selected IID for every sample in the same manner as the experiments in the paper). However, we also do the following in accordance with Claude’s training and recommended usage:
> 1. Follow the “Human:” “Assistant:” format recommended in the Claude user guide.
> 2. Add a prompt describing the general task and the input/output format. This prompt is the same for every dataset.
>
> Claude models are rate-limited. As such, due to the very large number of evaluations required (330 benchmark datasets at 1, 2, 3, 4, 8, 16, 32 shots, all with 128 samples each), we were not able to complete the evaluations on our entire benchmark suite prior to the closure of the rebuttal window. However, we provide the complete current set of results (74 total benchmark tasks at 16 shots, 46 of which are also 32-shot) which we believe provide a strong representative sample from our evaluation suite. We commit to adding the complete set of results to the camera-ready version of the paper.
>
> # Results Interpretation
>
> The results (shown in the rebuttal PDF) show the following:
> * **TabuLa outperforms both Claude models across all numbers of shots evaluated.** Our model achieves significantly higher few-shot accuracy (shown by the nonoverlapping confidence intervals) at every point evaluated on the curves. TabuLa outperforms Claude 3 Sonnet by 10-20% everywhere.
> * **Stronger commercial LLMs also have stronger few-shot performance.** Claude 3 Sonnet performs better than Claude Instant, which reflects the larger model capacity and (likely) larger training data of Claude 3 Sonnet. This also reflects their relative rankings according to various benchmark metrics and is a useful sanity check of our overall evaluation approach.
> * **Commercial LLMs do not always improve monotonically with more shots.** It has been noted in the literature (for example, GPT-2 and GPT-3 papers) that more shots do not always improve the performance of generalist LLMs in few-shot learning. Our models’ capacity to monotonically improve with more shots is a key advantage of our training procedure. Our results also show that the two Claude models evaluated demonstrate this behavior – the performance actually tends to level off after 4-8 shots, with no further improvements.
> * **RLHF may drive the observed differences in behavior.** It is widely known that commercial LLMs (likely including all Claude and ChatGPT models) undergo a post-training procedure where the model is aligned to human preferences using reinforcement learning from human feedback (RLHF). The Claude models are the *only* such models in our study. This suggests (but does not prove) that RLHF may be a factor driving the different performance of the Claude models vs. other models (and particularly the non-RLHF models like TabuLa and Llama 3 7B base model). It is possible that RLHF decreases models’ ability to learn our task in-context. Again, this contrast demonstrates the strength of our approach.
>
> # Prompt Used
>
> You are performing a classification task on a tabular dataset.
> Below you will be provided with a few rows of tabular data, with the label for each row.
> The final row in the tabular dataset will not contain a label.
> Your task is to predict the label for that row, using the choices provided.
>
> Instructions:
> * Predict the target label for the final row.
> * Always choose one of the provided classes.
> * Only return one of the provided classes; do not return any other text.
>
> Data:

---

> > ### Comment · Reviewer_7NU9 · 2024-08-12
> >
> > I appreciate the authors' detailed feedback. Thank you for answering my concerns in detail from the perspectives of novelty, baseline experiment, data statistics, etc. Although I think there are still some limitations in terms of novelty, the experiment is comprehensive, and the results are very detailed. I will raise the score.

---

> > > ### Author Response · Authors · 2024-08-12
> > >
> > > Thank you for your engagement during the dialogue window, and for your thoughtful consideration of both our initial submission and rebuttal/additional results. We are glad that our response and our additional results answer your concerns from the perspectives of "novelty, baseline experiment, data statistics, etc.", and that you found the results "comprehensive."
> > >
> > > We appreciate your increase in the score. If there are further clarifications we can provide before the dialogue window closes, please let us know.

---

### Official Review · Reviewer_3ko2 · 2024-07-13

**Soundness:** 3
**Presentation:** 3
**Contribution:** 3
**Rating:** 6
**Confidence:** 4

**Summary:**

The paper presents a framework in which it curates a large collection of tabular datasets, fine-tunes a language model that can readily be used for few-shot and zero-shot learning.

**Strengths:**

- The huge collection of tabular data for pretraining can be very useful.
- The proposed method show strong performances in zero-shot and few-shot settings.
- The proposed framework provides a heuristic, but effective way of selecting target column. This is important for the pretraining to have direct resemblence to the downstreams.

**Weaknesses:**

- It would be interesting to see some results with larger train-size.
- It is difficult to grasp the importance of few-shot and zero-shot learning for application perspectives. Some examples may aid the understandings.
- It would be interesting to see CatBoost and linear models or random forests to be included as baselines.
- It may be important to include a simple screening process in which the downstream datasets are not included the fine-tuning of language model step (as the paper concentrates on 'unseen' tables).
- The supplementary section can be cleaned.

**Questions:**

- What are the some scenarios of zero-shot and few-shot learning in real-world applications?
- Are there any screening process for leakage of downstream datasets?
- How does the model compare in terms of computation time for the downstream tasks? I would guess it would be the inference time, since it does not fine-tune on the specific target dataset.
- What are some difficulties of (or reasons for not employing) fine-tuning for specific target dataset?
- How does the model handle missing values? Is it simply ignored in the serialization step or is there a special token to handle them?
- One possible advantage of using language models for pretraning is that it provides a general representation across tables. In this sense, it can be used for transfer learning or domain adaptation across the tables. Would this model be applicable for such cases? (for instance, wine in France and wine in Italy).

**Limitations:**

Yes, the authors have addressed the limitations.

---

> ### Author Response · Authors · 2024-08-03
>
> Firstly, thank you for taking the time to review our paper and bringing up a number of interesting comments that we respond to below. We appreciate your dedication to the discussion period.
>
> _**Larger train size.**_
>
> Our model is trained on 8b tokens, which after various performance optimizations, took 6 days on a node of 8 80GB A100s. Due to financial and computational limitations, we could not afford to train for longer. We agree however that it is interesting and valuable to understand how scale, and the quality of the initial model, affects performance. We also believe that the open-source release of our data, model, and pretraining code will enable future work in this direction.
>
> To answer this question, regarding how train size affects performance while respecting our compute and time constraints, we follow the identical training procedure for the final Tabula8b model, except that we train for 1/10th as long (process 90% fewer tokens). We present these in Figure A in the pdf included in the general author response.By comparing Tabula-8b to the compute matched baseline that sees fewer examples, we see that increasing training size by 10x improves performance but only slightly. Hopefully this answers your question.
>
> As an additional experiment, we also inspect how the final performance of the model varies as we change the initial language model from llama 3 to llama 1 and 2. We train llama 1 and 2 models using our methodology for the same number of tokens as the Tabula 8b compute-matched baseline.
>
> This experiment highlights one of the core advantages of our proposal. More than proposing a specific algorithm, our paper develops a core methodology that adapts language models to tabular classification tasks. As open source LLMs continue to improve, so will the downstream performance of models that are adapted via our procedure -- as evidenced by the large gap between the final performance Llama 1 and the Tabula 8b compute matched baseline. By releasing our comprehensive software suite, we also empower the broader community to further develop this technology.
>
> _**Importance of Zero and Few Shot Learning.**_
>
> Prediction on tabular data in low data or few shot regimes is an important problem whenever we want to localize models to specific domains. This is often the case in regimes like health, finance, or education, where an institution (i.e a school, bank, or hospital)  wants to develop a predictor for their specific population but lacks a large historical database of similar cases. Our work shows how by leveraging Tabula-8b, one can significantly expand the scope of what’s possible relative to fitting standard methods like XGBoost on that data. Importantly, small improvements in prediction may be enough to solve important resource allocation problems, like vaccine distribution or targeted cash transfers [1]. Better few shot models democratizes access to prediction in low-resource, data-poor domains.
>
> Furthermore, few shot learning already has proven to be extremely impactful in other modalities like vision and language [3]. Please see this ACM article for a broader survey regarding the impact zero and few shot learning has had in practice [4].
>
> _**Other Baselines.**_
>
> Thank you for the suggestion. We are currently running these experiments and will add logistic regression and catboost as baselines (we note that CatBoost in particular has emerged as a very strong tabular classifier in the full-shot regime; random forest tends to be less competitive). We will include these baselines in a follow-up post before the close of the dialogue window. We do not compare to Random Forest as it does not support categorical features [2] due to the computational complexity of high-cardinality categories. Hence, it could not be evaluated on a large subset of our datasets.

---

> > ### Author Response · Authors · 2024-08-03
> >
> > _**Screening Downstream Datasets.**_
> >
> > In Section 5.6 and Appendix G, we describe how we tested for the possibility of downstream datasets being included in the pretraining set. Please see the paper for a precise description, but to test whether a table in the test set appears in the training set, we check whether there exists a table in T4 that contains the same set of columns and column names as a subset of its columns.
> >
> > Using this conservative procedure, which is likely to over report the true number of tables in the test set that appear in T4, we report results on the subset of tables from our eval set which we know are not on the training set. We report the results of these evaluations in Figure 8 and the bottom right subplot of Figure 4. As mentioned in the paper, we find that Tabula performs better on these tables that contain no overlap with the training set: both in an absolute sense (the average accuracy is higher) and also in a relative sense (the gap to XGBoost is higher than on the possibly contaminated set). We conclude from these results that overfitting to the training set is not a concern. We note that this finding is in line with previous high-profile evaluations of dataset contamination, including the original GPT2 and 3 papers [5,6].
> >
> > _**Supplementary Section.**_
> >
> > Thank you for the suggestion (and for reviewing the supplementary material). We will invest effort in improving the clarity of the appendix. In addition to incorporating various revisions proposed by other reviewers, we would be happy to incorporate any specific edits the reviewer feels are appropriate. We believe that the extensive supplementary results (e.g. extra ablation studies, detailed per-dataset results, etc.) are important to our paper and we will conduct a thorough revision for the camera-ready to maximize the clarity of the supplement.
> >
> > _**Inference time.**_
> >
> > One of the core advantages of our approaches relative to previous deep learning methods for tabular data, is that we only need to perform a single forward pass through the model to get a prediction, we do not need to fine tune on the target datasets. On average, it takes about 1s to do a forward pass through the model on a single 40GB A100 GPU.
> >
> > This is significantly faster than fine-tuning even a much smaller model in terms of wall clock time. It is comparable to the time it takes to fit an XGBoost model. However, making inferences (predictions) using a trained XGBoost is significantly cheaper (predicting on a single example using XGBoost takes about .001s).  Holistically speaking, the overall inference time costs of one method vs the other depend on the total size of the dataset.

---

> ### Author Response · Authors · 2024-08-03
>
> _**Handling Missing Values.**_
>
> Thank you for the question. This is another key advantage of our method. To perform inference on rows with missing values, we do not perform any imputation or preprocessing. For missing or null values, we simply retain the value (e.g. ‘nan’ or ‘None’) in the serialized data. No special token is used. This allows the model to directly learn to process and represent missing or null data during its training, without any additional preprocessing.
>
> _**Learning General Representations.**_
>
> Yes, absolutely! Thank you for raising this. This is one of the main motivations behind our work and one of the core strengths of the Tabula model. It is precisely because of its ability to leverage large scale pretraining on diverse datasets (wine in france) that tabula can learn meaningful representations and generate good predictions on downstream datasets. One core piece of evidence is its ability to predict labels zero shot much better than what is information-theoretically possible by random guessing. See Figure 1 in the paper.
>
> _**References:**_
>
> - [1] JC Perdomo, The Relative Value of Prediction in Algorithmic Decision Making; ICML 2024
> - [2] See this PR for scikit-learn to add the feature, which has been in progress for over five years: https://github.com/scikit-learn/scikit-learn/pull/12866
> - [3] Alayrac et al. Flamingo: a Visual Language Model for Few-Shot Learning. Neurips 2022.
> - [4] Wang et al. Generalizing from a Few Examples: A Survey on Few-Shot Learning. ACM 2020
> - [5] Radford et al. Language Models are Unsupervised Multitask Learners. 2022
> - [6] Brown et al. Language Models are Few-Shot Learners. 2023

---

> > ### Author Response · Authors · 2024-08-04
> > **Follow up - additional baseline results request by reviewer**
> >
> > Following up, in our rebuttal PDF we are also including results curves which include, in addition to the baselines from the original text, a linear model (logistic regression) and CatBoost. We use the same procedure for these baselines as described in the paper: specificaly, we conduct 10 completely separate sampling iterations for every dataset and number of shots, and we evaluate on the full test set for each dataset. For logistic regression, we conduct a complete grid search over the L2 regularization parameter over a grid of 51 values (50 regularization values plus no regularization). For catboost, due to the computational expense of training and the time limitations of the response window, we use the default hyperparameters. (Note that recent work has shown that the default hyperparameters of CatBoost are at or near SOTA for tabular classification [7] and that the difference due to tuning from CatBoost tends to be less than the difference between NN-GBDT algorithm selection [8], indicating that tuning does not tend to change the relative rankings of CatBoost vs. NN models).
> >
> > These results show the following:
> > * Logistic regression and CatBoost are both outperformed by TabPFN and XGBoost at nearly all points along the curves.
> > * Logistic regression has surprisingly competitive performance, particularly in the range of 4-16 shots.
> > * CatBoost performs less well across the benchmark, and is the lowest-performing baseline (vs. logistic regression, XGBoost, and TabPFN). We hypothesize that this may be due to the frequency of numeric data across our tasks. We also note that few-shot evaluations of CatBoost are rare or nonexistent in the literature, and the few-shot performance of CatBoost is simply not known -- it may be the case that CatBoost requires relatively larger datasets to reach peak performance. Furthermore, this is connsistent with studies that have found XGBoost to outperform CatBoost [9] (although we note that the relative performance of these two varies across studies and likely reflects nuances in the data and experimental setups across studies).
> >
> > We are grateful to the reviewer for suggesting these additional baselines, and will add them to the paper! We believe that these results also provide further evidence of the effectiveness of our model -- even these additional strong baselines do not outperform our proposed method. We hope that this further supports the reviewers' connclusion that our model achieves "strong performance in zero-shot and few-shot settings."
> >
> > References
> > * [7] Gorishniy, Yury, et al. "Revisiting deep learning models for tabular data." Advances in Neural Information Processing Systems 34 (2021): 18932-18943.
> > * [8] McElfresh, Duncan, et al. "When do neural nets outperform boosted trees on tabular data?." Advances in Neural Information Processing Systems 36 (2024).
> > * [9] Kadra, Arlind, et al. "Well-tuned simple nets excel on tabular datasets." Advances in neural information processing systems 34 (2021): 23928-23941.

---

### Official Review · Reviewer_mEH8 · 2024-07-22

**Soundness:** 3
**Presentation:** 3
**Contribution:** 3
**Rating:** 5
**Confidence:** 4

**Summary:**

The paper introduces TABULA-8B, a language model designed for tabular data prediction. The authors detail the creation of a large, high-quality dataset (T4) from the TabLib corpus, containing over 800 million rows from 1.5 million unique tables. TABULA-8B is fine-tuned from the Llama 3-8B model using techniques for tabular prediction. Extensive evaluation shows that TABULA-8B outperforms state-of-the-art models like XGBoost and TabPFN in both zero-shot and few-shot settings. The authors also discuss the robustness of TABULA-8B, its efficient attention masking scheme, and the potential impact of data contamination.

**Strengths:**

- The creation and use of the T4 dataset, a large-scale, high-quality collection of tabular data, provides a robust foundation for the model’s training and evaluation.
- TABULA-8B demonstrates superior performance in zero-shot and few-shot learning scenarios, outperforming SOTA models like XGBoost and TabPFN.
- The release of the model, code, and data promotes transparency and encourages further research and development in "LLM for tabular data".

**Weaknesses:**

The primary contribution of this article lies in constructing a meticulously curated large-scale corpus, which significantly aids in the application research of large language models (LLMs) in the domain of tabular data. **This can transform into an excellent benchmark paper. However, I think it is not suitable for the main track** for the following reasons:

- Regression tasks are crucial for tabular data prediction, but the paper converts the regression task into a four-class classification task.
- The evaluation of models is limited to the zero-shot and few-shot levels. Many of the datasets used in the benchmarks have data volumes far exceeding the few-shot range. Additionally, few-shot tasks for tabular data introduce significant randomness (e.g., the performance of xgboost models trained on data from different samples can vary greatly), necessitating a sufficient number of repeated samplings.
- The method primarily involves converting tabular data into textual data for NLP tasks. Tabular data tasks lean more towards numerical reasoning rather than text generation. Without leveraging external models, language models will face performance bottlenecks in full-shot scenarios.
- There is a need for more comparison with other relevant work on language models for tabular data prediction, such as CAAFE[1], TP-BERTa[2], FeatLLM[3], and TabLLM[4].

[1] Large language models for automated data science: Introducing caafe for context-aware automated feature engineering. NeurIPS, 2023

[2] Making pre-trained language models great on tabular prediction. ICLR, 2024

[3] Large language models can automatically engineer features for few-shot tabular learning. ICML, 2024

[4] Tabllm: Few-shot classification of tabular data with large language models. AISTATS, 2023

**Questions:**

Please see the weakness.

**Limitations:**

The authors adequately addressed the limitations.

---

> ### Author Response · Authors · 2024-08-03
>
> Thank you for taking the time to carefully read our paper and provide constructive comments that will improve the quality of our manuscript. We also appreciate your recognition of how “TABULA-8B demonstrates superior performance in zero-shot and few-shot learning scenarios”, as well as your belief that our paper will encourage further work in this area. We respond to your comments below, and thank you in advance for your dedication to the review process.
>
> _**Relationship to prior work.**_
>
> We would like to explicitly state that two key differentiators of our approach relative to the related works highlighted by the reviewer are scale and finetuning-free transfer of our method. With respect to scale, TabuLA is trained on four orders of magnitude more tabular data than any existing tabular LLM, including TP-BERTa (202 tabular datasets), TabFM [9] (115 datasets), and UniPredict (169 datasets). Our training set consists of 4.2 million datasets. With respect to training-free transfer, many existing tabular LLMs perform fine-tuning directly on the target dataset “shots”. This includes TabLLM (except in the zero-shot case) and UniPredict. In contrast, our method performs zero training on any downstream task on any setting in the paper. We believe these substantial differences mean that comparisons to these methods are not always informative – especially considering the significant cost of the downstream fine-tuning – and we kindly request that the reviewer consider these differences throughout our discussion. We compare to prior work in more detail below:
>
> - CAAFE [1] and FeatLLM [3]: Both of these papers provide feature engineering and data augmentation methods that can then be used by a base tabular prediction model like logistic regression ([1] uses TabPFN as the downstream classifier). They do not provide models for tabular prediction per se. As such, both of the methods are largely complementary/orthogonal to our work. One could in principle use these methods to engineer more features for a specific table, and then feed this new table into Tabula.
>
> - TP-BERTa [2] and TabLLM [4]: Both of these methods require explicitly fine-tuning a large model on every downstream/eval dataset. This is a significant limitation that has been noted in prior works [6, 7] that we explicitly aim to overcome with our method which only requires a forward pass at test time (no gradient updates). To provide further context, to evaluate TabLLM on 321 benchmark datasets with (0, 1, 2, 3, 4, 8, 16, 32) shots, this would require (321 * 8 = 2568) individual fine-tuning runs of the T-few model (even without performing hyperparameter tuning), which is not computationally feasible.
>
> _**Numerical reasoning**_
>
> Numeric features are indeed an important aspect of tabular data. However, we believe one of the core contributions of our work is to show that LLMs can indeed outperform classical SOTA baselines by treating numbers as text. This simplicity and power is a feature-- not a bug-- that allows us to scale up training to millions of tables, in ways previous work could not. The OpenML benchmarks and a large subset of the Grinsztajn datasets consist primarily of numeric features and our model outperforms TabPFN and XGBoost by a significant margin on these datasets. Please see Figure 4.
>
> _**Full-Shot vs Few Shot**_
>
> The primary focus of our work is to develop new methods of learning that expand the scope of what’s possible in data scarce regimes. That is, the goal of our paper is not to outperform XGBoost when the number of shots is large, but rather when the training set is small. We believe that tabular prediction in low data regimes is a fundamentally important problem. This is particularly true in settings like health, or education, where an institution (i.e a school or hospital) wants to develop a predictor for their specific, local population but lacks a large historical database of cases. We believe that it is clear from our results that Tabula-8B significantly expands the scope of what is possible.

---

> ### Author Response · Authors · 2024-08-03
>
> _**Randomness in Evaluation**_
>
> We share the reviewers’ concern about reliable evaluation and took careful steps to ensure that our estimates were robust to the potential issues related to random selection which the reviewer correctly identifies. We kindly remind the reviewer that, for every baseline method, we conduct 10 completely independent trials at every number of shots for every dataset; we also always evaluate on the full remainder of the data for testing, which allows for very large test sets. The gaps between our method and XGBoost hold robustly across a suite of over 300, independently evaluated, tabular benchmark datasets – an evaluation pool much larger than any of the related works mentioned below (e.g. TabLLM [4] uses less than 25 tables and [5] Unipredict evaluates their model on less than 70).
>
> As such, we believe these estimates provide a strong reliable signal of performance. We also kindly remind the reviewer that 95% Clopper-Pearson intervals are shown in all of the curves in our paper (in most cases they are extremely narrow due to the large test sets used, which is an indication of the low degree of statistical uncertainty of our point estimates); the intervals for TabuLa-8B indicate a high degree of statistical confidence that its performance is not equivalent to any of the baseline methods.  If the reviewer has a specific statistical test that they would like us to perform that they believe would provide further clarity into this question, please let us know.
>
> _**Regression vs. classification:**_
>
> We agree that regression is an important task in tabular prediction! This is part of why we include binned regression tasks in our approach. This is well motivated for a number of reasons. First, it follows the precedent set by prior work in the tabular LLM space that bins real valued targets, e.g.  [5]. Second, as has been classically observed by the learning theory community, one can reduce regression to classification [10]. That is, any algorithm capable of solving classification tasks can also be used to solve regression tasks by using binary search on a series of binned regression tasks. Once again, our method is directly compatible with performing more detailed regression inference out of the box – one can simply repeatedly narrow the “bins” based on the models’ predictions, fit a fixed sample, and make regression predictions to an arbitrary degree of precision – perhaps an advantage over a more rigid regression approach which would only allow for a fixed precision of the outputs. We will more clearly highlight this potential future direction in the discussion and future work sections. We also note that our release of the code, evaluation suite, and pretrained model will enable other researchers to conduct detailed further experiments in this direction as well.
>
> _**Benchmark vs. Main Track:**_
>
> The reviewer states that “This can transform into an excellent benchmark paper. However, I think it is not suitable for the main track”. We are glad the reviewer acknowledges our papers’ contribution. However, we strongly disagree that the paper is not suited for the main track.  A new, fully open model and dataset which enable, as the reviewer states, “superior performance in zero-shot and few-shot learning scenarios, outperforming SOTA models like XGBoost and TabPFN'' is a contribution in line with prior works which have appeared in the main track of NeurIPS and other top AI conferences. For example, TransTab [11] and CAAFE [1] appeared in the NeurIPS main track, [2] appeared in the ICLR main track (spotlight), [4] appeared in AISTATS main track, [3] and [12] in ICML main track. TabPFN [13] also appeared in ICLR main track (notable paper - top 25%) and, as the reviewer notes, our method substantially outperforms TabPFN. We note that [1], [2], [3], and [4] also all use existing pretrained LLMs as the backbone of their models. While not identical to the current work, we believe these papers make comparable contributions in the area of cross-table or few-shot classification for tabular data, and that our models’ “superior performance in zero-shot and few-shot learning scenarios” also justifies a contribution most relevant to the main track, not a benchmark.
>
> We also kindly note that our paper does not propose, or attempt to conduct, a benchmarking study of existing algorithms; we simply aggregate a large set of high-quality tabular benchmarks and use them to compare TabuLa to other relevant methods from the literature.

---

> > ### Author Response · Authors · 2024-08-03
> >
> > _**References**_
> >
> > - [1] Large language models for automated data science: Introducing caafe for context-aware automated feature engineering. NeurIPS, 2023
> > - [2] Making pre-trained language models great on tabular prediction. ICLR, 2024
> > - [3] Large language models can automatically engineer features for few-shot tabular learning. ICML, 2024
> > - [4] Tabllm: Few-shot classification of tabular data with large language models. AISTATS, 2023
> > - [5] Wang, Ruiyu, Zifeng Wang, and Jimeng Sun. "Unipredict: Large language models are universal tabular predictors." arXiv preprint arXiv:2310.03266 (2023).
> > - [6] Fang, Xi, et al. "Large language models (LLMs) on tabular data: Prediction, generation, and understanding-a survey." (2024).
> > - [7] Wen, Xumeng, et al. "From Supervised to Generative: A Novel Paradigm for Tabular Deep Learning with Large Language Models." arXiv e-prints (2023): arXiv-2310.
> > - [8] Yang, Yazheng, et al. "Unleashing the Potential of Large Language Models for Predictive Tabular Tasks in Data Science." arXiv preprint arXiv:2403.20208 (2024).
> > - [9] Zhang, Han, et al. "Towards foundation models for learning on tabular data." arXiv preprint arXiv:2310.07338 (2023).
> > - [10] Torgo, L. and Gama, J., 1996. Regression by classification.
> > - [11] Wang, Zifeng, and Jimeng Sun. "Transtab: Learning transferable tabular transformers across tables." Advances in Neural Information Processing Systems 35 (2022): 2902-2915.
> > - [12] Zhu, Bingzhao, et al. "XTab: cross-table pretraining for tabular transformers." Proceedings of the 40th International Conference on Machine Learning. 2023.
> > - [13] Hollmann, Noah, et al. "TabPFN: A Transformer That Solves Small Tabular Classification Problems in a Second." The Eleventh International Conference on Learning Representations.

---

> ### Comment · Reviewer_mEH8 · 2024-08-13
>
> Thanks for your responses. Some of my concerns remain as follows:
>
> 1. The method's outperforming results are **insufficient to demonstrate that treating numbers as text enables LLMs to fully handle complex numerical reasoning tasks in tabular data**. The improvement could stem from various factors, such as carefully curated datasets during LLM training or the high-quality text transformation of the test datasets. There are many tabular datasets that can't be reasoned by text, such as pump-sensor-data (https://www.kaggle.com/datasets/nphantawee/pump-sensor-data).  **LLMs are unable to make reasonable predictions on these numerical datasets based solely on text**.
>
> 2. Similarly, superior performance alone **is not a sufficient condition to make a significant contribution to the main track**. TransTab has introduced a general transfer learning approach, CAAFE employs LLMs for feature engineering, and TabPFN has pioneered the application of PFN families in tabular data, all of which hold a unique position in the field of tabular data.
>
> 3. Repeatedly narrowing the "bins" based on the models' predictions for regression tasks is a theoretical approach, but it is essentially still a classification method. but achieving the desired precision requires considerable computational resources. I think this approach **cannot be considered a practical solution for regression tasks**.
>
> 4. Since large language models (LLMs) are being used, it would be **unfair not to compare them with recent works that also leverage LLMs**. While a comparison across all benchmarks isn't necessary, I believe a fair comparison on several datasets is essential. For instance, as far as I know, fine-tuning TP-BERTa does not incur an unbearable cost.
>
>
> In summary, the paper's main contribution lies in the meticulous preparation of the dataset and the training of the corresponding language models, which are helpful for LLM tabular prediction. **All the aforementioned concerns represent challenges for LLMs in tabular prediction. However, compared to other methods that serialize tabular data into text for prediction, this paper does not offer sufficient breakthroughs. In terms of field contribution and application scope, I think it may not be entirely suitable for the main track.**

---

> ### Author Response · Authors · 2024-08-14
> **Follow up response to Reviewer mEH8 [1/3]**
>
> Thank you to the reviewer for your thoughtful engagement with our work. The reviewer expresses reservations that we ourselves were curious about when we started this work– the capacity of LLMs to model numeric data; comparison to the strongest possible baselines – and we take these concerns seriously. Indeed, we designed our experiments to address such questions, using rigorous evaluation at a scale beyond any other recent tabular prediction method. We believe that the reviewers’ questions and concerns can be addressed by focusing on different subsets of our results (e.g., tables with numeric data), and that doing so will further improve the paper’s presentation and discussion.
>
> Since we will refer to our evaluation results below, please recall that our evaluations cover 330 tables comprising five high-quality tabular benchmarks (OpenML-CC18, OpenML-CTR23, Grinsztajn, UniPredict (which is comprised entirely of quality-curated Kaggle datasets), and the AutoML Multimodal Benchmark (AMLB)). These benchmarks have been previously proposed and vetted by the tabular data modeling community and are widely used across many studies as performance references for classification and regression on tabular data.
>
>  We offer specific responses to each concern in the reviewer’s comment below.
>
>   # 1. **Performance on numeric data**
>
> We share the reviewer’s understanding that numeric data is a fundamentally important component of tabular data. However, we believe that the reviewers’ response does not  reflect the research goals of our work, nor the strength of our results. We provide a few specific responses regarding numeric data below.
>
>   * 1a. **Objective of this work:** The goal of our work is to demonstrate an end-to-end recipe for transfer learning on tabular data prediction tasks. All reviewers (including mEH8) appear to agree that our work considerably outperforms the current state-of-the-art (SOTA) in the few-shot tabular prediction setting and enables zero-shot prediction not possible with existing tabular models. It is not our goal to demonstrate that LLMs “fully handle complex numerical reasoning tasks in tabular data” – this objective, while important, is both out of scope for the current work and currently difficult to assess. Improving the SOTA on established benchmarks is the most common way to empirically demonstrate progress on research problems in AI. Our results show that we achieve this over both classical methods (such as XGBoost and TabPFN) and LLM-based models (such as variants of Claude). Popular methods like XGBoost and TabPFN are also widely impactful, yet only address prediction problems.
>
>   * 1b. **Results on numeric features:** The reviewer states that “The method's outperforming results are insufficient to demonstrate that treating numbers as text enables LLMs to fully handle complex numerical reasoning tasks in tabular data.” In the submitted version of the paper, we do not differentiate between datasets that contain numeric data and those that do not – we thank the reviewer for pushing us to make this distinction, and we discuss our evaluation results on numeric data below.
>
>     * (i) **Numeric data is prevalent in our evaluation datasets:** We analyzed the distribution of data types across all columns in our evaluation suite. The distribution is as follows. float: 6,637 columns (33.6% of columns),  int: 11,881 columns (60.2% of columns), object: 1,201 columns (6.1% of columns),  bool: 15 columns (<1% of columns). In total, 318 tables (95.8%) contain one or more numeric columns, and 78 tables (23.5%) contain only numeric columns. **Numeric data is thus by far the most prevalent type of data across our evaluation suite, comprising 93.8% of the columns across our evaluation tables** (33.6% float + 60.2% int). These benchmarks thus reflect the reviewers' emphasis on numeric data.
>     * (ii) **Results on tables with numeric data:** To further characterize our model’s performance on numeric data, we provide two additional views of our results below. First, we give the performance on tables in our evaluation suite which contain at least one numeric column (int or float dtype). Second, we give the performance on tables which contain *only* numeric columns. Our results indicate that our model still outperforms all baselines on tasks containing numeric data (related to the previous point). Furthermore, the results show that our model is competitive with (matching or outperforming) baselines even on tables that contain *entirely* numeric data – a setting that advantages TabPFN and XGBoost which can operate in continuous space.
>
> *(continued in following post due to character limits)*

---

> ### Author Response · Authors · 2024-08-14
> **Follow up response to Reviewer mEH8 [2/3]**
>
> **Table A: Accuracy on tables containing 1 or more numeric columns (random baseline: 0.331; average over 266 tasks; Clopper-Pearson confidence intervals are width ≤0.01):**
>
> | Num. Shots | TabuLa-8B | Llama 3 8B (no fine-tuning) | XGBoost trained + tuned on k samples | TabPFN on k samples |
> |------------|-----------|----------------------------|--------------------------------------|---------------------|
> | **0**          | 0.492     | 0                          | N/A                                  | N/A                 |
> | **1**          | 0.535     | 0.376                      | 0.403                                | N/A                 |
> | **2**          | 0.551     | 0.406                      | 0.423                                | 0.351               |
> | **3**          | 0.563     | 0.414                      | 0.420                                 | 0.397               |
> | **4**          | 0.569     | 0.423                      | 0.433                                | 0.424               |
> | **8**          | 0.598     | 0.436                      | 0.494                                | 0.503               |
> | **16**         | 0.623     | 0.459                      | 0.57                                 | 0.58                |
>
> **Table B: Accuracy on tables composed entirely of numeric columns (random baseline: 0.437; average over 51 tasks; Clopper-Pearson confidence intervals are width ≤0.025):**
>
> | Num. Shots | TabuLa-8B | Llama 3 8B (no fine-tuning) | XGBoost trained + tuned on k samples | TabPFN on k samples |
> |------------|-----------|----------------------------|--------------------------------------|---------------------|
> | **0**          | 0.486     | 0                          | N/A                                  | N/A                 |
> | **1**          | 0.535     | 0.45                       | 0.521                                | N/A                 |
> | **2**          | 0.551     | 0.483                      | 0.555                                | 0.448               |
> | **3**          | 0.563     | 0.498                      | 0.54                                 | 0.511               |
> | **4**          | 0.565     | 0.505                      | 0.555                                | 0.518               |
> | **8**          | 0.592     | 0.518                      | 0.579                                | 0.588               |
> | **16**         | 0.619     | 0.546                      | 0.637                                | 0.637               |
>
>   * 1c. **Additional dataset:** The reviewer provides a link to an additional Kaggle dataset. This is useful context regarding the types of data the reviewer feels would be useful, thank you! This dataset (54 feature columns, 98.1% numeric) closely resembles our evaluation suite's composition (93.8% numeric columns).  Please refer to the results on tables containing 1 or more numeric columns (since this table is not entirely numeric) above, which show that our model achieves SOTA zero- and few-shot performance, outperforming the baselines, on tables of similar composition. We believe that performance on these high-quality benchmarks is a more reliable indicator than performance on a single Kaggle dataset.
>
> # 2. Main track vs. datasets and benchmarks
>
> We are glad that the reviewer acknowledges how our method does indeed improve upon the state-of-the-art for tabular prediction, and is working to ensure that our paper is published in the correct venue. We share this objective.
>
> We believe that there is a strong and established precedent that works like ours which introduces a new core methodology (i.e, a web-scale dataset and new training recipe for tabular data) and significantly expands what’s possible on a fundamental ML problem (tabular prediction) should appear in the main track.
>
> Please see our response above with numerous examples of similar papers that appeared in the main track of NeurIPS, ICML, and ICLR. The reviewer says that “superior performance alone is not a sufficient condition to make a significant contribution to the main track”; however, we feel (and all reviewers, this one included) have acknowledged that our work does more than simply present “superior performance” for an existing method. As a further example, consider Kadra et al., “Well-tuned simple nets excel on tabular datasets”, NeurIPS main track 2021 – which is also a purely empirical tabular data study that shows that a simple, pre existing method (MLP with carefully tuned regularization) achieves superior performance with no new model or algorithm.

---

> > ### Author Response · Authors · 2024-08-14
> > **Follow up response to Reviewer mEH8 [3/3]**
> >
> > Additionally, we acknowledge that there is some ambiguity between D&B track and main track papers. Indeed, even the D&B call for papers acknowledges this (https://neurips.cc/Conferences/2023/CallForDatasetsBenchmarks), and it is possible that our work could also be a fit for this track. However, we emphasize, as previously, that the main contribution is not strictly “a new dataset, benchmark, or other work that falls into the scope of the track” (to quote the D&B frequently asked questions), and so we feel it is best suited for the main track.
> >
> > # 3. Regression tasks
> >
> > The reviewer claims that our method would be computationally infeasible to extend to regression tasks.
> >
> >  First, our paper is focused on classification where, as the reviewer points out, we make a significant advance.
> >
> > Second, we do not agree with the reviewer’s claim that our method “cannot be considered a practical solution for regression tasks.” Practicality is a subjective judgment that depends on context; we do not believe that any highly-effective prediction method can be dismissed purely on these grounds. The existence of many widely-used LLMs with size equal or greater than our method seems to indicate that, for at least some applications, users consider access to computationally-intensive models to be a practical solution. We believe there are likely to be users who consider significant improvements in predictive performance achieved by our model to be worth the computational cost, including in settings where they may perform multiple forward passes to iteratively refine predictions. As we mention in the supplement, prediction currently takes roughly one second for a single sample; even a 10x increase in this latency (due to repeated predictions at higher granularity for a specific example) would still put the model on par with many commercial LLMs, which have response latency in the range of seconds.
> >
> > # 4. Comparison to recent works leveraging LLMs
> >
> > As part of our author response, we do compare against state-of-the-art commercial LLMs like Claude that have been pre-trained on huge amounts of data. Our method significantly outperforms both Claude variants evaluated, as acknowledged by the other reviewers.  We encourage the reviewer to check the rebuttal PDF for these results.
> >
> > The reviewer specifically mentioned TP-BERTa, which we discuss in the bullets below.
> >
> > * As we mentioned in our initial response, **TP-BERTa requires fine-tuning on every individual dataset. Due to this, we do not consider this method to be an applicable baseline.** As we mention in the previous response, TP-BERTa would require 2,568 individual fine-tuning runs of our 8B parameter model for evaluation (even without accounting for hyperparameter tuning or multiple runs to account for randomness over the selected shots).  We feel that this scale is not feasible for a baseline that is not directly comparable to our work.
> >
> > * **TP-BERTa is missing the prediction code to do inference/evaluation.** An issue flagging this has been open on the repository for over one month https://github.com/jyansir/tp-berta/issues/3 , which currently makes reproducing the TP-BERTa inference/evaluation procedure impossible.
> >
> > * **TP-BERTa does not support multiclass classification**, which are a significant component of our evaluations and would again make it not directly comparable to our model.
> >
> > # Conclusion
> >
> > Once again, we are extremely grateful for their thoughtful engagement with the work and their expert assessment and recommendations for improving it. We appreciate the reviewer’s commitment to constructive dialogue – your suggestions will considerably improve the paper.

---

> ### Comment · Reviewer_mEH8 · 2024-08-14
>
> Dear Author,
>
> Thanks for your detailed reply.
>
> Although I still think the issue of "applying it to regression tasks" and "comparing it to more LLM-related work" still exists. (I’m not suggesting that you need to perform comparisons on numerous datasets; you can select some representative datasets for comparison.) **I decide to raise my score to 5, showing my encouragement.**
>
> I hope the authors can make improvements in these two aspects, especially with respect to regression tasks that require precise numerical reasoning. This will make the contribution of the paper more comprehensive.
>
> Regarding the Kaggle dataset I mentioned, it is an example of a case where I hope you can address the potential failures of LLMs.
>
> I am very eager to see this paper improved into a comprehensive LLM-tabular work.
>
> Best wishes,
>
> The reviewer

---

### Author Rebuttal · Authors · 2024-08-03

Thank you to all the reviewers and the AC for their time and dedication to review our paper, we have responded to all the reviewers individually. However, we have run a number of new experiments and we report the results in the figures attached in the pdf here.

Update August 4th: We have included a new figure (C), that includes extra baselines (logistic regression and catboost) in our few shot learning comparisons. We find that Logistic Regression is comparable to XGBoost in the few shot regime, but both are still significantly outperformed by Tabula-8b

Update August 5th: We have updated figure C to include comparisons to top end commercial LLMs like Claude.

---

### Author Response · Authors · 2024-08-12
**Request from authors to review author responses**

Dear reviewers,

Thank you once again for your thoughtful and insightful engagement with our work during the review process. We sincerely believe that each reviewers' expert feedback has improved our work considerably, and that the additional results and discussion generated during the review process have only strengthened the clarity of our paper's results in zero- and few-shot transfer learning for tabular data and clarified the significance of its contributions in this area.

Given that the author response window is closing soon, we would like to follow up on our previous responses to encourage reviewers to review our detailed responses. Please let us know if there is additional clarification that we can provide while the author dialogue window is still open. Several reviewers requested additional results or clarifications, which have been provided in our responses, and we would greatly appreciate your review of our responses and the opportunity to continue to provide clarification as time allows. Additionally, if our responses have properly addressed your concerns, please consider adjusting your score to reflect your current assessment of our paper's contributions.

Warmly,

Paper authors

---

> ### Comment · Area_Chair_n5Uf · 2024-08-13
> **Please engage the discussion.**
>
> Dear Reviewer 3ko2 and mEH8,
>
> Please read the authors' response and give your comments!
>
> Thanks

---

> > ### Comment · Reviewer_mEH8 · 2024-08-13
> >
> > Dear AC and authors,
> >
> > Thank you for the reminder. As I am currently working, I will reply within eight hours.

---

### Decision · Program_Chairs · 2024-09-25

**Decision:**

Accept (poster)

**Comment:**

This paper presents a language model designed for tabular data prediction. It is trained using a large, high-quality dataset (T4) from the TabLib corpus, which contains over 800 million rows from 1.5 million unique tables. The model can readily be used for few-shot and zero-shot learning and achieve much better accuracy than baselines.

Reviewers raised several questions including the experiment design and concerns of low novelty. After rebuttal and discussion, several reviewers increase their scores.